# The elite haplotype *OsGATA8*-H coordinates nitrogen uptake and productive tiller formation in rice

Wei Wu[1,9], Xiaoou Dong[1,9], Gaoming Chen[1], Zhixi Lin[1], Wenchao Chi[1], Weijie Tang ⓞ[1], Jun Yu[1], Saisai Wang[1], Xingzhou Jiang[1], Xiaolan Liu[1], Yujun Wu[1], Chunyuan Wang[1], Xinran Cheng[1,2,3], Wei Zhang[4], Wei Xuan ⓞ[5], William Terzaghi ⓞ[6], Pamela C. Ronald ⓞ[7,8], Haiyang Wang ⓞ[2], Chunming Wang ⓞ[1,3] ✉ & Jianmin Wan ⓞ[1,2] ✉

Excessive nitrogen promotes the formation of nonproductive tillers in rice, which decreases nitrogen use efficiency (NUE). Developing high-NUE rice cultivars through balancing nitrogen uptake and the formation of productive tillers remains a long-standing challenge, yet how these two processes are coordinated in rice remains elusive. Here we identify the transcription factor OsGATA8 as a key coordinator of nitrogen uptake and tiller formation in rice. OsGATA8 negatively regulates nitrogen uptake by repressing transcription of the ammonium transporter gene *OsAMT3.2*. Meanwhile, it promotes tiller formation by repressing the transcription of *OsTCP19*, a negative modulator of tillering. We identify *OsGATA8*-H as a high-NUE haplotype with enhanced nitrogen uptake and a higher proportion of productive tillers. The geographical distribution of *OsGATA8*-H and its frequency change in historical accessions suggest its adaption to the fertile soil. Overall, this study provides molecular and evolutionary insights into the regulation of NUE and facilitates the breeding of rice cultivars with higher NUE.

Nitrogen is an essential macronutrient vital for plant growth and development. Insufficient nitrogen fertilizer in the soil can severely restrict crop growth, while overapplication of nitrogen fertilizers negatively impacts the environment[1,2]. To increase crop productivity in a sustainable fashion, there is increasing interest in breeding cultivars with high nitrogen use efficiency (NUE). As a major determinant of crop yield, plant NUE is an inherently complex trait governed by multiple intertwined biological processes, including nitrogen uptake, transport, assimilation and remobilization[3–5]. The identification of key genetic components involved in NUE regulation holds great promise for crop improvement.

Rice serves as a major staple crop for over half of the world's population. Rice yield is largely influenced by the number of effective panicles per unit of land area[6]. The introduction of the semi-dwarf gene *sd1* into modern rice cultivars during the first 'Green Revolution' and the application of synthetic nitrogen fertilizers since the 1960s greatly improved lodging resistance and increased yield in modern rice cultivars[6,7]. Nevertheless, the large amount of nitrogen fertilizers applied during rice production increases the carbon footprint associated with fertilizer production and may accelerate environmental degradation due to the run-off of excessive fertilizers into waterways[1,2]. In addition, excessive nitrogen promotes the formation of tillers that fail to bear effective panicles, known as nonproductive tillers. Nonproductive tillers channel nutrients away from grain production, while a larger proportion of productive tillers in rice is often associated with higher NUE[8,9]. Thus, breeding high-NUE rice cultivars with a high

proportion of productive tillers is an essential route to sustainable agriculture.

Over the past few decades, extensive efforts have been devoted to dissecting the molecular basis of nitrogen uptake and NUE in rice[10–12]. OsNRT1.1B[13] and OsNPF6.1 (ref. 14) were discovered as the major nitrate transporters in rice, whose overexpression leads to enhanced nitrate uptake. The transcription factor nitrogen-mediated tiller growth response 5 (NGR5) promotes tiller formation upon nitrogen perception in the absence of gibberellins (GA) signaling[10]. The transcription regulator OsTCP19 inhibits tiller formation, whose activity varies in response to nitrogen availability[11]. Despite these advancements, the molecular mechanisms coordinating nitrogen uptake and productive tiller formation remain unclear, hindering the molecular breeding of rice cultivars with higher NUE and yield.

In this study, through the identification and characterization of the transcription factor OsGATA8, we uncover a mechanistic connection between NUE and productive tiller formation. We show that OsGATA8 coordinates nitrogen uptake and tiller formation in rice through transcriptional regulation of key components involved in these two biological processes. For breeding application, we identify the elite haplotype OsGATA8-H, which confers high nitrogen uptake efficiency (NUpE) while promoting the formation of productive tillers. These results provide insights into how nitrogen uptake and the formation of productive tillers are coordinated in rice and demonstrate a strategy for balancing these two processes to breed rice cultivars with high NUE and yield.

## Results

### OsGATA8 was identified as a putative negative regulator of NUE

From an agronomical perspective, NUE in rice is defined as grain yield divided by the amount of nitrogen input[3,15]. NUE is often correlated with plant height and the number of tillers. Increasing nitrogen application promotes stem elongation and tiller formation, but overapplication of nitrogen promotes excessive tiller formation, especially nonproductive tillers (Extended Data Fig. 1 and Supplementary Fig. 1). Thus, the productive tiller number ratio (PTNR; productive tiller number low nitrogen (LN) condition/productive tiller number under high nitrogen (HN) condition) and plant height ratio (PHR; plant height under LN condition/plant height under HN condition) are often used as proxies of NUE[16]. PTNR is also called effective panicle number ratio in previous studies[14,16]. Accordingly, NUE is often assessed by measuring the yield per plant (YPP), PHR and PTNR[14,16]. Rice with high PTNR and PHR tended to have higher NUE compared to rice with low PTNR and PHR under both HN and LN conditions (Supplementary Fig. 2). Previously, we designed a genome-wide association study (GWAS)-based strategy to uncover candidate NUE genes followed by functional validation through genetic complementation tests[14,17]. Using PTNR and PHR as the proxies of NUE on a core collection of rice germplasm consisting of 117 varieties[14], we identified a major quantitative trait locus in linkage disequilibrium block between the coordinates 13,548,357 and 13,572,267 on chromosome 1 (Extended Data Fig. 2a and Supplementary Table 1). In total, 51 SNPs causing missense mutations were identified in the three genes present in this linkage disequilibrium block (Supplementary Table 2). We quantified the expression of these three genes in response to different nitrogen concentrations in rice seedlings using qRT–PCR and found that only *Os01g0343300* was significantly induced by high concentrations of nitrogen (1 mM and 5 mM $NH_4NO_3$; Extended Data Fig. 2b). *Os01g0343300* encodes *OsGATA8* (ref. 18), a member of the GATA family transcription factors, which are widespread among eukaryotes and participate in diverse biological processes[19]. *OsGATA8* tends to be expressed at lower levels in rice cultivars with higher PTNR and PHR compared with rice cultivars with lower PTNR and PHR (Extended Data Fig. 2c), indicating a negative correlation between *OsGATA8* expression level and NUE. Confocal microscopy using rice protoplasts expressing

*OsGATA8-GFP* revealed that OsGATA8 is localized to the nucleus (Supplementary Fig. 3a). We characterized the tissue-wide expression pattern of *OsGATA8* in various rice tissues after the flowering stage using qRT–PCR assays and found that *OsGATA8* is highly expressed in roots and tiller buds (Supplementary Fig. 3b), suggesting that OsGATA8 may have a role in these tissues.

To verify the role of *OsGATA8* in NUE, we generated *OsGATA8* knockout plants using CRISPR–Cas9 in two *japonica* rice cultivars Nipponbare (Nip; cr1, cr2 and cr3) and Zhonghua 11 (cr4), as well as *OsGATA8* overexpression lines with the constitutive *ZmUbi1* promoter in the Nip background (OE1, OE2 and OE3) and evaluated their phenotypes under LN and HN field conditions at the maximum tillering stage and the mature stage (Fig. 1a,b and Supplementary Fig. 3c,d). Compared with the wild-type (WT) plants, both PTNR and the proportion of productive tillers were higher in the *OsGATA8* knockout plants but were lower in the *OsGATA8* overexpression plants (Fig. 1c–f and Supplementary Fig. 3e–g). These results indicate that OsGATA8 is a negative regulator of PTNR and the proportion of productive tillers.

### The *OsGATA8–OsAMT3.2* module restricts nitrogen uptake

To investigate how OsGATA8 downregulates NUE, we examined the total nitrogen content in the shoots and roots of the knockout line *OsGATA8*-cr1 and WT seedlings grown under LN and HN conditions. The nitrogen content of the *OsGATA8*-cr1 line was significantly higher than that of the WT in both shoots and roots under both LN and HN conditions (Extended Data Fig. 3a). Consistently, a higher $^{15}NH_4^+$ influx rate was observed in the *OsGATA8*-cr1 plants under both HN (2.0 mM) and LN (0.2 mM) conditions compared with WT plants (Extended Data Fig. 3b). On the contrary, we did not observe any effect of knocking out *OsGATA8* on $^{15}NO_3^-$ influx rate (Extended Data Fig. 3b). These results suggest that *OsGATA8* regulates the uptake of ammonium but not nitrate.

To gain insights into how *OsGATA8* affects ammonium uptake in rice, we performed an RNA-sequencing (RNA-seq) analysis on seedlings of WT and the *OsGATA8* knockout lines and identified 619 differentially expressed genes (DEGs; Supplementary Table 3), among which 19 genes were annotated to the term 'transporter activity' (GO:0005215) in Gene Ontology analysis (Extended Data Fig. 3c). Among them, *OsAMT3.2* was the only ammonium transporter gene (Extended Data Fig. 3d). Notably, knocking out *OsGATA8* significantly increased the expression of *OsAMT3.2* (Fig. 2a). Consistently, qRT–PCR assay verified that the expression of *OsAMT3.2* was significantly upregulated in the *OsGATA8* knockout lines but was repressed in the *pOsGATA8::OsGATA8* transgenic lines (Fig. 2b). Moreover, a DNA affinity purification sequencing (DAP-seq) analysis identified *OsAMT3.2* as a putative direct target of OsGATA8 (Supplementary Table 4 and Extended Data Fig. 3e). Furthermore, electrophoretic mobility shift assays (EMSA) and luciferase (LUC) assays in rice protoplasts showed that OsGATA8 directly binds to the promoter of *OsAMT3.2* to repress its transcription (Fig. 2c–e and Supplementary Fig. 4). The direct interaction between OsGATA8 and the promoter of *OsAMT3.2* was further verified by a chromatin immunoprecipitation-quantitative PCR (ChIP–qPCR) assay using shoot tissue of *p35S::Flag-OsGATA8* transgenic plants (Fig. 2c,f). In addition, DAP-seq analysis showed that another ammonium transporter gene, *OsAMT1.2*, is also a putative target of OsGATA8 (Extended Data Fig. 3e). qRT–PCR, EMSA, LUC and ChIP–qPCR assays showed that OsGATA8 also represses the transcription of *OsAMT1.2* by directly binding to its promoter (Supplementary Figs. 4 and 5). These results suggest that *OsGATA8* represses ammonium uptake in rice by downregulating the expression of *OsAMT3.2* and *OsAMT1.2*.

Ammonium transporters in rice are important membrane proteins involved in the uptake, transport and allocation of ammonium in plants[20]. Phylogenetic analysis of the nine ammonium transporters in rice shows that they are mainly divided into the following two subfamilies: AMT1 and AMT2. *OsAMT3.2* and *OsAMT1.2* belong to the AMT2 and AMT1 subfamilies, respectively[21] (Supplementary Fig. 6).

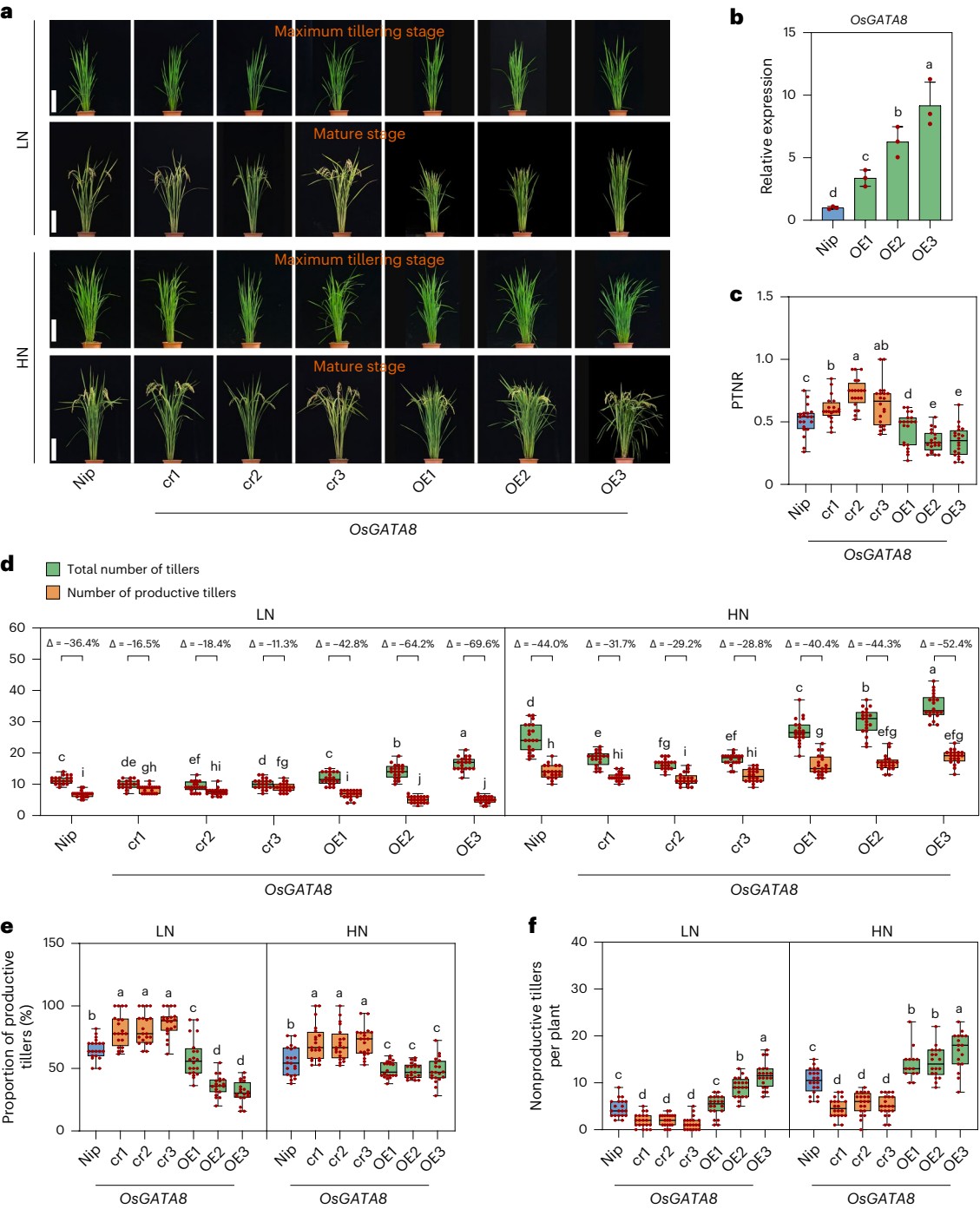

**Fig. 1 | OsGATA8 negatively regulates PTNR and the proportion of productive tillers in rice. a**, The phenotypes of Nip, the *OsGATA8*-cr knockout lines and the *OsGATA8* overexpression lines at the maximum tillering stage (about 40 days after transplanting) and the mature stage under LN and HN conditions. Scale bars, 20 cm. LN, 75 kg ha$^{-1}$ net nitrogen; HN, 300 kg ha$^{-1}$ net nitrogen. **b**, Relative expression of *OsGATA8* in Nip and *OsGATA8* overexpression lines. Total RNA was extracted from the root tissue of 2-week-old seedlings. Each analysis was repeated with three independent root tissues of seedlings. Data are presented as mean ± s.d. **c**, The PTNR of Nip, the *OsGATA8*-cr knockout lines and the *OsGATA8* overexpression line at the mature stage under LN and HN conditions. *n* = 20 plants. PTNR, productive tiller number under LN condition/productive tiller number under HN condition. **d**, The total number of tillers and productive tillers of Nip, the *OsGATA8*-cr knockout lines and the *OsGATA8* overexpression lines under LN and HN conditions. 'Δ' represents the percentage difference compared with the total number of tillers. *n* = 20 plants. **e**,**f**, The proportion of productive tillers (PT%; **e**) and the number of nonproductive tillers (**f**) of Nip, the *OsGATA8*-cr knockout lines and the *OsGATA8* overexpression lines under LN and HN conditions. *n* = 20 plants. In **b**–**f**, different letters indicate significant differences (*P* < 0.05, one-way ANOVA, Duncan's new multiple range test); for *P* values, see source data. In **c**–**f**, box plots denote the 25th percentile, the median and the 75th percentile, with minimum to maximum whiskers.

It was previously reported that knocking out or overexpressing *OsAMT1.2* alone did not affect ammonium uptake in rice, nor did it incur any phenotypic change[22,23]. Thus, we focused on testing the role of *OsAMT3.2* in nitrogen uptake in rice. Notably, *OsAMT3.2* and *OsGATA8* display similar tissue-wide and nitrogen-responsive expression patterns (Extended Data Fig. 4a,b), suggesting a plausible *OsGATA8*–*OsAMT3.2*

molecular module in the regulation of nitrogen uptake. To test the role of *OsAMT3.2* in regulating NUE, we constructed two *OsAMT3.2* knockout lines (cr1 and cr2) with different mutational sites and two *OsAMT3.2* overexpression lines (OE1 and OE2; Extended Data Fig. 4c,d). We measured nitrogen content and $^{15}NH_4^+$ influx rates in seedlings of the *OsAMT3.2* knockout and overexpression lines and found that *OsAMT3.2* positively regulates ammonium uptake in rice (Extended Data Fig. 4e,f). Consistently, the number of productive tillers, biomass and yield decreased in the *OsAMT3.2*-cr1 and *OsAMT3.2*-cr2 lines, but increased in the *OsAMT3.2*-OE1 and *OsAMT3.2*-OE2 lines, compared with the WT (Supplementary Fig. 7a–e). NUpE and nitrogen utilization efficiency (NUtE) are two major factors determining the overall NUE[3]. Therefore, we quantified NUE, NUpE and NUtE in rice plants at the mature stage under LN and HN conditions and found that NUE, NUpE and NUtE all decreased in the *OsAMT3.2*-cr1 and *OsAMT3.2*-cr2 lines, but increased in the *OsAMT3.2*-OE1 and *OsAMT3.2*-OE2 lines, compared with the WT (Supplementary Fig. 7f–h), thus verifying that *OsAMT3.2* promotes high NUE.

To genetically test whether *OsAMT3.2* functions downstream of *OsGATA8*, we generated the *OsGATA8*-cr1/*OsAMT3.2*-cr1 double mutant by crossing *OsGATA8*-cr1 with *OsAMT3.2*-cr1. The homozygous *OsGATA8*-cr1/*OsAMT3.2*-cr1 line showed decreased PTNR, reduced number of productive tillers and reduced proportion of productive tillers, compared with the *OsGATA8*-cr1 line (Fig. 2g–j), suggesting that *OsAMT3.2* functions downstream of *OsGATA8*. These results indicate that *OsGATA8* negatively regulates nitrogen uptake by repressing *OsAMT3.2* expression.

### The *OsGATA8*–*OsTCP19* module promotes tillering

An intriguing observation with the *OsGATA8* overexpression lines was its excessive tiller production (Fig. 1d and Extended Data Figs. 1b and 5a–c). This is unexpected as reduced nitrogen uptake in the *OsGATA8* overexpression lines due to reduced expression of *OsAMT3.2* (Fig. 2b) would presumably reduce tiller formation. We thus hypothesized that *OsGATA8* may affect tiller development by regulating additional targets besides *OsAMT3.2*. We thus examined the expression levels of five candidate genes that have been reported to be involved in rice tillering formation and nitrogen response, including *TB1* (ref. 24), *DLT*[11], *OsNGR5* (ref. 10), *OsTCP19* (ref. 11) and *OsMADS57* (ref. 25) in the *OsGATA8*-cr1 and *pOsGATA8::OsGATA8* overexpression plants. We found that the expressions of *OsTB1* and *OsTCP19* negatively correlated with the expression level of *OsGATA8* (Supplementary Fig. 8). Based on the results of DAP-seq, OsGATA8 primarily recognizes the TTCCKAATTTT (K represents T or G or A) motif (Supplementary Fig. 9), which exists only in the promoter of *OsTCP19* among the five candidate genes examined (Fig. 3a), which suggests that *OsTCP19* is a

direct target for transcription regulation by OsGATA8. In situ hybridization assay showed that *OsGATA8* and *OsTCP19* were both expressed in the shoot apical meristem (Supplementary Fig. 10). EMSA and LUC assay in the rice protoplasts showed that OsGATA8 directly binds to the promoter of *OsTCP19* and represses its transcription (Fig. 3b,c). The direct association between OsGATA8 and the promoter of *OsTCP19* was further validated by a ChIP–qPCR assay using young tillers of the *p35S::Flag-OsGATA8* transgenic plants at the four-leaf stage (Fig. 3d).

To determine the genetic relationship between *OsGATA8* and *OsTCP19*, we generated *OsGATA8*-cr/*OsTCP19*-cr double-knockout lines (Fig. 3e and Supplementary Fig. 11a,b). Consistent with the putative role of OsGATA8 as a transcriptional repressor of *OsTCP19*, the *OsGATA8*-cr/*OsTCP19*-cr double-mutant lines in backgrounds of Nip and ZH11 exhibited decreased PTNR, increased nonproductive tillers and decreased proportion of productive tillers compared with *OsGATA8*-cr under both LN and HN conditions (Fig. 3e–h and Supplementary Fig. 11). These results suggest that OsGATA8 promotes the formation of nonproductive tillers by directly repressing the expression of *OsTCP19*.

### Natural variation in the *OsGATA8* promoter influences NUE

To locate the causal genetic variations in *OsGATA8* that affect NUE, we analyzed the sequence of *OsGATA8* in 117 rice varieties. Twenty-one insertions and deletions (InDels) and 108 SNPs were detected in the promoter of *OsGATA8*, but no variation was found in the coding region of *OsGATA8* (Supplementary Tables 5 and 6). We resequenced the promoter of *OsGATA8* and conducted an association analysis with the variants we identified. Association analysis revealed that a 245-bp deletion (chr1: 13,569,676) is significantly associated with PTNR ($P = 1.78 \times 10^{-5}$; Supplementary Fig. 12 and Supplementary Tables 5, 7 and 8). Given the association of the 245-bp deletion in the promoter of *OsGATA8* with higher NUE, we classified the 117 rice cultivars into the following two groups based on the presence or absence of the 245-bp deletion: group 1 with HapH (*OsGATA8*-H exhibiting high NUE, 14 varieties) and group 2 with HapL (*OsGATA8*-L displaying low NUE, 103 varieties; Supplementary Fig. 12).

To test whether the presence or absence of the 245-bp sequence affects the transcription activity of the *OsGATA8* promoter, we generated constructs carrying promoter variants with sequence features from *OsGATA8*-H or *OsGATA8*-L and performed an LUC reporter assay in rice protoplasts. Our results confirmed that only the 245-bp deletion (888 bp upstream of the start codon) affects the activity of the *OsGATA8* promoter (Extended Data Fig. 6a,b). To verify the effect of the 245-bp deletion in planta, we generated a genome-edited mutant line (D403) carrying a 403-bp deletion in the *OsGATA8* promoter spanning the

**Fig. 2 | OsGATA8 negatively regulates nitrogen uptake by repressing the transcription of *OsAMT3.2*. a**, The relative expression of two DEGs in WT and *OsGATA8* knockout lines. The color key (blue to red) indicates gene expression as fold changes (fragments per kilobase of exon model per million mapped fragments (FPKM)). For each gene, the minimum FPKM value was set as 1.00. **b**, Relative expression of *OsAMT3.2* in the *OsGATA8* knockout lines and the *OsGATA8* overexpression lines driven by native promoter. Total RNA was extracted from the root tissue of 2-week-old seedlings. Values represent mean ± s.d. derived from three independent seedlings. **c**, Schematic diagram of *OsAMT3.2* displaying the promoter and the transcribed region. Horizontal bars indicate the location of the probes used in the EMSA. P1 and P2 of *OsAMT3.2* correspond to the predicted OsGATA8 binding motifs, while P3 of *OsAMT3.2* is a negative control site without predicted OsGATA8 binding motifs. **d**, An EMSA testing the binding strength of OsGATA8 to the predicted binding motifs in *OsAMT3.2* promoters using probes as shown in **c**. MBP, maltose-binding protein. The results are representative of three independent experiments. **e**, LUC assay in rice protoplasts on the effect of OsGATA8 on the transcription of *OsAMT3.2*. Values represent mean ± s.d. derived from three independent samples of rice protoplasts. **f**, ChIP–qPCR

assay of the interaction between OsGATA8 and the promoter of *OsAMT3.2* in the shoot of *35S::Flag-OsGATA8* transgenic plants at the four-leaf stage. Values represent mean ± s.d. derived from three independent samples; *P* values were calculated with two-tailed Student's *t* test. **g**, Phenotypes of Nip, the *OsGATA8* knockout mutant, the *OsAMT3.2* knockout mutant and the *OsGATA8/OsAMT3.2* double-knockout mutant at the maximum tillering stage (about 40 days after transplanting) and the mature stage under LN and HN conditions. LN, 75 kg ha⁻¹ net nitrogen; HN, 300 kg ha⁻¹ net nitrogen. Scale bars, 20 cm. **h**, PTNR of the genotypes in **g** under LN and HN conditions. *n* = 10 plants. PTNR, productive tiller number under LN condition/productive tiller number under HN condition. **i**,**j**, The total number of tillers, productive tillers (**i**) and proportion of productive tillers (PT%; **j**) of the genotypes in **g** under LN and HN conditions. 'Δ' represents the percentage difference compared with the total number of tillers. *n* = 20 plants. In **b** and **h**–**j**, different letters indicate significant differences (*P* < 0.05, one-way ANOVA, Duncan's new multiple range test). For *P* values, see source data. In **e** and **f**, significant difference was determined by two-tailed Student's *t* test. In **h**–**j**, box plots denote the 25th percentile, the median and the 75th percentile, with minimum to maximum whiskers.

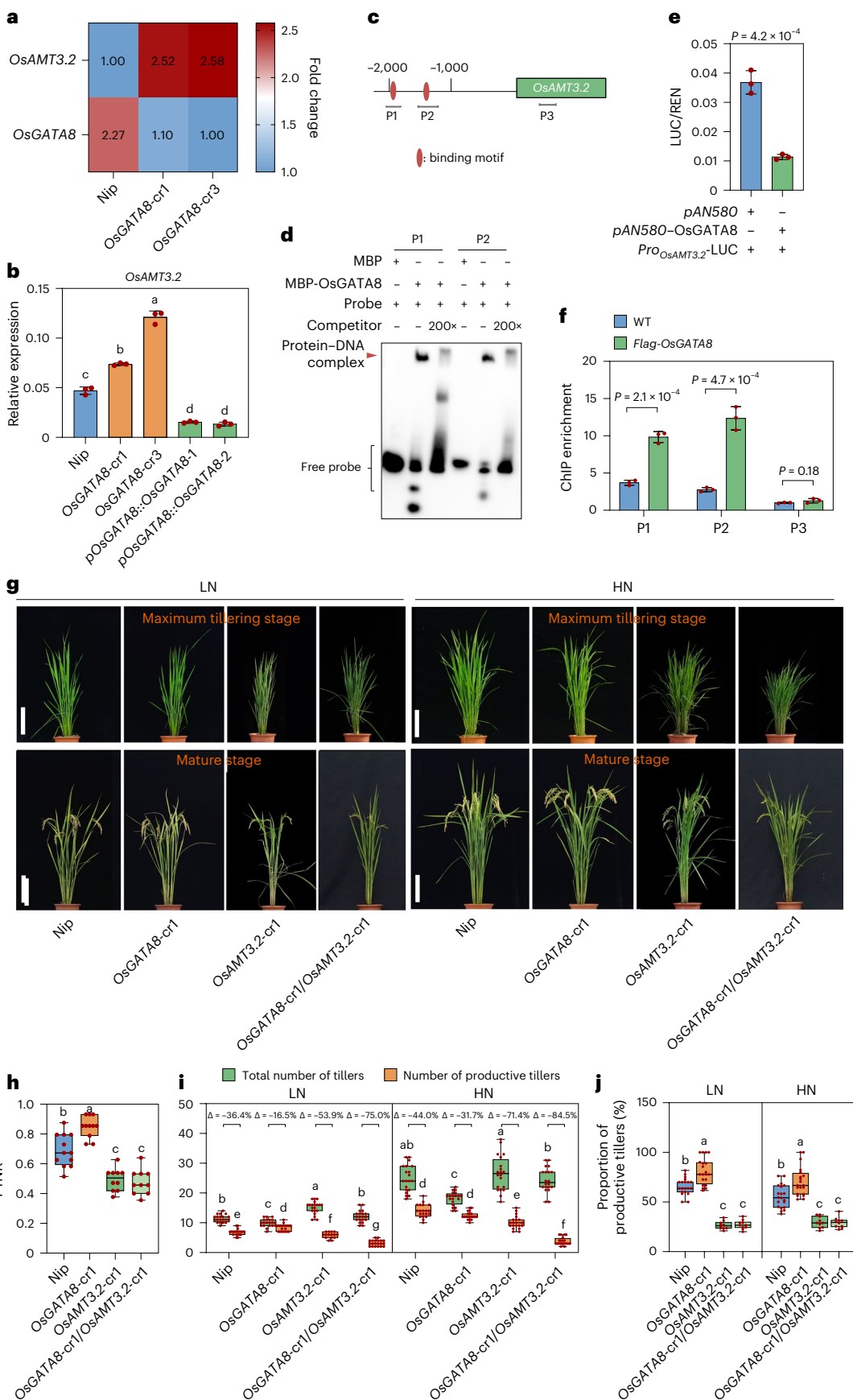

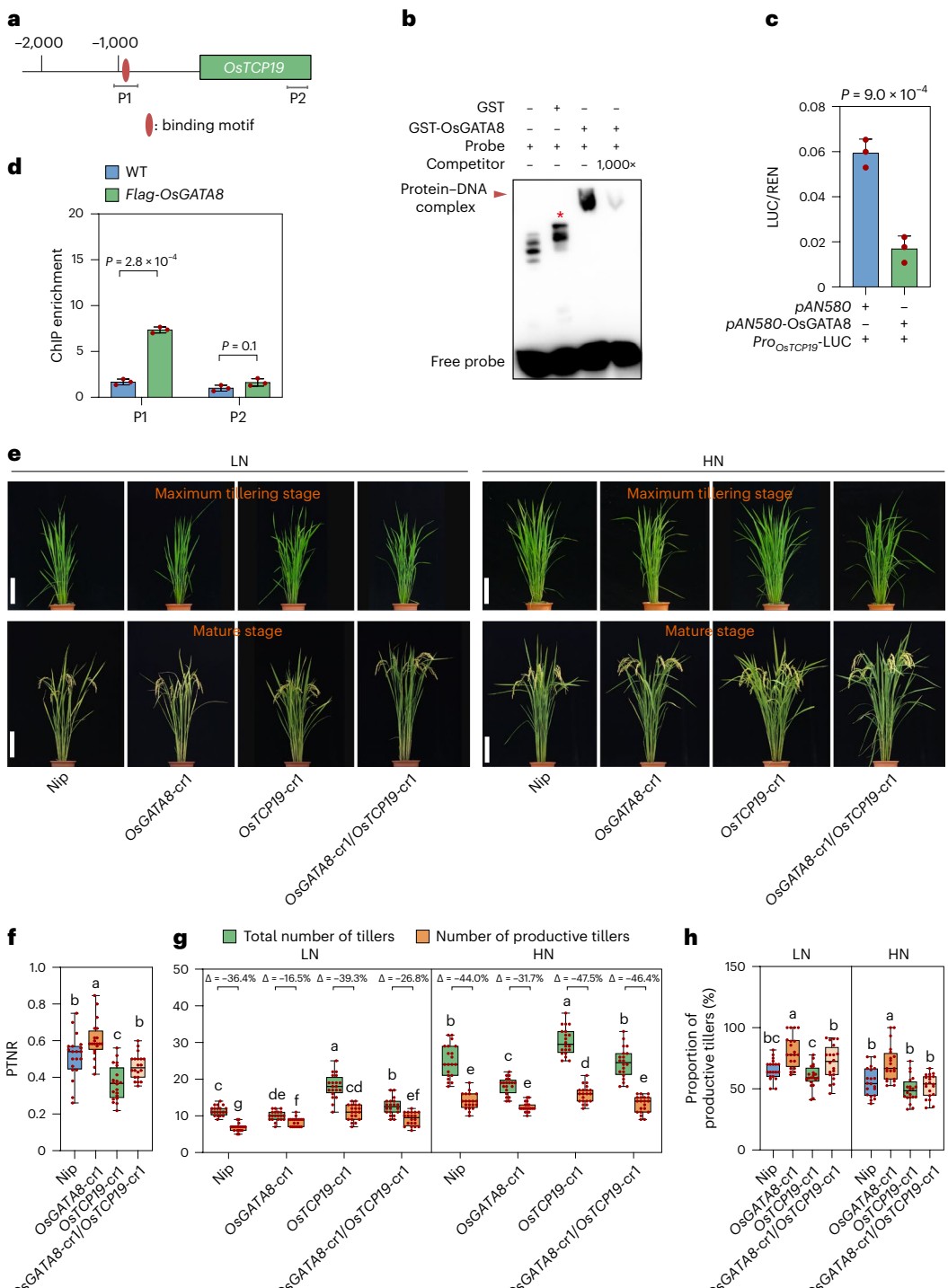

**Fig. 3 | OsGATA8 promotes nonproductive tillers by transcriptionally repressing *OsTCP19*. a**, Schematic diagram of *OsTCP19* with the promoter and transcribed region. Horizontal bars indicate the location of the probes used in the EMSA. P1 corresponds to the predicted OsGATA8 binding motif, while P2 is a negative control in the coding region without the predicted OsGATA8 binding motif. **b**, An EMSA testing the binding strength of OsGATA8 to the predicted binding motifs in the *OsTCP19* promoter using probes as shown in **a**. The results are representative of three independent experiments. An asterisk represents a nonspecific signal. **c**, LUC assays in rice protoplasts on the effect of OsGATA8 on the transcription of *OsTCP19*. Values represent mean ± s.d. derived from three independent samples of rice protoplasts. **d**, ChIP–qPCR assay of the interaction between OsGATA8 and the promoters of *OsTCP19* in the shoot of *p35S::Flag-OsGATA8* transgenic plants at the four-leaf stage. Values represent mean ± s.d. derived from three independent samples. **e**, Phenotypes of Nip, the *OsGATA8*

knockout mutant, the *OsTCP19* knockout mutant and the *OsGATA8/OsTCP19* double-knockout mutant at the maximum tillering stage (about 40 days after transplanting) and the mature stage under LN and HN conditions. Scale bars, 20 cm. LN, 75 kg ha⁻¹ net nitrogen; HN, 300 kg ha⁻¹ net nitrogen. **f**, PTNR of the genotypes in **e** under LN and HN conditions. $n = 20$ plants. PTNR, productive tiller number under LN condition/productive tiller number under HN condition. **g**,**h**, The total number of tillers, productive tillers (**g**) and proportion of productive tillers (PT%; **h**) of the genotypes in **e** under LN and HN conditions. 'Δ' represents the percentage difference compared with the total number of tillers. $n = 20$ plants. In **c** and **d**, significant difference was determined by two-tailed Student's *t* test; in **f**–**h**, different letters indicate significant differences ($P < 0.05$, one-way ANOVA, Duncan's new multiple range test). For *P* values, see source data. Box plots denote the 25th percentile, the median and the 75th percentile, with minimum to maximum whiskers.

245-bp deletion in the backgrounds of Nip (Fig. 4a and Supplementary Fig. 13) and a current cultivar Ningjing 4 (N4; Supplementary Figs. 13 and 14a). In addition, we also obtained three other genome-edited mutant lines in the Nip background. These mutant lines carry 4-bp (line D4-Nip), 11-bp (line D11-Nip) and 20-bp (line D20-Nip) of deletions outside the 245-bp deletion region, respectively (Supplementary Fig. 13). We quantified the expression of OsGATA8 in the seedlings of these four lines and found that only the D403 line showed a significant decrease in OsGATA8 expression (Extended Data Fig. 6c and Supplementary Figs. 13 and 14b). Consistently, line D403 showed higher PTNR and proportion of productive tillers compared with the WT (Fig. 4b–d and Supplementary Fig. 14c–f). To verify that the reduced OsGATA8 expression in the D403 line is caused by the deletion of the 245-bp region, but not the flanking sequences, we performed LUC assays using segments of the OsGATA8 promoter. We found that the flanking sequences (115-bp upstream and 43-bp downstream) of the 245-bp deletion did not significantly affect the promoter activity of OsGATA8 (Extended Data Fig. 6a,b). These results suggest that the OsGATA8 allele with the 245-bp deletion in its promoter is a hypomorphic allele with reduced expression.

To identify the superior allele of OsGATA8 associated with higher NUE, we compared the averaged relative expression of OsGATA8 and PTNR in the HapL and HapH varieties. Transcriptome assays showed that the averaged relative expression of OsGATA8 in the HapH varieties was significantly lower than that in the HapL varieties under HN conditions (Supplementary Fig. 15a), while phenotypic data revealed that the PTNR of the HapH varieties was significantly higher than that of the HapL varieties (Supplementary Fig. 15b), suggesting that HapH is the elite haplotype conferring higher NUE. To further validate this notion, we generated OsGATA8 transgenic lines with the OsGATA8 coding sequence from Nip driven by a 1.5-kb upstream sequence derived from the two OsGATA8 haplotypes (pHapH or pHapL, three independent lines for each; Fig. 4e and Supplementary Fig. 15c). Consistent with the results from the haplotype analysis, the pHapH::OsGATA8 transgenic plants exhibited significantly higher PTNR and the proportion of productive tillers than the pHapL::OsGATA8 transgenic plants, under both LN and HN conditions (Fig. 4f–i). These results verify that OsGATA8-H is an elite haplotype with higher NUE.

To further investigate the function of OsGATA8-H in regulating rice tillering formation and nitrogen uptake, we generated a near-isogenic line (NIL) of the japonica variety 'Asominori' (Aso) with OsGATA8-H from the indica variety 'IR24' through marker-assisted selection (Fig. 5a and Supplementary Fig. 16a). We measured the total tiller numbers and productive tiller numbers in OsGATA8 transgenic plants and Aso^OsGATA8-H from seedling to maturing stages and found that the total tiller numbers were lowered in the Aso^OsGATA8-H plants at the maximum tillering stage under HN condition (Fig. 5a,b and Extended Data Figs. 5 and 7a). We quantified the expression of OsGATA8 in rice tiller buds and roots from seedling to maturing stage in the Aso and Aso^OsGATA8-H lines and found that the expression of OsGATA8 increased since the seedling

stage, peaked at the maximum tillering stage and decreased thereafter. Consistent with the lower total tiller numbers in the Aso^OsGATA8-H plants, the expressions of OsGATA8 were generally lowered in the Aso^OsGATA8-H plants and specifically repressed in the young tillers that can be transformed into the productive tillers (Extended Data Fig. 7b–d). We also found that the expression of OsAMT3.2 was repressed by OsGATA8 in roots under both LN and HN conditions (Supplementary Fig. 17a), and OsTCP19 was only repressed by OsGATA8 in tiller buds under HN conditions (Supplementary Fig. 17b,c). In addition, we found that the expression of OsAMT3.2 and OsTCP19 was dynamically regulated by OsGATA8 during rice growth and development (Supplementary Fig. 17a,b). Thus, OsGATA8-H increased proportion of productive tillers predominantly via upregulating OsAMT3.2 to uptake ammonium under LN, whereas downregulating OsTCP19 to reduce nonproductive tillers under HN where the availability of nitrogen is not a rate-limiting factor (Supplementary Fig. 17). We thus speculate that this temporal expression pattern of OsGATA8 is conducible for its dual role in promoting nitrogen supply to the developing tillers during vegetative growth while reducing nonproductive tillers at the onset of reproductive growth[26].

To further elucidate how OsGATA8 regulates NUE in rice during vegetative growth and reproductive growth, we quantified the biomass, nitrogen content and yield in the OsGATA8 NIL line (Aso^OsGATA8-H) at the seedling, maximum tillering and mature stages and calculated the NUE accordingly (Fig. 5c–e and Supplementary Fig. 18). We found that at the seedling and maximum tillering stages, Aso^OsGATA8-H plants displayed increased biomass, nitrogen content and NUE under LN condition, but no differences were observed in biomass and NUE compared with Aso under HN condition (Supplementary Fig. 18a,b). At the mature stage, Aso^OsGATA8-H plants exhibited increased yield, biomass nitrogen content and NUE under both LN and HN conditions (Fig. 5c–e and Supplementary Fig. 18c). These results suggest that OsGATA8 has a dual role in regulating NUE. On one hand, OsGATA8 downregulates nitrogen uptake by repressing the expression of OsAMT3.2 (Fig. 2). On the other hand, OsGATA8 promotes tiller formation by downregulating the expression of OsTCP19, which encodes an inhibitor of tillering in rice (Fig. 3). As a consequence of N-induced expression of OsGATA8, tillering is promoted while nitrogen uptake is downregulated. This leads to nonproductive tillers and a decrease in yield and NUE[27].

## OsGATA8-H has been artificially selected in fertile regions

To examine whether OsGATA8 has been under artificial selection during the domestication of rice, we calculated the nucleotide diversity within a 30-kb region covering OsGATA8 in the rice 3K population[28] and found a selective sweep signal between Oryza rufipogon and indica, indicating that OsGATA8-H was artificially selected in rice during the domestication process of rice (Supplementary Fig. 19a). Next, we analyzed the selection process of OsGATA8-H over the breeding history of Asian cultivated rice. We first developed a 245-bp InDel marker in the OsGATA8 promoter and genotyped OsGATA8 in 146 wild rice varieties

**Fig. 4 | Natural variation in the OsGATA8 promoter affects NUE in rice. a**, The phenotypes of Nip and the D403-Nip line at the maximum tillering stage and mature stage compared with Nip under LN and HN conditions. LN, 75 kg ha$^{-1}$ net nitrogen; HN, 300 kg ha$^{-1}$ net nitrogen. Scale bars, 20 cm. **b**, The PTNR of Nip and the D403-Nip line at the maximum tillering stage and mature stage under LN and HN conditions. $n$ = 20 plants. **c**, The number of total number of tillers and productive tillers of Nip and the D403-Nip line under LN and HN conditions. 'Δ' represents the percentage difference compared with the total number of tillers. $n$ = 20 plants. **d**, The proportion of productive tillers (PT%; left) and the number of nonproductive tillers (right) of Nip and the D403-Nip lines under LN and HN conditions. $n$ = 20 plants. **e**, The phenotypes of Nip and six OsGATA8 transgenic lines with the HapL and HapH promoters (pHapL::OsGATA8 and pHapH::OsGATA8) at the maximum tillering stage (about 40 days after

transplanting) and the mature stage under LN and HN conditions. Scale bars, 20 cm. **f**, PTNR of the genotypes in **e**. $n$ = 20 plants. PTNR, productive tiller number under LN condition/productive tiller number under HN condition. **g**, The number of total number of tillers and productive tillers in the genotypes in **e** under LN and HN conditions. 'Δ' represents the percentage difference compared with the total number of tillers. $n$ = 20 plants. **h,i**, The proportion of productive tillers (PT%; **h**) and nonproductive tillers (**i**) of the genotypes in **e** under LN and HN conditions. $n$ = 20 plants; values represent mean ± s.d. In **b** and **d**, P values were calculated with two-tailed Student's $t$ test; in **c** and **f–i**, different letters indicate significant differences ($P$ < 0.05, one-way ANOVA, Duncan's new multiple range test). For P values, see source data. In **b–d** and **f–i**, box plots denote the 25th percentile, the median and the 75th percentile, with minimum to maximum whiskers.

from different regions of Asia. We identified *OsGATA8*-H in most of the surveyed wild rice varieties distributed in southern China, Myanmar and Sri Lanka (Supplementary Fig. 19b and Supplementary Table 9). Haplotype analysis of *OsGATA8* in the 446 wild rice[29] varieties revealed that the

*OsGATA8*-H originated from two haplotypes in wild rice, Hap (*O.rufip*)-5 and Hap (*O.rufip*)-9 (Supplementary Fig. 19c). We also investigated the distribution of *OsGATA8*-H in cultivated rice varieties and found that *OsGATA8*-H exists in only 17% of the *indica* and tropical *japonica*

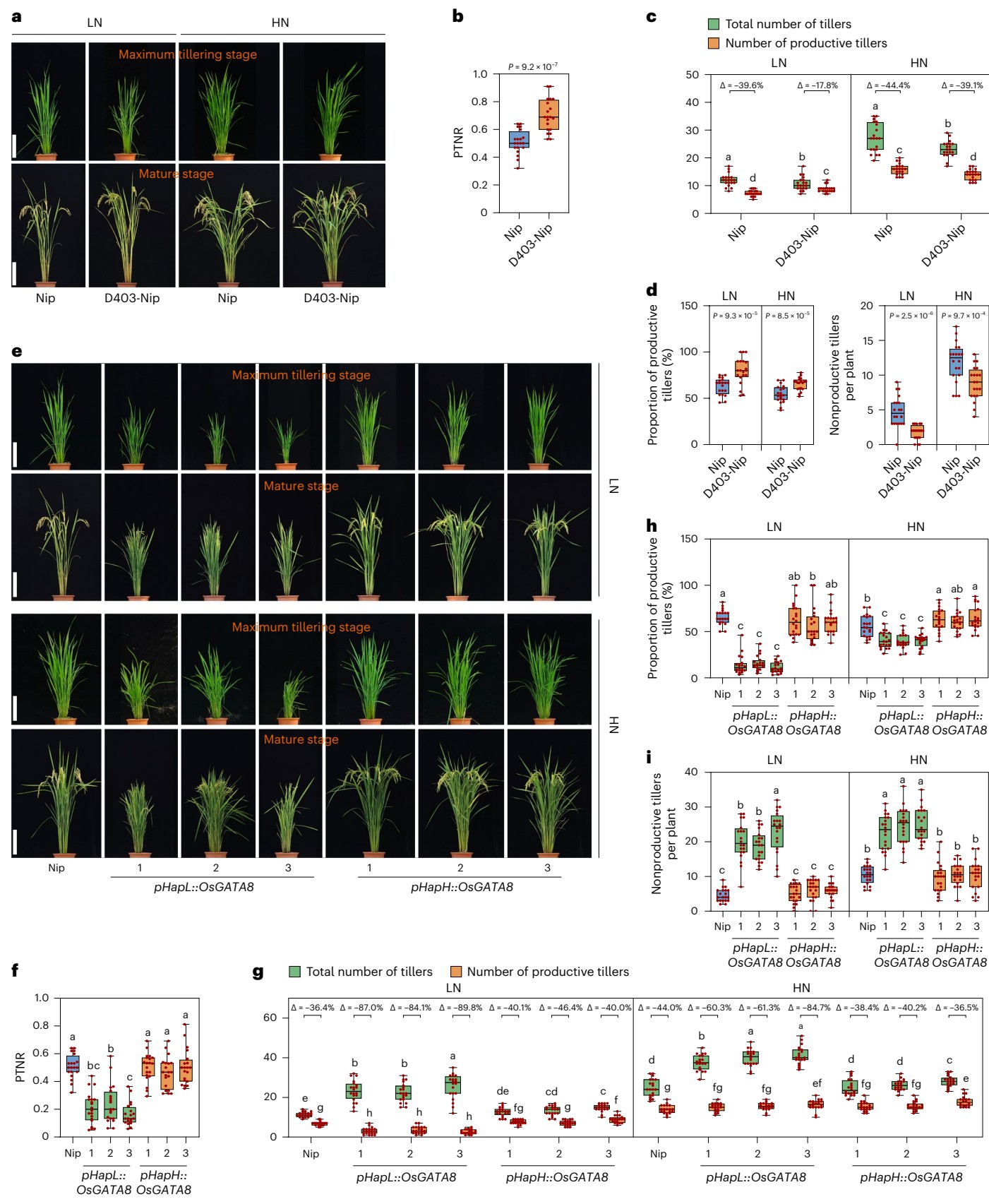

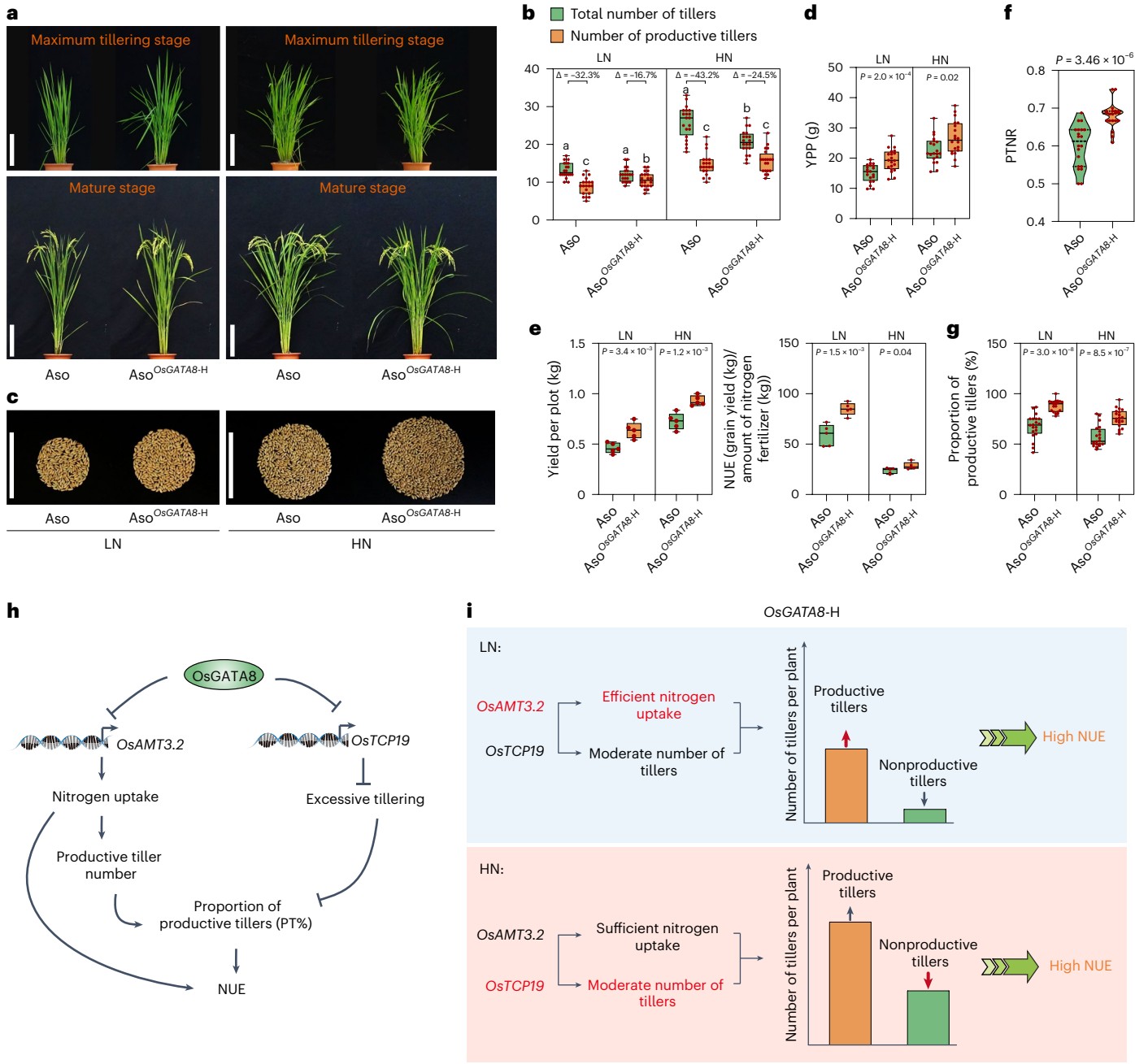

**Fig. 5 | An elite haplotype of rice confers higher NUE. a**, Phenotypes of Aso and Aso$^{OsGATA8\text{-}H}$ under LN and HN conditions at the maximum tillering and mature stages. LN, 75 kg ha$^{-1}$ net nitrogen; HN, 300 kg ha$^{-1}$ net nitrogen. Scale bars, 20 cm. **b**, The number of total number of tillers and productive tillers of the genotypes in **a** under LN and HN conditions. 'Δ' represents the percentage difference compared with the total number of tillers. $n = 20$ plants. **c**, Grains from one Aso plant and one Aso$^{OsGATA8\text{-}H}$ plant under LN and HN conditions. Scale bars, 10 cm. **d**, Comparison of the YPP between Aso and Aso$^{OsGATA8\text{-}H}$ under LN and HN conditions. $n = 20$ plants. **e**, The yield per plot (10 plants × 4 rows) and NUE of Aso and Aso$^{OsGATA8\text{-}H}$ under LN and HN conditions ($n = 5$ plots). NUE = grain yield (kg)/amount of nitrogen fertilizer (kg). **f**,**g**, PTNR (**f**) and proportion of productive tillers (PT%, **g**) of the genotypes in **a**. $n = 20$ plants. PTNR, productive tiller number under LN condition/productive tiller number under HN condition.

**h**, A proposed model of OsGATA8 regulating rice NUE by balancing nitrogen uptake and tillering growth in rice. **i**, A proposed model of the role of OsGATA8 in coordinating nitrogen uptake and tiller development. Under LN conditions, *OsGATA8*-H confers high proportion of productive tillers (PT%) predominantly via enhanced ammonium uptake through upregulated *OsAMT3.2* (highlighted in red). Under HN conditions, *OsGATA8*-H confers a high PT% predominantly via maintaining a moderate number of tillers through upregulated *OsTCP19* (highlighted in red). In **b**, different letters indicate significant differences ($P < 0.05$, one-way ANOVA, Duncan's new multiple range test); for $P$ values, see source data; in **d**–**g**, $P$ values were calculated with two-tailed Student's $t$ test. In **b**, **d**, **e** and **g**, box plots denote the 25th percentile, the median and the 75th percentile, with minimum to maximum whiskers; in **f**, the bars in the violin plots represent the 25th percentile, median and 75th percentile.

varieties in the rice 3 K population (Extended Data Fig. 8a), while it is completely absent in the *aus* varieties (Extended Data Fig. 8b and Supplementary Table 10). We further analyzed 135 *indica* rice varieties from the 1950s to the 2000s and found that the frequency of the presence of

*OsGATA8*-H in varieties before 1960 was relatively low, which is consistent with the frequency of this haplotype in the 146 wild rice varieties, but was significantly increased since that time (Extended Data Fig. 8c and Supplementary Table 11). This observation correlates with the

sharply increased popularization and large-scale use of industrially synthesized chemical fertilizers in agricultural production since the 1960s (Extended Data Fig. 8d and Supplementary Table 12). Additionally, we analyzed the frequencies of *OsGATA8* haplotypes in the rice 3 K population from countries with different amounts of soil nitrogen in Asia. We found that the frequency of *OsGATA8*-H positively correlates with the regional soil nitrogen content (Extended Data Fig. 8e and Supplementary Table 13). These observations together suggest that *OsGATA8*-H may have been under artificial selection for adaptation to fertile soil and HN conditions.

### Breeding potential of *OsGATA8*-H/*OsTCP19*-H

Excessive nitrogen fertilizer reduces the proportion of productive tillers and NUE in rice. To investigate whether *OsGATA8*-H has the breeding potential to improve the proportion of productive tiller and NUE in rice, we examined the proportion of productive tiller, PTNR and yield using the *OsGATA8*-H/L varieties under LN and HN conditions (Supplementary Figs. 20 and 21). Our results revealed that the increased application of nitrogen results in a dramatic increase in the formation of nonproductive tillers, and artificial selection toward *OsGATA8*-H led to decreased nonproductive tiller formation under HN. The Aso$^{OsGATA8-H}$ line not only exhibited higher PTNR (Fig. 5f), increased proportion of productive tillers (Fig. 5g), panicle length, grain number per panicle (Extended Data Fig. 9a,b) and grain YPP (Fig. 5c,d), but also showed increased yield per plot and NUE at the mature stages (Fig. 5e and Extended Data Fig. 9a). Consistently, we observed increased *OsAMT3.2* expression in the roots of the Aso$^{OsGATA8-H}$ line and higher nitrogen concentrations in the shoots of the Aso$^{OsGATA8-H}$ line, which is consistent with the expected effect of reduced *OsGATA8* expression (Supplementary Figs. 18 and 22). These results further demonstrate that *OsGATA8* negatively regulates yield and NUE.

Previous studies identified *OsTCP19* as a modulator of tillering in response to nitrogen and found that the elite haplotype *OsTCP19*-H (harboring a 29-bp InDel in its promoter) is prevalent in wild rice, but has been largely lost in modern cultivars. Moreover, excessive nitrogen leads to an increased number of nonproductive tillers in *OsTCP19*-H cultivars[11]. Here we showed that *OsGATA8*-H is associated with reduced tiller formation and efficient ammonium uptake. These observations prompted us to test the potential of achieving high NUE under both HN and LN conditions by combining the two elite haplotypes via cross-breeding. We constructed two NILs in the *indica* rice 9311 background, one with *OsGATA8*-H from wild rice (*O. rufipogon*) and the other carrying both *OsGATA8*-H and *OsTCP19*-H, an elite haplotype with high NUE[11] (Supplementary Fig. 16b,c and Extended Data Fig. 10a). Compared with the WT 9311, both the 9311$^{OsGATA8-H}$ and 9311$^{OsGATA8-H/OsTCP19-H}$ lines exhibit increased PTNR, productive tiller numbers and proportion of productive tillers (Extended Data Fig. 10b–d). An increase in YPP and NUE was also observed for 9311$^{OsGATA8-H}$ compared with 9311 (Extended Data Fig. 10e). Notably, 9311$^{OsGATA8-H/OsTCP19-H}$ exhibited even higher NUE and grain yield than 9311$^{OsGATA8-H}$ (Extended Data Fig. 10e). These results demonstrate that *OsGATA8*-H and *OsTCP19*-H represent a superior haplotype combination for high NUE and yield.

### Discussion

Excessive nitrogen input promotes the formation of nonproductive tillers, which fail to accumulate photo-assimilated products[30]. Yet how this process is regulated remains poorly understood. In this study, we identified a new transcription factor OsGATA8 as a coordinator of NUE and tiller formation. OsGATA8 negatively regulates nitrogen uptake by repressing transcription of the ammonium transporter gene *OsAMT3.2* and promotes tiller formation by repressing transcription of *OsTCP19*, a negative modulator of tillering. Thus, our results establish an intrinsic link between nitrogen uptake and the development of productive tillers.

Moreover, we identify *OsGATA8*-H as an elite haplotype with reduced expression, which confers enhanced nitrogen uptake, an increased proportion of productive tillers and higher NUE under both high and LN conditions (Fig. 5h,i). Under LN conditions, the relatively higher expression of the ammonium transporter gene *OsAMT3.2* leads to increased ammonium uptake, allowing an increased supply of nitrogen to rice tillers to promote their development into effective panicles (Figs. 2 and 5h). Meanwhile, the relatively higher expression of *OsTCP19* prevents excessive tiller formation under HN conditions (Figs. 3 and 5h). Therefore, *OsGATA8*-H may promote NUE and yield in rice under a broad range of nitrogen conditions given its dual role in the transcriptional regulation of *OsAMT3.2* and *OsTCP19* (Fig. 5i).

Previous studies showed that *OsTCP19* has a role in geographical adaptation to fertile soil in rice and that its elite haplotype, *OsTCP19*-H, is found mainly in *aus* rice varieties from regions with LN[11]. Here we found that *OsGATA8*-H is mainly present in *indica* and tropical *japonica* rice cultivars but completely absent in the *aus* varieties and is associated with high soil nitrogen content (Extended Data Fig. 8b,e). NUE and yield can be improved under both HN and LN conditions by combining *OsGATA8*-H and *OsTCP19*-H (Extended Data Fig. 10e). As varieties harboring both *OsGATA8*-H and *OsTCP19*-H are extremely rare, accounting for only ~7% of the varieties in the rice 3K population (Extended Data Fig. 10f), creation of *OsGATA8* promoter alleles that functionally resemble *OsGATA8*-H via genome editing[31,32] in rice cultivars carrying *OsTCP19*-H offers an expedite way of generating rice germplasm with optimized NUE in diverse rice genetic backgrounds.

### Online content

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

¹State Key Laboratory of Crop Genetics & Germplasm Enhancement and Utilization, Nanjing Agricultural University, Zhongshan Biological Breeding Laboratory, Nanjing, China. ²State Key Laboratory of Crop Gene Resources and Breeding, Institute of Crop Sciences, Chinese Academy of Agricultural Sciences, Beijing, China. ³Jiangsu Collaborative Innovation Center for Modern Crop Production, Southern Japonica Rice R&D Corporation Ltd, Nanjing, China. ⁴College of Animal Science and Technology, Nanjing Agricultural University, Nanjing, China. ⁵MOA Key Laboratory of Plant Nutrition and Fertilization in Lower-Middle Reaches of the Yangtze River, Nanjing Agricultural University, Nanjing, China. ⁶Department of Biology, Wilkes University, Wilkes-Barre, PA, USA. ⁷Department of Plant Pathology and the Genome Center, University of California, Davis, Davis, CA, USA. ⁸Joint BioEnergy Institute, Lawrence Berkeley National Laboratory, Berkeley, CA, USA. ⁹These authors contributed equally: Wei Wu, Xiaoou Dong. ✉e-mail: wangchm@njau.edu.cn; wanjianmin@caas.cn

## Methods

### Plant materials and growth conditions

The seeds of the 117 accessions were collected, stored and supplied by the State Key Laboratory of Crop Genetics & Germplasm Enhancement and Utilization, Jiangsu Collaborative Innovation Center for Modern Crop Production, Nanjing Agricultural University, China. Germination, transplantation and cultivation of the 117 cultivars were performed concurrently in the same fields (HN or LN). All 117 rice cultivars are capable of developing productive tillers with normal grains for harvest in Nanjing, China (31°139′ N, 119°22′ E, 30 m above sea level). All materials were planted in the field at the experimental farm of the Nanjing Agricultural University, Nanjing, China (31°139′ N, 119°22′ E, 30 m above sea level). For the field experiments, the accessions were grown in a completely randomized block design with four replicates. The field experiments were carried out as a randomized block design with two nitrogen levels (300 kg ha⁻¹ net nitrogen and 75 kg ha⁻¹ net nitrogen) in two blocks. All the SNP data and phenotype data are shown previously[14]. Phosphate and potassium fertilizers were both applied at 100 kg ha⁻¹. There were 20 cm and 17 cm between rows and individuals, respectively. Rice seedlings were cultured in International Rice Research Institute nutrient solution[33]. All transgenic materials were obtained through *Agrobacterium tumefaciens*-mediated transformation as previously described[34]. Daytime conditions in the greenhouse were 30 °C for 14 h; nighttime conditions were 28 °C and dark for 10 h.

### GWAS

We investigated plant height and productive tiller number at the mature stage under LN and HN conditions. We calculated the ratio of plant height and productive tiller number under LN/HN (PHR and PTNR) and used them as proxies of NUE to carry out GWAS. We carried out GWAS and prioritization of the candidate genes as described in our previously published papers[14,17] with minor modifications.

### Rice genome editing by CRISPR–Cas9

Single-guide RNAs were designed with CRISPR-P (v2.0)[35] (http://crispr.hzau.edu.cn/CRISPR2/). Constructs for the genome editing of rice plants were generated using a CRISPR plasmid toolbox as described before[36]. For single-target edits, a pair of oligonucleotides bearing the spacer sequence were annealed and ligated into a *Bsa*I-digested binary vector backbone carrying the rice codon-optimized *SpCas9* driven by the maize ubiquitin promoter and the single-guide RNA scaffold driven by the *OsU6* promoter. For two-target edits, a pair of oligonucleotides bearing one of the two spacer sequences were annealed and ligated into *Bsm*BI-digested intermediate vector backbones pYPQ131c or pYPQ132c, respectively. A Goldengate reaction with *Bsa*I was performed on the completed pYPQ131c-sgRNA1, pYPQ132c-sgRNA2 and pYPQ142 to generate an intermediate plasmid bearing two guide RNA-encoding genes. Finally, a multiway LR reaction was performed with the completed double-guide RNA plasmid pYPQ142-sgRNA1 + 2, the Cas9 plasmid pYPQ167 and the binary vector backbone pCam1300 to generate the final construct using the LR Clonase II Plus Kit (Invitrogen). *Agrobacterium*-based rice transformation was performed as described above to obtain individual transformation events. Homozygous, *Cas9*-free mutants were obtained through genetic segregation and genotyping by PCR–Sanger sequencing.

### Observation of material phenotype

The number of tillers in rice was surveyed at 7-day intervals, starting 14 days after transplanting of rice seedlings. The total tiller number was scored at the maximum tillering stage of rice (about 40 days after transplanting), and the productive tiller number was scored at the mature stage of rice. The proportion of productive tillers is calculated by dividing the number of productive tillers at the mature stage by the number of total number of tillers at the maximum tillering stage. The number of nonproductive tillers is calculated by subtracting the number of productive tillers at the mature stage from the total number of tillers at the maximum tillering stage (Extended Data Fig. 1a). The productive tillers referred to panicles with more than five full grains. Normally, the nitrogen uptake and usage during the grain-filling stage of rice determine the formation of rice grains and productive tillers. PTNR was evaluated by calculating the relative PTNR in plants grown under HN and LN conditions.

### Real-time PCR and RNA-seq

Total RNA was isolated from tissues of rice seedlings, leaves, tiller buds or roots using a plant RNA purification kit (Invitrogen). Real-time PCR was conducted with I-Cycle (Bio-Rad). The reaction system contained 200 ng complementary DNAs, 0.5 µl of 10 mmol l⁻¹ gene-specific primers and 20 µl of real-time PCR SYBR Mix (2X SYBR Green Pro Taq HS Premix II; Accurate Biotechnology, AG11702). *OsACTIN1* was used as the internal control. All of the quantitative real-time primers and primers involved in this paper are listed in Supplementary Table 14. At least three independent biological replicates were collected for each experiment. Seedlings of Nip and *OsGATA8*-cr lines were grown for 2 weeks in a basic nutrient solution containing 1.44 mM nitrogen. The seedlings of 175 rice varieties from the rice 3K population were grown in a basic nutrient solution containing 1.44-mM nitrogen for 2 weeks and treated with nitrogen-free nutrient solution for 1 h. Seedling tissues weighing 2 g were collected for subsequent RNA-seq. Sequencing and data analysis were conducted by Shenzhen BGI using Illumina HiSeq 2000 Plus.

### Transient transactivation assay in rice protoplasts

Rice protoplasts were prepared from 2-week-old seedlings as previously described[37]. The vector pGreenII 0800-LUC was used to analyze the activity of the different promoters. The 1500-bp upstream start codons of *OsAMT1.2* and *OsAMT3.2* were cloned into vector pGreen II0800-LUC to generate reporters, and the full-length coding sequence of *OsGATA8* was inserted into vector pAN580 to generate the effector. Empty vectors of pGreen II0800-LUC and Pan580 were used as controls. After 16 h of penetration at 28 °C in the dark, the protoplast protein was extracted, and firefly LUC activity and *Renilla* (REN) LUC activity were measured using the dual-LUC reporter assay system (Promega, E1910). The ratio between LUC and REN activities was measured three times.

### Phylogenetic analysis

The amino acid sequences of ammonium transporters (AMTs) in rice and their homologs in *Arabidopsis thaliana* were aligned by MEGA7 software. Phylogenetic trees were constructed with the aligned protein sequences using MEGA7 software with the neighbor-joining method. The accession numbers and databases of sequences for constructing these phylogenetic trees can be found in the Michigan State University (MSU) Rice Genome Annotation Project (http://rice.plantbiology.msu.edu/) and the National Center for Biotechnology Information database (https://www.ncbi.nlm.nih.gov/).

### ¹⁵N accumulation assay

Rice seedlings were grown in International Rice Research Institute nutrient solution (1.44 mM NH₄NO₃) for 3 weeks and were changed once a day. Uniform seedlings were chosen for further treatments. Then, the seedlings were pretreated with 2 mM (NH₄)₂SO₄ and 2 mM KNO₃ for 1 week and transferred to a nitrogen-free solution for starvation treatment for 4 days, then transferred to 0.1 mM CaSO₄ for 1 min and treated with 0.2 mM and 2 mM (¹⁵NH₄)₂SO₄ and Ca(¹⁵NO₃)₂ for 10 min, respectively. Finally, the roots of the seedlings were collected after being washed with 0.1 mM CaSO₄ solution and deionized water. The samples were dried at 70 °C for 7 days and then detected the ¹⁵N content using an isotope ratio mass spectrometer system (model Flash 2000 HT; DELTAV Advantage; Thermo Fisher Scientific).

## Nitrogen content determination and NUE calculation

After drying, the aboveground parts of the plants were ground into a uniform powder at the maximum tillering and mature stages. One gram of homogeneous powder was weighed, and the nitrogen content of the plant was determined using the micro Kjeldahl method. NUpE was calculated by dividing the total nitrogen in a shoot by the amount of nitrogen fertilizer. NUtE was calculated by dividing the dry shoot biomass or grain yield by the total nitrogen in the shoot (NUE = NUpE × NUtE[3]).

## EMSA

The full-length coding sequence of *OsGATA8* was inserted into vector pMAL-c4x and transformed into competent *Escherichia coli* BL21 (DE3) cells. *E. coli* DE3 containing pMAL-*OsGATA8* was added to one-thousandth of isopropyl-β-ᴅ-thioglycolopy-ranoside at a concentration of 0.1 M and incubated at 16 °C for 20 h. Purification of recombinant protein using maltose magnetic beads. The LightShiftTM Chemiluminescent EMSA Kit (Thermo Fisher Scientific) was used for performing the EMSA. All primers used for probes and competitors are listed in Supplementary Table 14. Detailed experimental steps of EMSA were described previously[12].

## ChIP assays

ChIP assays were performed as described previously[37]. In brief, 4 g of transgenic rice seedlings of *p35S::Flag-OsGATA8* were collected and cross-linked with 1% formaldehyde under a vacuum for 10 min. Glycine was added to the sample to a final concentration of 125 mM for quenching cross-linking, and the samples were ground into powder in liquid nitrogen. Chromatin was separated and ultrasonically fragmented. Anti-Flag antibodies were used for immunoprecipitation. DNA was purified and used for real-time PCR. The relevant primers are shown in Supplementary Table 14.

## Subcellular localization

The full-length coding sequences of *OsGATA8* and *OsTCP19* were inserted into the vector pAN580 (primers listed in Supplementary Table 14) to generate the *OsGATA8-GFP* and *OsTCP19-mCherry* constructs, respectively. The *OsGATA8-GFP* and *OsTCP19-mCherry* plasmids were cotransformed into rice protoplasts, and fluorescence signals were detected using a laser confocal scanning microscope (Leica TCS SP8) at 16 h after transformation.

## Fluorescence in situ hybridization

RNA probes with fluorescence-labeled *OsGATA8* and *OsTCP19* were synthesized (primers listed in Supplementary Table 14). Rice seedlings grown for 10 days were selected, and the shoot base was sampled for longitudinal sectioning to obtain rice tiller bud slices. The experiment was performed at Shanghai Rochenpharma Biotechnology. Experimental operation was performed as described in http://www.rochenpharma.com/.

## DAP-seq

The full-length coding sequences of *OsGATA8* were inserted into the vector Halo (provided by Genedenovo Biotechnology) to obtain the expression vector *OsGATA8*-Halo. DNA was extracted from the leaves of rice seedlings grown for 1 week and used to construct a cDNA library. The experimental process and data analysis were performed at Guangzhou Genedenovo Biotechnology.

## Nucleotide diversity estimation

Sequence data of *OsGATA8* and the 30-kb flanking regions that cover *OsGATA8* were obtained from the 3,000 Rice Genomes Project of 2,832 varieties and the rice HapMap3 of 376 *O. rufipogon* accessions. The primers used to verify the distribution of *OsGATA8*-H in wild rice are shown in Supplementary Table 14. The DNA of 146 wild rice and sequence data of 135 *indica* cultivars were provided by the Chinese Academy of Agricultural Sciences. To construct the haplotype network of *OsGATA8*, we also obtained the SNPs from the 3,000 Rice Genomes Project of 2,832 varieties and the rice HapMap3 of 376 *O. rufipogon* accessions.

## Statistics and reproducibility

Numbers (*n*) of samples or replicates are indicated in the figure legends (Figs. 1–5, Extended Data Figs. 1–10, and Supplementary Figs. 3, 5–8, 11, 14, 15, 17, 18, 21 and 22) and Methods. For bar charts, all values are presented as mean ± s.d. For box plots, box plots denote the 25th percentile, the median and the 75th percentile, with minimum to maximum whiskers. For violin plots, the bars represent the 25th percentile, median and 75th percentile. For pairwise comparisons, significance analysis was calculated by two-tailed Student's *t* test using Excel 2010, and the exact *P* values are displayed. For multiple-group comparisons, significance analysis was calculated by one-way analysis of variance (ANOVA; *P* < 0.05) followed by Duncan's new multiple range test as indicated in figure legends (Figs. 1b–f, 2b,h,i, 3f–h, 4c,f–i and 5b; Extended Data Figs. 4a,c,e,f, 6b,c, 9b and 10b–e; and Supplementary Figs. 3f–h, 5a, 7b–h, 8, 11c–g, 14d, 15c, 17c and 22) using SPSS version 18.0 (IBMA) and indicated with different letters.

## Reporting summary

Further information on research design is available in the Nature Portfolio Reporting Summary linked to this article.

## Data availability

Data supporting the findings of this work are available within the paper and its Supplementary Information. A reporting summary for this article is available as Supplementary Information. All datasets have been deposited to public databases. All genetic materials in the current study are available from the corresponding author. Sequence data of rice varieties can be found in our previous study[10] (https://doi.org/10.1038/s41467-019-13187-1). Sequence data from this study can be found in the MSU database (http://rice.plantbiology.msu.edu/) under the following accessions: OsGATA8 (LOC_Os01g24070), OsTCP19 (LOC_Os06g12230), OsAMT3.2 (LOC_Os03g62200) and OsAMT1.2 (LOC_Os02g40730). Sequence for constructing the phylogenetic tree of AMTs (Supplementary Fig. 6) can be found in the *Arabidopsis* Information Resource database (https://www.arabidopsis.org/) or the MSU database (http://rice.plantbiology.msu.edu/). The DAP-seq and RNA-seq data have been uploaded to the website National Center for Biotechnology Information Sequence Read Archive database (DAP-seq data: SRR28790977, SRR28790978; RNA-seq data: SRR28799732, SRR28799733, SRR28799734). Source data are provided with this paper.

## Code availability

All software used in the study are publicly available on the Internet as described in Methods and Reporting summary.

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

## Acknowledgements

The National Key Research and Development Project (2021YFF1000400 and 2022YFD1201702 awarded to Chunming Wang), National Natural Science Foundation of China (32272030 awarded to Chunming Wang), Scientific Innovation 2030 Project (2022ZD0401703 awarded to Chunming Wang), the Natural Science Foundation of Jiangsu Province (BK20210384 awarded to X.D. and BK20230982 awarded to G.C.) and China Postdoctoral Science Foundation (grant 2023M741753 awarded to G.C.) supported this study. We thank C. Zhu (Department of Immunology, The University of Texas Southwestern Medical Center, USA) for technical support in statistical analysis. The funding agencies had no role in the study design, data collection and analysis, decision to publish or paper preparation.

## Author contributions

J.W. and Chunming Wang directed the project. Chunming Wang, X.D., W.W. and W.X. designed the experiments. W.W. and G.C. performed most of the experiments. Chunming Wang, X.D., W.X., P.C.R. and H.W. wrote the paper and finalized the paper. Z.L., W.C., W. Tang, J.Y., S.W., X.Z., X.J., X.L., Y.W., Chunyuan Wang, X.C. and W.Z. participated in the experiments. H.W. and W. Terzaghi revised the paper. All the co-authors approved the paper.

## Competing interests

The authors declare no competing interests.

## Additional information

**Extended data** is available for this paper at https://doi.org/10.1038/s41588-024-01795-7.

**Correspondence and requests for materials** should be addressed to Chunming Wang or Jianmin Wan.

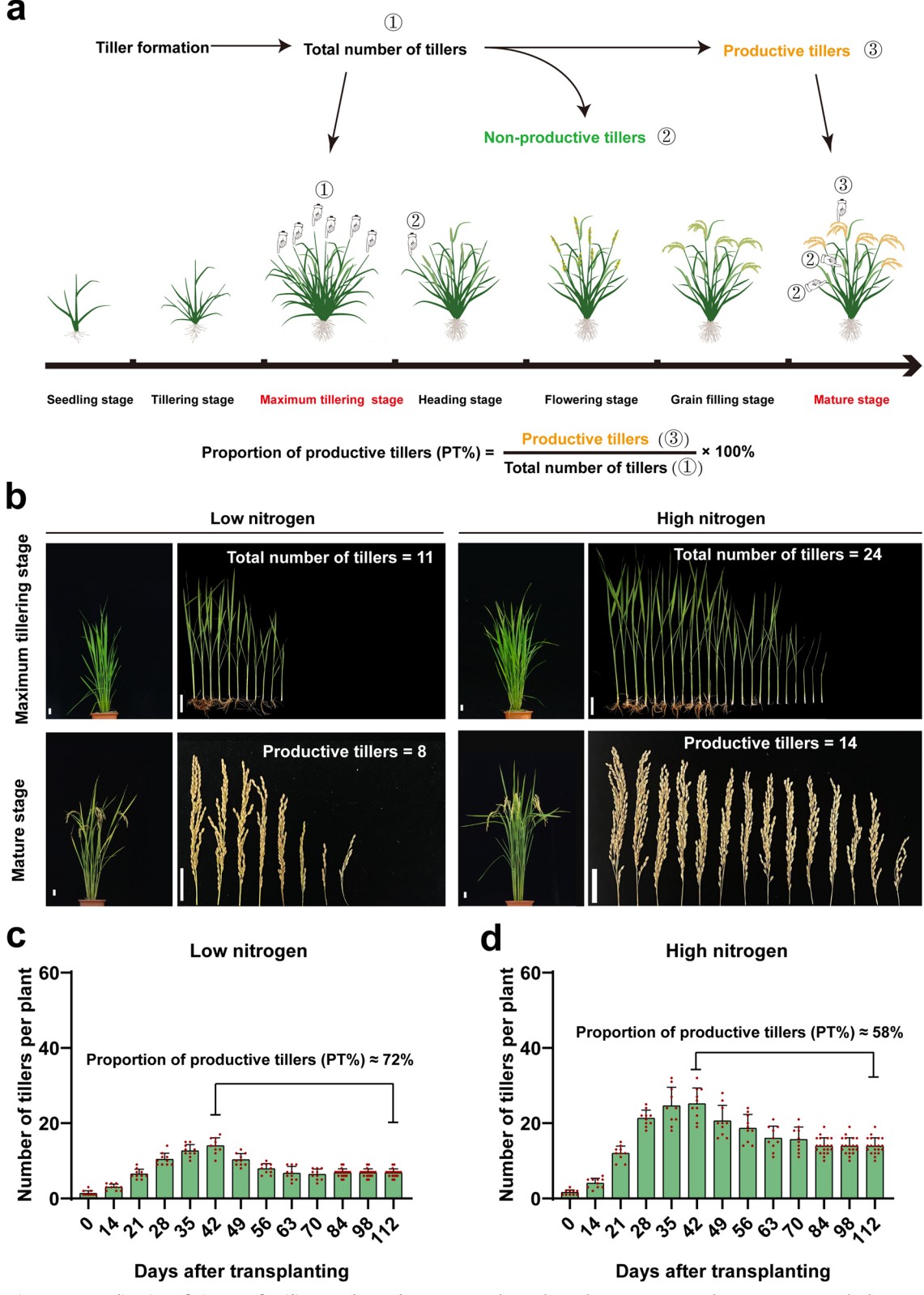

**Extended Data Fig. 1 | Overapplication of nitrogen fertilizers reduces the proportion of productive tillers in rice.** (a) An illustration of tiller development in rice. Tillers that produce effective panicles become productive tillers, while the rest become nonproductive tillers. The proportion of productive tillers (PT%) is defined as the ratio between the number of productive tillers and the total number of tillers at the maximum tillering stage. (b) The development of tillers and panicles in the *japonica* rice cultivar Nip grown under low nitrogen (75 kg/ha net nitrogen) and high nitrogen (300 kg/ha net nitrogen) conditions. (c,d) Tillers counts of Nipponbare rice grown under low nitrogen (c; 75 kg/ha net nitrogen) and high nitrogen (d; 300 kg/ha net nitrogen) conditions at various time points after transplanting. *n* = 10 plants. PT% is displayed.

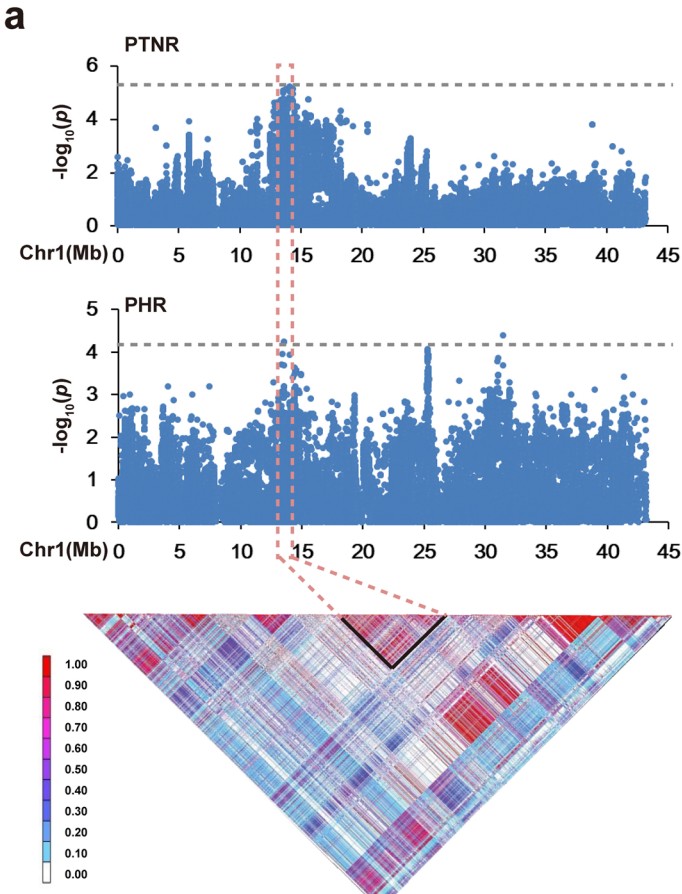

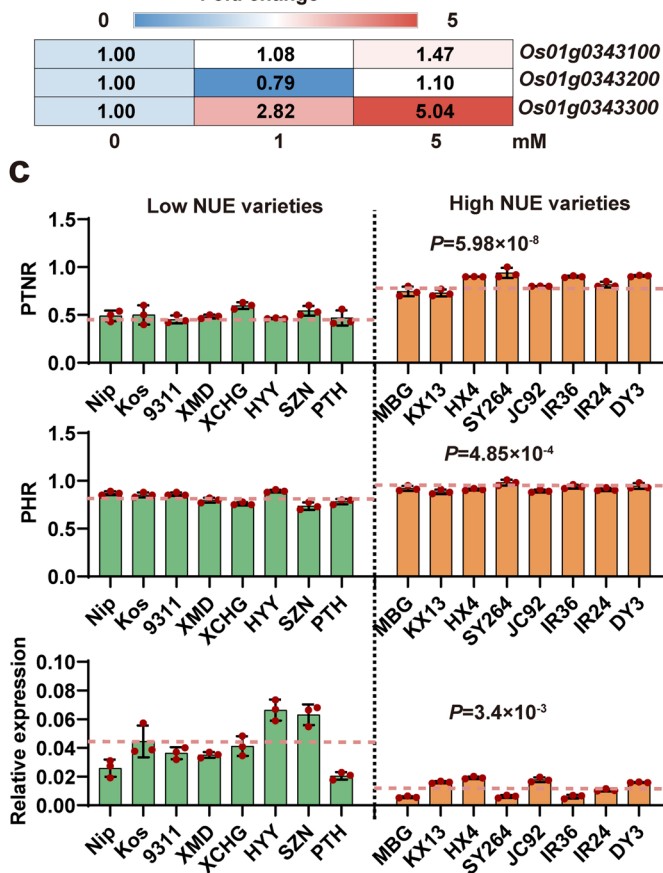

**Extended Data Fig. 2 | Identification of *OsGATA8* on chromosome 1. (a)** Local Manhattan plot (top) and linkage disequilibrium (LD) heatmap (bottom) surrounding the peak on chromosome 1 using *P* values of PTNR and PHR. The LD heatmap displays SNPs with $P < 1.0 \times 10^{-4}$. *P* values were determined under the mixed linear model and implemented in Tassel 5. PTNR, productive tiller number ratio (productive tiller number under LN condition/productive tiller number under HN condition); PHR, plant height ratio (plant height under LN condition/ plant height under HN condition). HN, high nitrogen (300 kg/ha net nitrogen); LN, low nitrogen (75 kg/ha net nitrogen). The color key (white to red) represents linkage disequilibrium values ($r^2$). (**b**) Heatmap of the relative expression of the three candidate genes near the peak in **a** under three nitrogen supply conditions (0-, 1- and 5-mM $NH_4NO_3$) based on qRT–PCR of rice seedlings. The color key (blue to red) represents fold changes of gene expression. (**c**) The PTNR, PHR and the relative expression of *OsGATA8* in 16 varieties with extreme NUE. Values represent mean ± SD derived from three individual plants. The dotted lines represent the average values of all varieties with extreme NUE, respectively. In **c**, *P* values were calculated with two-tailed Student's *t* test. PTNR, productive tiller number ratio (productive tiller number under LN condition/productive tiller number under HN condition); PHR, plant height ratio (plant height under LN condition/plant height under HN condition).

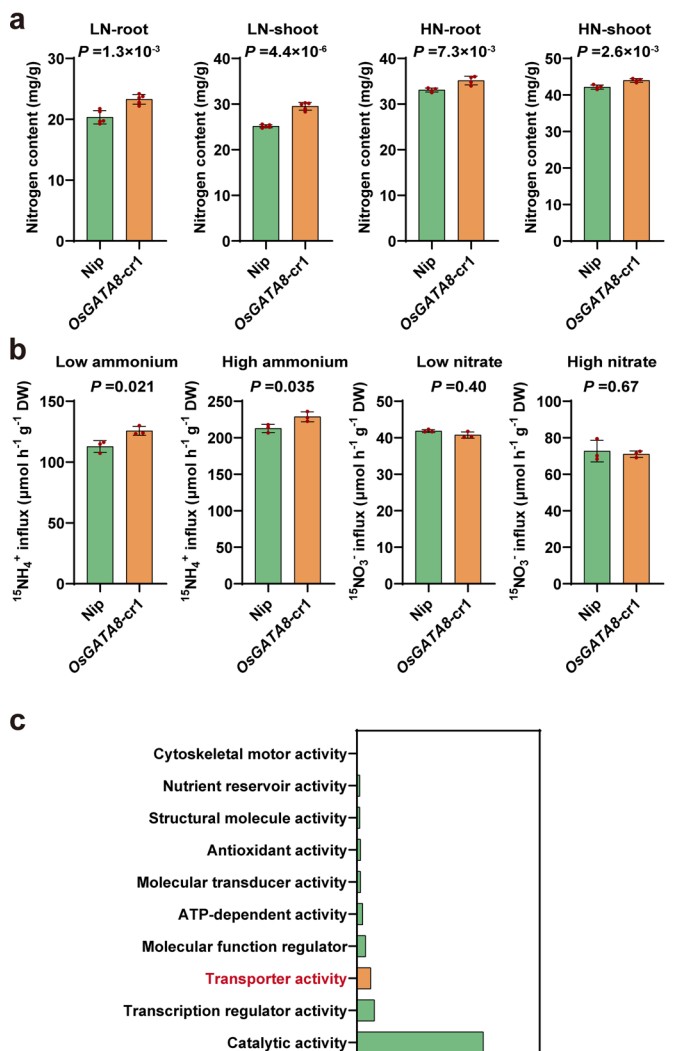

**d**

| | | | | | |
|---|---|---|---|---|---|
| polyol transporter 5 (PLT5) | | | | | |
| Inorganic H%2B pyrophosphatase (AVP1) | | | | | |
| Amino acid transporter (OsATL15) | | | | | |
| Similar to Amino acid carrier (OsAAP11E) | | | | | |
| low-affinity nitrate transporter (OsNPF7.2) | | | | | |
| Amino acid transporter (OsATL3) | | | | | |
| Similar to Cyclic nucleotide-gated channel A (OsCNGC6) | | | | | |
| **Ammonium transporter (OsAMT3.2)** | | | | | |
| Potassium transporter (OsHAK1) | | | | | |
| auxin-induced protein 5NG4 (OsUMAMIT7) | | | | | |
| Monosaccharide transporter (HEX6) | | | | | |
| Ferroportin1 family protein (OsFPN10) | | | | | |
| Potassium transporter (OsHAK17) | | | | | |
| Cation/H+ exchanger domain containing protein (OsCHX10) | | | | | |
| Plant iron carrier efflux transporter protein (OsTOM2) | | | | | |
| PIN1-like auxin transport protein (PIN1B) | | | | | |
| Amino acid transporter (OsLAX5) | | | | | |
| ABC transporter (OsABCG48) | | | | | |
| Cation/H+ exchanger domain containing protein (OsCHX09) | | | | | |

WT  CR#1  CR#3

Fold change (color key: blue to red, scale −5 to 5)

**e**

| Fold_enrichment | Annotation | GeneId | DistToTSS | Symbol |
|---|---|---|---|---|
| 5.05725 | Promoter (12kb) | Os02g0770800 | -1692 | *OsNR2* |
| 4.2946 | Promoter (<=1kb) | Os03g0375966 | -766 | *OsLAT4* |
| 4.10697 | Promoter (12kb) | *Os03g0838400* | -1465 | *OsAMT3.2* |
| 4.02597 | Promoter (<=1kb) | Os02g0753000 | -976 | *OsTPP4* |
| 3.36603 | Promoter (<=1kb) | Os03g0375300 | -996 | *OsLAT3* |
| 3.13744 | Promoter (<=1kb) | Os06g0228600 | -159 | *OsAAP12C* |
| 3.04943 | Promoter (<=1kb) | *Os02g0620600* | -691 | *OsAMT1.2* |
| 2.33766 | Promoter (<=1kb) | Os02g0102200 | -128 | *OsAAP7C* |
| 2.63849 | Promoter (12kb) | Os03g0609500 | -1777 | *OsLBD38* |
| 4.09397 | Promoter (12kb) | Os08g0155400 | -1048 | *OsNRT1.1A* |

**Extended Data Fig. 3 | OsGATA8 regulates nitrogen uptake in rice through the direct transcriptional regulation of nitrogen metabolism-related genes.** (**a**) The nitrogen content of Nip and *OsGATA8*-cr1 plants in roots and shoots grown under low nitrogen (LN, 0.2 mM $NH_4NO_3$) and high nitrogen (HN, 2 mM $NH_4NO_3$) concentrations for four weeks. Values represent mean ± SD derived from five individual plants. (**b**) $^{15}N$ accumulation assay in roots of the Nip and *OsGATA8*-cr1 lines under low ammonium (0.1 mM ($^{15}NH_4)_2SO_4$), high ammonium (1 mM ($^{15}NH_4)_2SO_4$), low nitrate (0.1 mM Ca($^{15}NO_3)_2$) and high nitrate (1 mM Ca($^{15}NO_3)_2$) conditions. Values represent mean ± SD derived from root tissues of three individual seedlings. (**c**) Gene Ontology enrichment analysis

of 619 DEGs (differentially expressed genes) in *OsGATA8* knockout lines under Nip background. (**d**) The relative expression of DEGs (differentially expressed genes) in transporter activity process (GO:0005215). The color key (blue to red) represents gene expression (FPKM) as fold changes. The proteins encoded are described on the left. (**e**) A list of nitrogen metabolism genes that OsGATA8 binds to according to DNA affinity purification sequencing (DAP-seq). Fold enrichment represents the number of reads in the purified sample relative to the input sample. In **a** and **b**, displayed *P* values were calculated with two-tailed Student's *t* test.

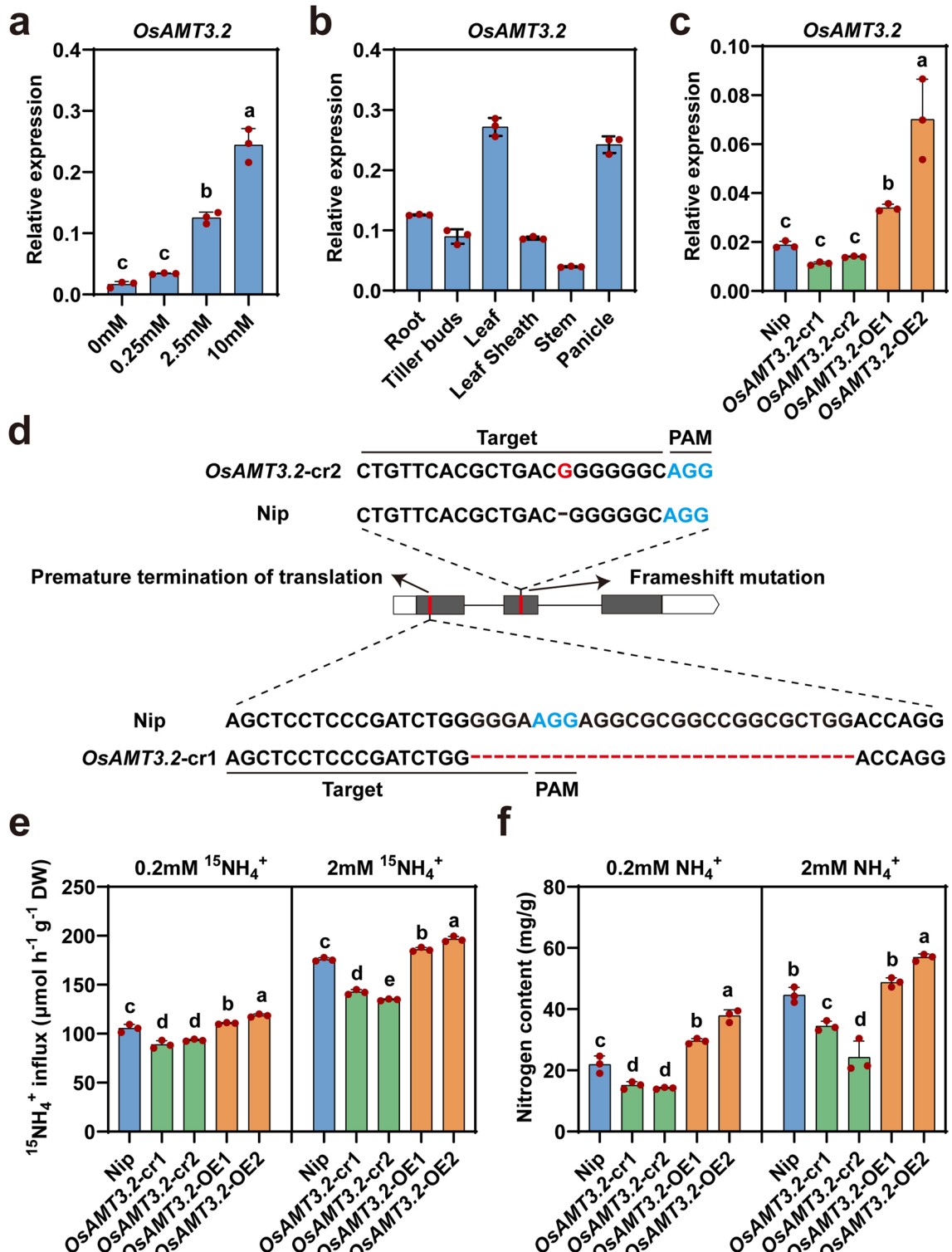

**Extended Data Fig. 4 | OsAMT3.2 positively regulates rice ammonium uptake.**
(**a**) Relative expression of *OsAMT3.2* under different nitrogen conditions (0-,
0.25-, 2.5- and 10-mM $NH_4^+$). Values represent mean ± SD derived from root
tissues from three individual rice seedlings. (**b**) Relative expression of *OsAMT3.2*
in various rice tissues after rice flowering. Values represent mean ± SD derived
from three individual plants. (**c**) Relative expression of *OsAMT3.2* in Nipponbare
(Nip), *OsAMT3.2* knockout and overexpression lines. Values represent mean ± SD
derived from root tissues from three individual rice seedlings. (**d**) The mutation
in the *OsAMT3.2* CRISPR knockout line in the Nipponbare background. Black bars:

the coding region; white bars: the UTRs; red lines: locations of the editing targets.
(**e,f**) $^{15}$N accumulation of ammonium in roots and total nitrogen content of shoots
in *OsAMT3.2* knockout (cr) and *OsAMT3.2* overexpression (OE) lines under low
ammonium (0.1 mM ($^{15}NH_4$)$_2$SO$_4$) and high ammonium (1.0 mM ($^{15}NH_4$)$_2$SO$_4$)
conditions. Values represent mean ± SD derived from three individual rice
seedlings. In **a** and **c**, total RNA was extracted from root tissue of two-week-
old seedlings. In **a**, **c**, **e** and **f**, different letters indicate significant differences
($P < 0.05$, one-way ANOVA, Duncan's new multiple range test), for $P$ values, see
source data.

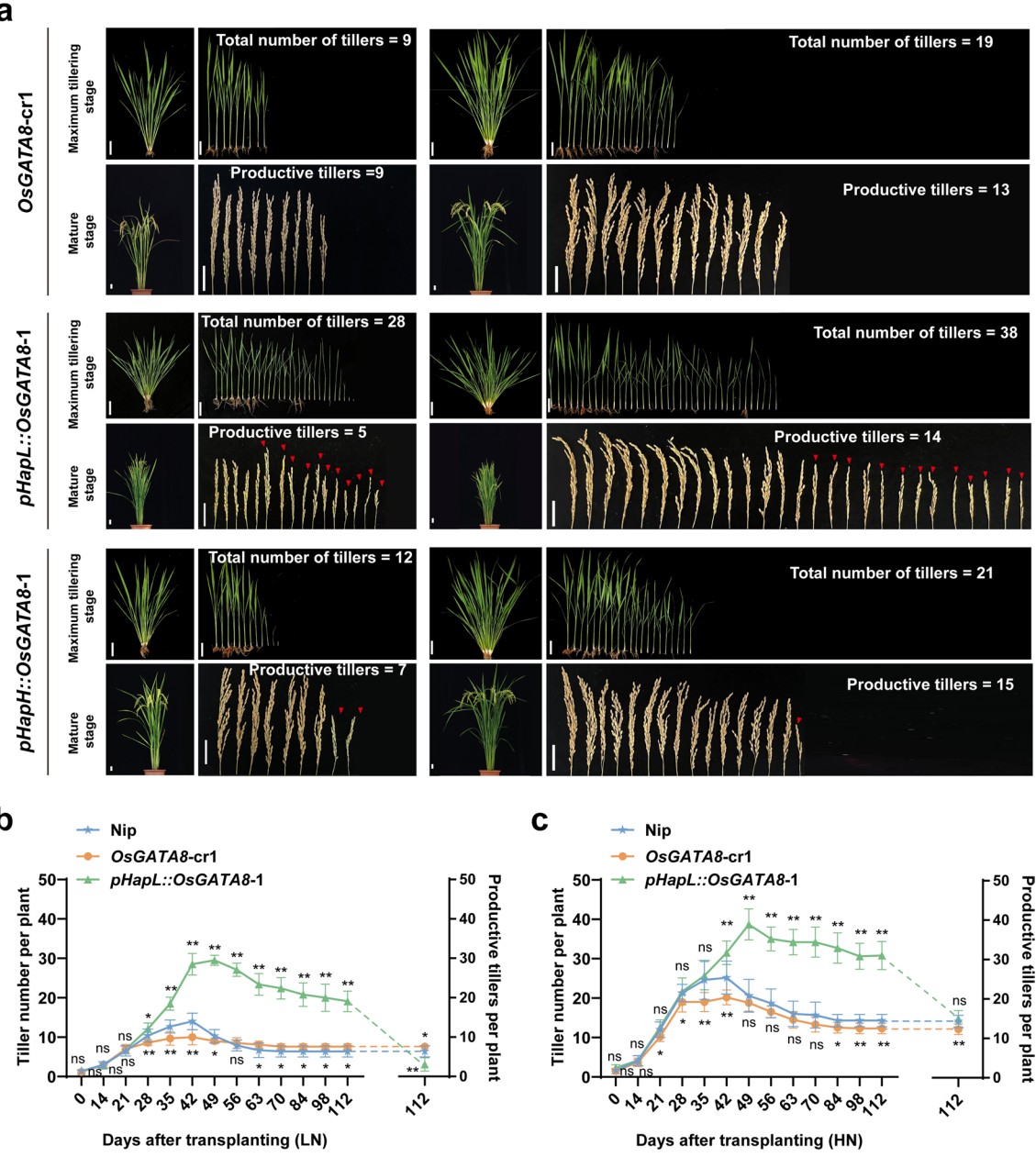

**Extended Data Fig. 5 | OsGATA8 positively regulates rice tillering formation.** (**a**) The phenotypes of the *OsGATA8*-cr knockout lines and *OsGATA8* transgenic lines exhibited tillering phenotypes at maximum tillering stage and effective panicle (productive tillers) phenotypes at mature stage under LN and HN conditions. The Nipponbare control has been shown in Extended Data Fig. 1b. Scale bar, 5 cm. LN, low nitrogen (75 kg/ha net nitrogen); HN, high nitrogen (300 kg/ha net nitrogen). Red arrows indicate nonproductive tillers. (**b**,**c**) The tillering dynamic changes of *OsGATA8* knockout and transgenic lines in **a** with rice transplanting days under LN and HN conditions. *n* = 10 plants. Data are presented as mean ± SD. *P* values were calculated with two-tailed Student's *t* test by comparing the values with those of the Nipponbare control (**P* < 0.05; ***P* < 0.01); for *P* values, see source data.

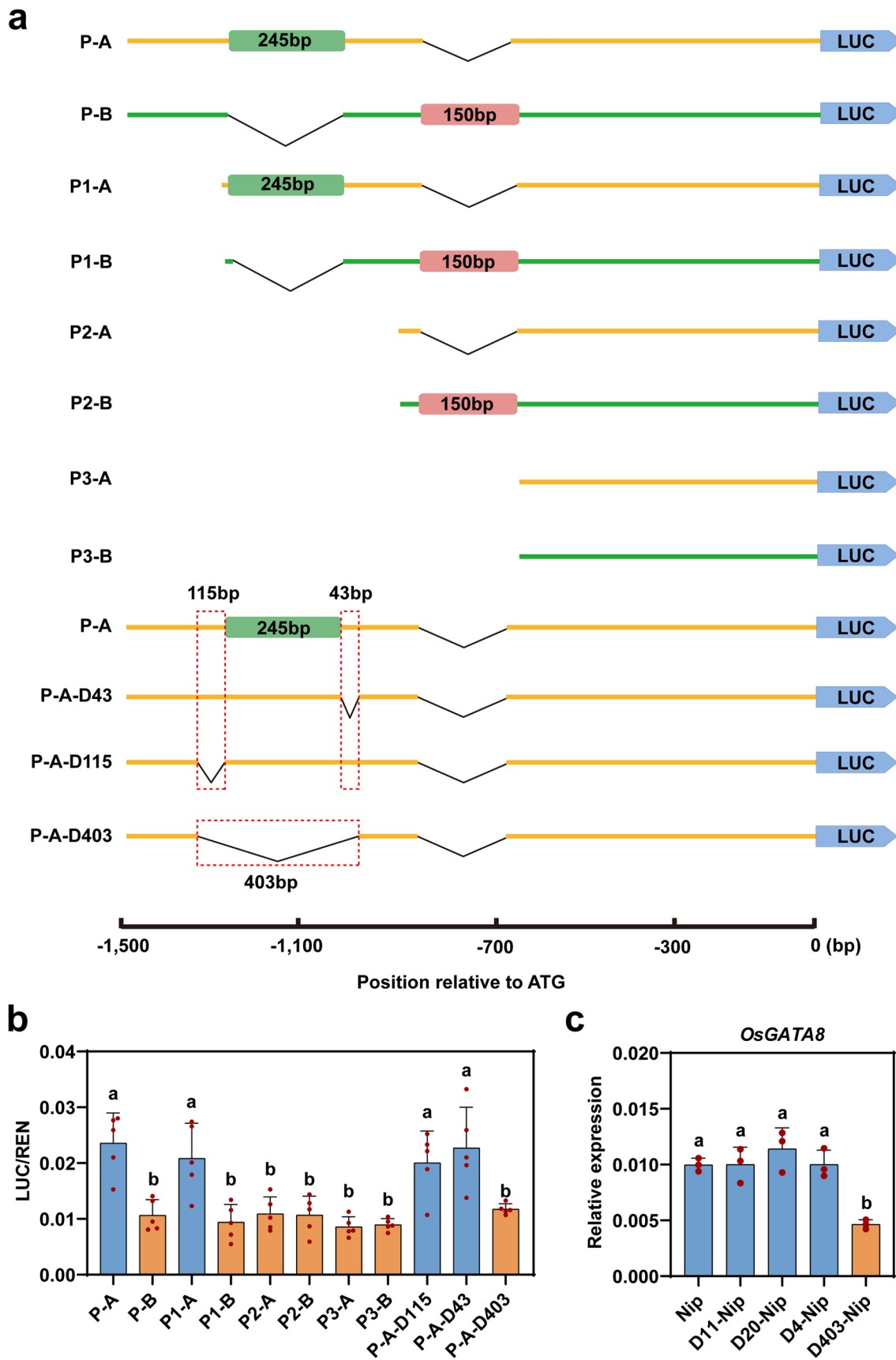

**Extended Data Fig. 6 | See next page for caption.**

**Extended Data Fig. 6 | A 245-bp deletion in the promoter of *OsGATA8* reduces the expression of the gene.** (a) Schematic representation of the reporter constructs for the luciferase assay. The black folded lines in P-A, P1-A and P2-A represent a 150-bp deletion relative to the red box. The black folded lines in P-B and P1-B represent a 245-bp deletion relative to the green box. The black folded lines in P-A-D43, P-A-D115 and P-A-D403 represent deletions relative to P-A. (b) Luciferase assays of the promoter activities of P-A, P-B, P1-A, P1-B, P2-A,

P2-B, $P_3$-A, P3-B, P-A-D115, P-A-D43 and P-A-D403 using rice protoplasts. Values represent mean ± SD derived from five independent samples of rice protoplasts. (c) Relative expression of four homozygous lines. Values represent mean ± SD derived from three individual rice seedlings. In **b** and **c**, different letters indicate significant differences ($P < 0.05$, one-way ANOVA, Duncan's new multiple range test); for $P$ values, see source data.

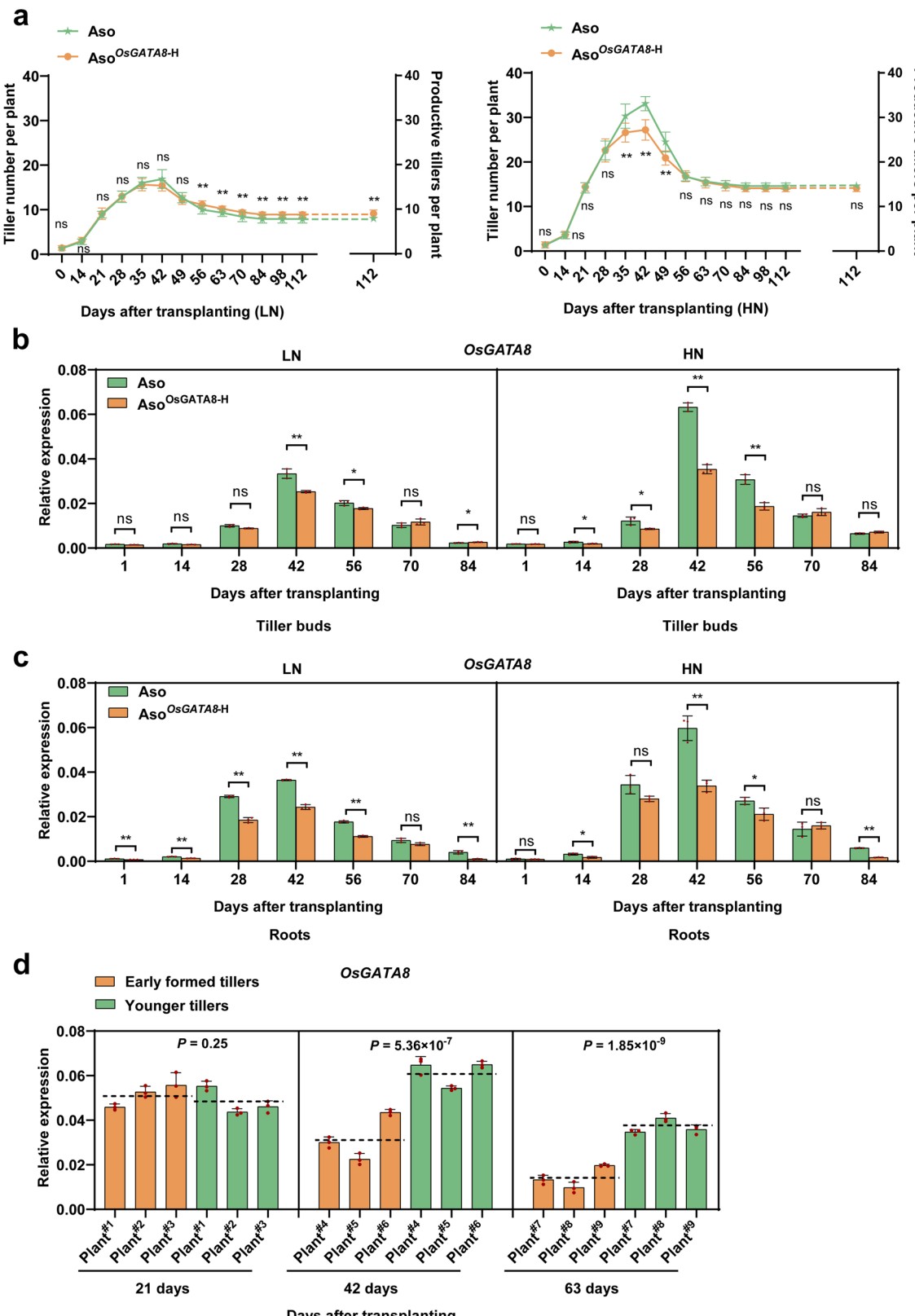

**Extended Data Fig. 7 | See next page for caption.**

**Extended Data Fig. 7 | OsGATA8 has a dual role in regulating rice tiller formation during vegetative growth and reproductive growth.** (**a**) The tillering dynamic changes of Aso and Aso$^{OsGATA8-H}$ lines with rice transplanting days under LN and HN conditions. $n$ = 10 plants. Data are presented as mean ± SD. (**b,c**) Dynamic changes of *OsGATA8* transcription levels in Aso and Aso$^{OsGATA8-H}$ tiller buds and roots with transplanting days. In **b**, total RNA was extracted from tiller buds of rice on different days after transplanting; in **c**, total RNA was extracted from roots on different days after transplanting. Values represent mean ± SD derived from root tissues of three individual plants. (**d**) Expression levels of *OsGATA8* in younger and older tillers at various developmental stages. Total RNA was extracted on different days after transplanting from the base of tillers that formed earlier or later. For each plant of a given developmental stage, three tillers that formed earlier and three that formed later were used for RNA extraction. Three plants were sampled for each developmental stage. Values represent mean ± SD derived from the tiller base of three individual plants. In **a**–**d**, *P* values were calculated with two-tailed Student's *t* test (*$P < 0.05$; **$P < 0.01$), for *P* values, see source data.

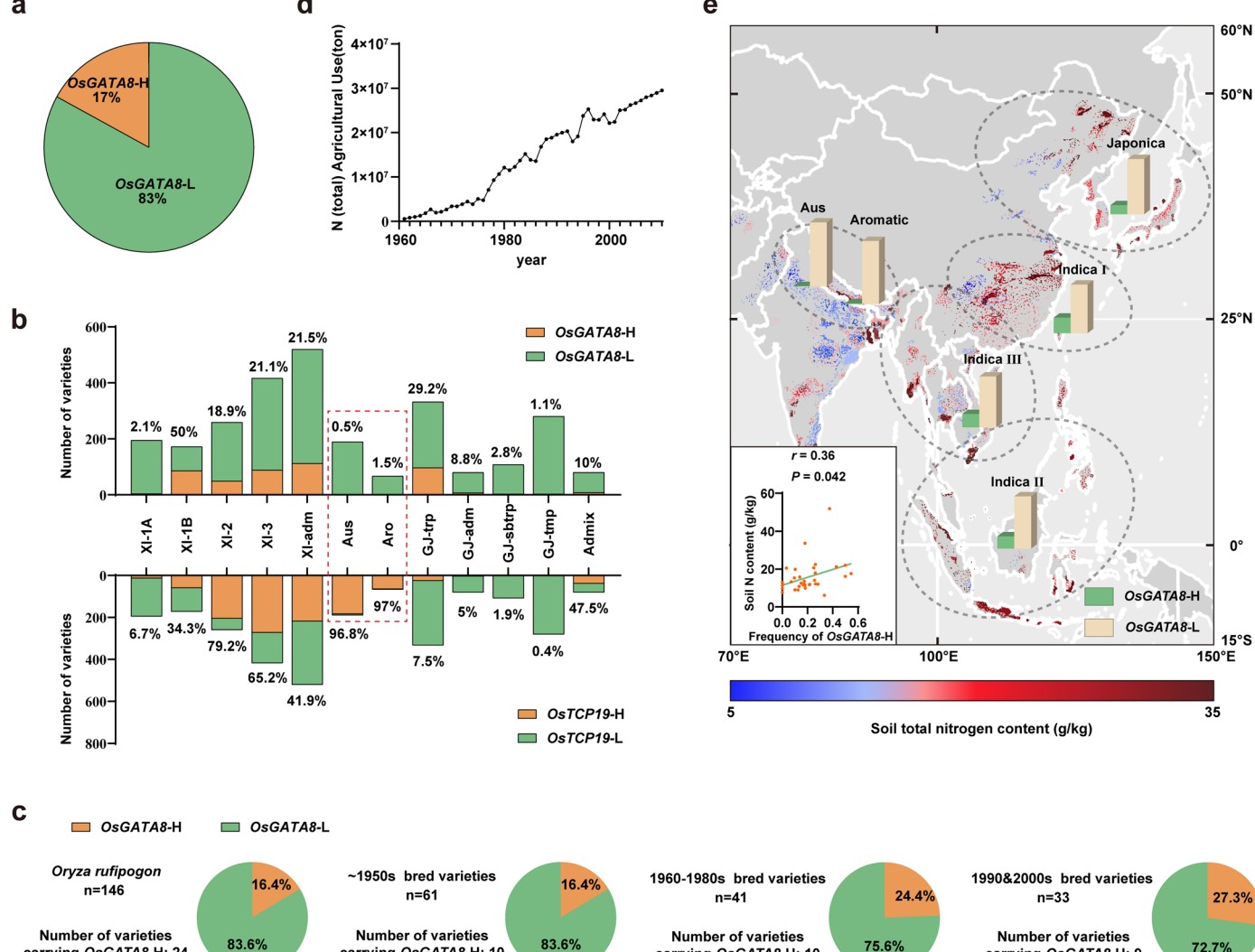

**Extended Data Fig. 8 | OsGATA8-H is associated with soil nitrogen availability.**
(**a**) The frequency of *OsGATA8*-H in the rice 3K population. (**b**) The frequency of *OsGATA8*-H and *OsTCP19*-H among different rice subgroups. The red box highlights the distribution difference between *OsGATA8*-H and *OsTCP19*-H in *Aus* and *Aro* rice. (**c**) The frequency of *OsGATA8*-H in wild rice populations and cultivated varieties from different years (1950–2000). (**d**) The amount of nitrogen fertilizer applied in China from 1961 to 2010. The original data are from the FAOSTAT Database (Food and Agriculture Organization Statistics). (**e**) Distribution of *OsGATA8*-H in different regions and subpopulations. The red area represents the soil nitrogen content in the region. The map was made using the 'maps' package in R[38]. Inset, Pearson's correlation coefficient analysis of the frequency of *OsGATA8*-H with soil total nitrogen content (g/kg) in 33 countries or regions worldwide. Data on soil nitrogen content in regions or countries are taken from a published article[11].

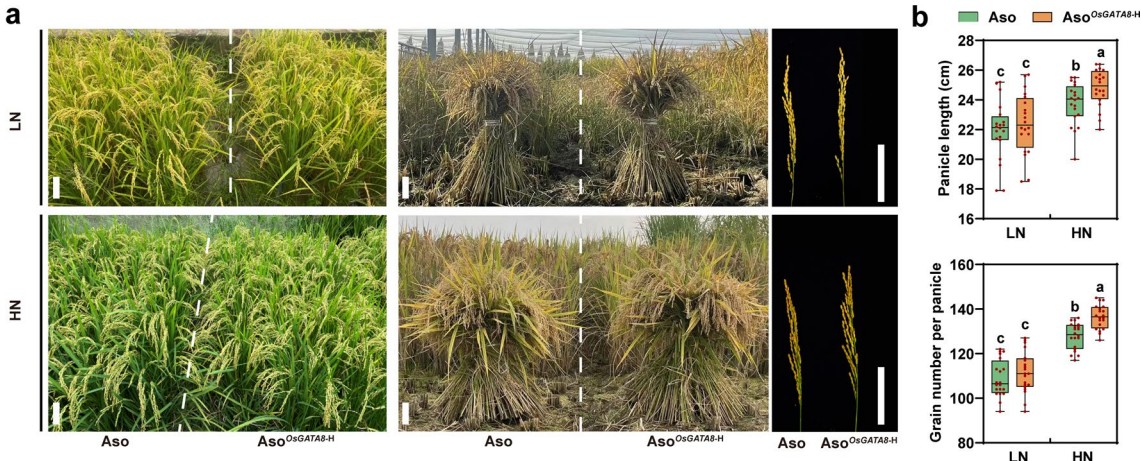

**Extended Data Fig. 9 | *OsGATA8*-H significantly improves rice yield and NUE.** (**a**) The phenotypes of *OsGATA8* NIL line at the mature stage under LN and HN conditions. (**b**) The panicle length and grain number per plant of Aso*^{OsGATA8-H}* and line at the mature stage under LN and HN conditions (*n* = 20 plants). In **a** and **b**, LN, low nitrogen (75 kg/ha net nitrogen); HN, high nitrogen (300 kg/ha net nitrogen). In **b**, box plots denote the 25th percentile, the median and the 75th percentile, with minimum to maximum whiskers; different letters indicate significant differences (*P* < 0.05, one-way ANOVA, Duncan's new multiple range test); for *P* values, see source data.

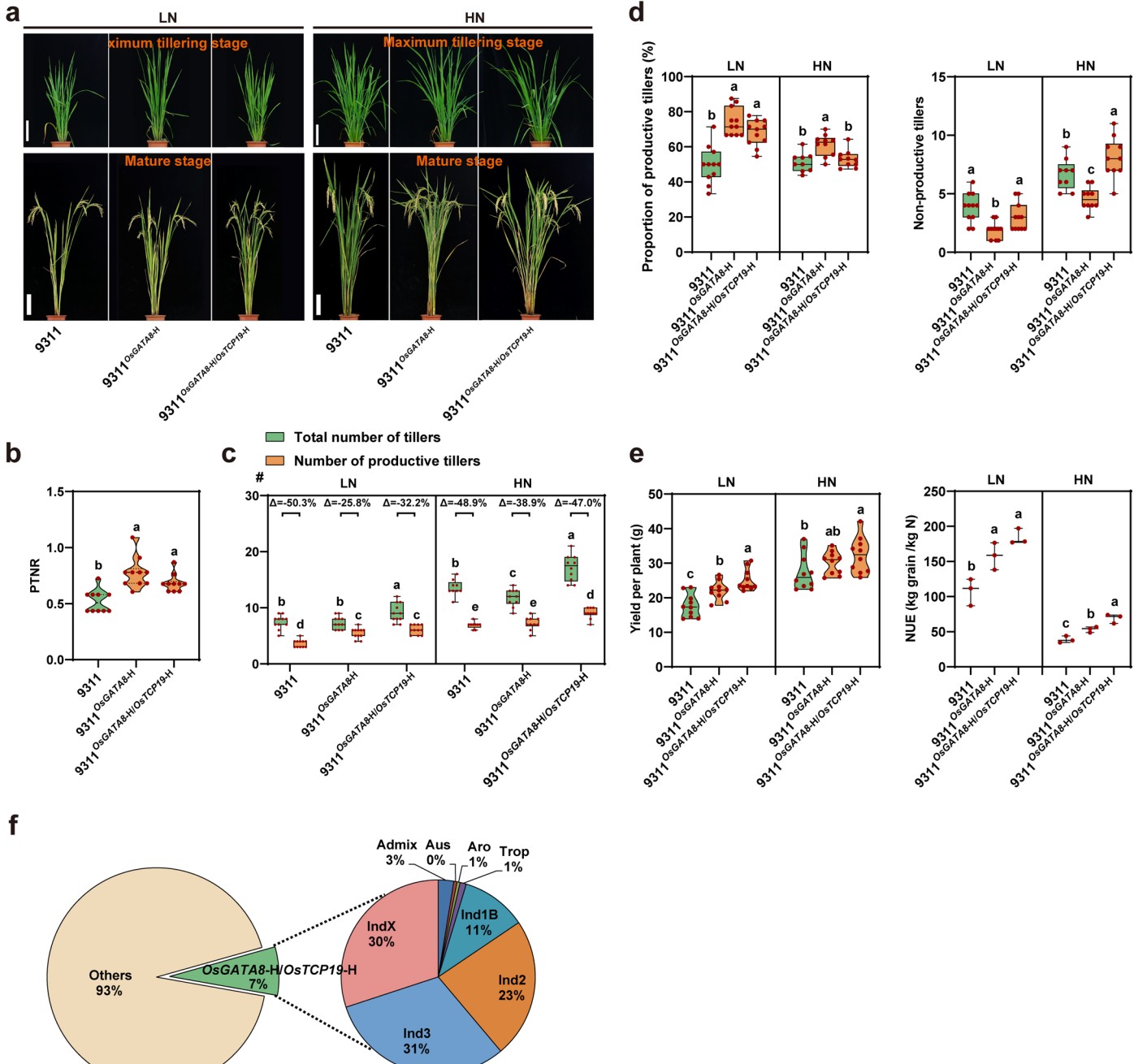

**Extended Data Fig. 10 | Combining the elite haplotypes *OsGATA8*-H and *OsTCP19*-H has breeding potential.** (**a**) Phenotypes of 9311, 9311$^{OsGATA8-H}$ and 9311$^{OsGATA8-H/OsTCP10-H}$ under LN and HN conditions at the maximum tillering stage and the mature stage ($n$ = 10 plants). Scale bars, 20 cm. (**b**) PTNR and the number of productive tillers of the genotypes in **a** under LN and HN conditions ($n$ = 10 plants). PTNR, productive tiller number ratio (productive tiller number under LN condition/productive tiller number under HN condition). (**c**) The numbers of the total number of tillers and productive tillers of the genotypes in **a** under LN and HN conditions. 'Δ' represents the percentage difference compared with the total number of tillers ($n$ = 10 plants). (**d**) The proportion of productive

tillers (PT%) and nonproductive tillers of the genotypes in **a** under LN and HN conditions ($n$ = 10 plants). (**e**) The yield per plant ($n$ = 10 plants) and NUE ($n$ = 3 plots) of the genotypes in **a** under LN and HN conditions. (**f**) Frequency of *OsGATA8*-H/*OsTCP19*-H in the rice 3 K population. In **b**–**e**, different letters indicate significant differences ($P$ < 0.05, one-way ANOVA, Duncan's new multiple range test); for $P$ values, see source data. LN, low nitrogen (75 kg/ha net nitrogen); HN, high nitrogen (300 kg/ha net nitrogen). In **b** and **e**, the bars within violin plots represent 25th percentile, median and 75th percentile. In **c** and **d**, box plots denote the 25th percentile, the median and the 75th percentile, with minimum to maximum whiskers.

# Reporting Summary

## Statistics

For all statistical analyses, confirm that the following items are present in the figure legend, table legend, main text, or Methods section.

| n/a | Confirmed | |
|---|---|---|
| ☐ | ☒ | The exact sample size (*n*) for each experimental group/condition, given as a discrete number and unit of measurement |
| ☐ | ☒ | A statement on whether measurements were taken from distinct samples or whether the same sample was measured repeatedly |
| ☐ | ☒ | The statistical test(s) used AND whether they are one- or two-sided *Only common tests should be described solely by name; describe more complex techniques in the Methods section.* |
| ☒ | ☐ | A description of all covariates tested |
| ☒ | ☐ | A description of any assumptions or corrections, such as tests of normality and adjustment for multiple comparisons |
| ☐ | ☒ | A full description of the statistical parameters including central tendency (e.g. means) or other basic estimates (e.g. regression coefficient) AND variation (e.g. standard deviation) or associated estimates of uncertainty (e.g. confidence intervals) |
| ☐ | ☒ | For null hypothesis testing, the test statistic (e.g. *F*, *t*, *r*) with confidence intervals, effect sizes, degrees of freedom and *P* value noted *Give P values as exact values whenever suitable.* |
| ☒ | ☐ | For Bayesian analysis, information on the choice of priors and Markov chain Monte Carlo settings |
| ☒ | ☐ | For hierarchical and complex designs, identification of the appropriate level for tests and full reporting of outcomes |
| ☐ | ☒ | Estimates of effect sizes (e.g. Cohen's *d*, Pearson's *r*), indicating how they were calculated |

*Our web collection on statistics for biologists contains articles on many of the points above.*

## Software and code

Policy information about availability of computer code

| Data collection | Sequencing reads were undertaken with the Hiseq-2500 sequencer (DAP-Seq and RNA-seq data); Bio-Rad CFX Manger v3.1 (qPCR data); Leica TCS SP5 (subcellular location);3000 Rice Genomes Project (genomic data of Rice 3k population);Rice Haplotype Map Project database (http://www.ncgr.ac.cn/RiceHap3). |
|---|---|
| Data analysis | Tassel 5 were used for GWAS. Microsoft Excel 2010, GraphPad Prism v8.0.1 and SPSS v17.0 (one-way ANOVA, Duncan's multiple range test) were used for statistical analysis. MEME chip and ChIPseeker were used for Motif prediction. 3DHISTECH and Caseviewer2.4 were used for FISH. R package pegas (version 4, paradis, 2010) was used for average nucleotide diversity and haplotype network of OsGATA8. MEGA7 (version 7.0) was used for constructing phylogenetic tree of OsAMTs. |

For manuscripts utilizing custom algorithms or software that are central to the research but not yet described in published literature, software must be made available to editors and reviewers. We strongly encourage code deposition in a community repository (e.g. GitHub). See the Nature Portfolio guidelines for submitting code & software for further information.

## Data

Policy information about <u>availability of data</u>

All manuscripts must include a <u>data availability statement</u>. This statement should provide the following information, where applicable:
- Accession codes, unique identifiers, or web links for publicly available datasets
- A description of any restrictions on data availability
- For clinical datasets or third party data, please ensure that the statement adheres to our <u>policy</u>

Data supporting the findings of this work are available within the paper and its Supplementary Information files. A reporting summary for this article is available as a Supplementary Information file. The datasets and genetic materials generated and analyzed during the current study are available from the corresponding author upon request. Sequence data of rice varieties can be found in our previous study (https://doi.org/10.1038/s41467-019-13187-1). Sequence data from this study can be found in the MSU database (http://rice.plantbiology.msu.edu/) under the following accession numbers: OsGATA8 (LOC_Os01g24070), OsTCP19 (LOC_Os06g12230), OsAMT3.2 (LOC_Os03g62200) and OsAMT1.2 (LOC_Os02g40730). Sequence for constructing the phylogenetic tree of AMTs (Extended Data Fig. 9) can be found in The Arabidopsis Information Resource database (https://www.arabidopsis.org/) or the MSU database (http://rice.plantbiology.msu.edu/). The DAP-sequencing and RNA-sequencing data have been uploaded to the website NCBI SRA (Sequence Read Archive) Database (https://www.ncbi.nlm.nih.gov/sra/PRJNA1103982 and https://www.ncbi.nlm.nih.gov/sra/PRJNA1104402). The source data are provided with this paper.

## Research involving human participants, their data, or biological material

Policy information about studies with <u>human participants or human data</u>. See also policy information about <u>sex, gender (identity/presentation), and sexual orientation</u> and <u>race, ethnicity and racism</u>.

| | |
|---|---|
| Reporting on sex and gender | No human participants involved. |
| Reporting on race, ethnicity, or other socially relevant groupings | No human participants involved. |
| Population characteristics | No human participants involved. |
| Recruitment | No human participants involved. |
| Ethics oversight | No human participants involved. |

Note that full information on the approval of the study protocol must also be provided in the manuscript.

# Field-specific reporting

Please select the one below that is the best fit for your research. If you are not sure, read the appropriate sections before making your selection.

☒ Life sciences ☐ Behavioural & social sciences ☐ Ecological, evolutionary & environmental sciences

For a reference copy of the document with all sections, see nature.com/documents/nr-reporting-summary-flat.pdf

# Life sciences study design

All studies must disclose on these points even when the disclosure is negative.

| | |
|---|---|
| Sample size | We planted 117 rice varieties and transgenic rice lines in the high nitrogen (HN, 300 kg/ha nitrogen fertilizer) and low nitrogen (LN, 0 kg/ha nitrogen fertilizer) fields. Samples sizes for each experiment is indicated in legends. Each experiment contains at least three biololgical replicates. |
| Data exclusions | No data was excluded from the analyses. |
| Replication | All experiments in this study were repeated independently at least three times. For qRT-PCR at least three biologically independent samples were used each time. For subcellular location, field and physiological and biochemical experiments, the results representative of three independent experiments. The number of biological replicates in each experiment was indicated in the legends. |
| Randomization | For field experiments, the varieties and transgenic lines were grown in a completely-randomized block design with four replicates. |
| Blinding | The investigators were blinded to group allocation during data collection. |

# Reporting for specific materials, systems and methods

We require information from authors about some types of materials, experimental systems and methods used in many studies. Here, indicate whether each material, system or method listed is relevant to your study. If you are not sure if a list item applies to your research, read the appropriate section before selecting a response.

## Materials & experimental systems

| n/a | Involved in the study |
|---|---|
| ☐ | ☒ Antibodies |
| ☒ | ☐ Eukaryotic cell lines |
| ☒ | ☐ Palaeontology and archaeology |
| ☒ | ☐ Animals and other organisms |
| ☒ | ☐ Clinical data |
| ☒ | ☐ Dual use research of concern |
| ☐ | ☒ Plants |

## Methods

| n/a | Involved in the study |
|---|---|
| ☒ | ☐ ChIP-seq |
| ☒ | ☐ Flow cytometry |
| ☒ | ☐ MRI-based neuroimaging |

## Antibodies

| Antibodies used | DYKDDDDK Tag Antibody (MA1-91878-BTIN) was used for IP. |
|---|---|
| Validation | DYKDDDDK Tag Antibody (MA1-91878-BTIN) can be found at ThermoFisher (https://www.thermofisher.cn/) |

## Dual use research of concern

Policy information about dual use research of concern

### Hazards

Could the accidental, deliberate or reckless misuse of agents or technologies generated in the work, or the application of information presented in the manuscript, pose a threat to:

| No | Yes | |
|---|---|---|
| ☒ | ☐ | Public health |
| ☒ | ☐ | National security |
| ☒ | ☐ | Crops and/or livestock |
| ☒ | ☐ | Ecosystems |
| ☒ | ☐ | Any other significant area |

### Experiments of concern

Does the work involve any of these experiments of concern:

| No | Yes | |
|---|---|---|
| ☒ | ☐ | Demonstrate how to render a vaccine ineffective |
| ☒ | ☐ | Confer resistance to therapeutically useful antibiotics or antiviral agents |
| ☒ | ☐ | Enhance the virulence of a pathogen or render a nonpathogen virulent |
| ☒ | ☐ | Increase transmissibility of a pathogen |
| ☒ | ☐ | Alter the host range of a pathogen |
| ☒ | ☐ | Enable evasion of diagnostic/detection modalities |
| ☒ | ☐ | Enable the weaponization of a biological agent or toxin |
| ☒ | ☐ | Any other potentially harmful combination of experiments and agents |

## Plants

| | |
|---|---|
| Seed stocks | The seeds of the 117 accessions were collected, stored, and supplied by the State Key Laboratory of Crop Genetics & Germplasm Enhancement and Utilization, Jiangsu Collaborative Innovation Center for Modern Crop Production, Nanjing Agricultural University, China. |
| Novel plant genotypes | We used the CRISPR/Cas9 technology in generating genome-edited plants. Agrobacterium-based transformation was used to generate the overexpression lines and the genome-edited rice plants. |
| Authentication | PCR and Sanger sequencing was used to verify the genome edits generated at the designated genomic targets. RT-qPCR are used to verify the expression of the transgenes. |

