## [Peer Review File · Nature Genetics]

Peer Review Information

Manuscript Title: The elite haplotype OsGATA8-H coordinates nitrogen uptake and productive tiller formation in rice

Corresponding author name(s): Professor Jianmin Wan, Professor Chunming Wang

Reviewer Comments & Decisions:

Decision Letter, initial version:
--

31st Jul 2023

Dear Professor Wan,

Your Article, "The elite haplotype OsGATA8-H confers yield increase by coordinating nitrogen uptake and productive tiller formation in rice" has now been seen by 3 referees. You will see from their comments copied below that while they find your work of considerable potential interest, they have raised quite substantial concerns that must be addressed. In light of these comments, we cannot accept the manuscript for publication, but would be interested in considering a substantially revised version that addresses these serious concerns.

We hope you will find the referees' comments useful as you decide how to proceed. If you wish to submit a substantially revised manuscript, please bear in mind that we will be reluctant to approach the referees again in the absence of major revisions.

To guide the scope of the revisions, the editors discuss the referee reports in detail within the team with a view to identifying key priorities that should be addressed in revision. As you will see from these comments, all the referees have identified important aspects of the study that need to be substantially improved. Reviewer #1 notes that EPNR should not be regarded as a measure for NUE but rather as a proxy, and that the role of OsGATA8 in regulating NUE needs to be better verified in a field study. Reviewer #2 doesn't think that the major conclusion about the coordination of nitrogen uptake and productive tiller formation by OsGATA8 is sufficiently supported by the current data. Reviewer #3 points out a technical flaw due to ambiguous definition of NUE and EPNR and suggests caring more about the timing of floral transition in addition to the tiller number controls. We particularly ask that you address all referee comments as thoroughly as possible with appropriate revisions. We hope that you will find the prioritized set of referee points to be useful when revising your study.

If you choose to revise your manuscript taking into account all reviewer and editor comments, please

highlight all changes in the manuscript text file. At this stage we will need you to upload a copy of the manuscript in MS Word .docx or similar editable format.

*2) If you have not done so already please begin to revise your manuscript so that it conforms to our Article format instructions, available here. Refer also to any guidelines provided in this letter.

Please be aware of our guidelines on digital image standards.

[redacted]

If you wish to submit a suitably revised manuscript we would hope to receive it within 6 months. If you cannot send it within this time, please let us know. We will be happy to consider your revision so long as nothing similar has been accepted for publication at Nature Genetics or published elsewhere. Should your manuscript be substantially delayed without notifying us in advance and your article is eventually published, the received date would be that of the revised, not the original, version.

Nature Genetics is committed to improving transparency in authorship. As part of our efforts in this direction, we are now requesting that all authors identified as 'corresponding author' on published papers create and link their Open Researcher and Contributor Identifier (ORCID) with their account on the Manuscript Tracking System (MTS), prior to acceptance. ORCID helps the scientific community achieve unambiguous attribution of all scholarly contributions. You can create and link your ORCID from the home page of the MTS by clicking on 'Modify my Springer Nature account'. For more

information please visit please visit www.springernature.com/orcid.

Thank you for the opportunity to review your work.

Sincerely,
Wei

Wei Li, PhD
Senior Editor
Nature Genetics
New York, NY 10004, USA
www.nature.com/ng

Reviewers' Comments:

Reviewer #1:

Remarks to the Author:

In the present study, the authors cultivated 117 rice varieties under high and low nitrogen conditions and subjected some proxies related to NUE, such as plant height and effective panicle number ratio (LN/HN), to GWAS. They identified a QTL within a linkage disequilibrium block on chromosome 1, in which three genes are located. One of them showed the lowest expression in rice varieties with higher NUE and encoded the transcription factor GATA8.

Two GATA8 knockout lines showed higher effective panicle number, while the overexpressed line exhibited less, confirming the involvement of GATA8 in tiller formation. To investigate the cause for variability in GATA8 expression among haplotypes, the authors identified a deletion of 245-bp in its promoter region, significantly reducing its expression. This was validated by generating a genome-edited mutant line with a 403-bp deletion in the GATA8 promoter, which includes the 245-bp indel region. Again, this line showed reduced GATA8 expression and higher effective panicle number than the wild-type.

Interestingly, the 245-bp deletion in the GATA8 promoter region defines a distinct haplotype (GATA8-H), which is geographically and historically well-differentiated. Specifically, GATA8-H is predominantly found in regions with nitrogen-rich soils and has been cultivated since the 1960s. This suggests that the GATA8-H haplotype may have been under human selection under high nitrogen conditions.

The authors further showed that high GATA8 expression represses the expression of AMT3.2 and TCP19, resulting in lower ammonium uptake and a higher number of productive tillers, respectively. To support their conclusion, the researchers conducted several in-vitro experiments that provided strong evidence of GATA8 direct binding to the promoters of AMT3.2 and TCP19, leading to the repression of their transcription.

Finally, building upon a previous study showing that a TCP19 haplotype exhibits higher NUE under low nitrogen conditions, the authors created another NIL carrying both GATA8-H and TCP19-H.

Remarkably, this line displayed higher NUE than NILGATA8-H alone. These results demonstrate that rice germplasm carrying both GATA8-H and TCP19-H leads to higher yield and NUE.

These findings provide valuable information on developmental genes that significantly contribute to NUE in rice, which will have implications for improving rice varieties and their NUE. The researchers'

approach of not only identifying GATA8 as a critical transcription factor affecting NUE but also exploring the allelic diversity of this gene is both insightful and original. Indeed, the congruence between the GATA8-H haplotype found in more modern rice varieties, and the 245-bp deletion in the GATA8 promoter associated with higher NUE, create a substantial scientific advancement.

However, to enhance the credibility and robustness of the above statements, some critical points and open questions should be addressed:

Major points:

- Lines 82 to 89: EPNR should not be regarded as a measure for NUE but rather as a proxy, because panicle number is not correlating with plant biomass or grain yield, esp. when grain number and size (TKW) are low. EPNR may sometimes serve as proxy, but this statement as well as the conclusion in line 110/111 are not justified.
- Fig. S1a: The Manhattan plot does not really identify a significant QTL, as the probability score of the marker-trait association in the QTL does not appear to be significant but similar as outside of the confidence interval.
- Fig. 3: the genetic and phenotypic analysis of the effect of OsGATA8 deletion in crispr-lines has been done with -cr2, whereas the mRNA-seq. and transcriptional regulation of AMTs has been done with another independent line -cr1. Thus, transcriptional results in the -cr1 line may be due to different off-target or side effects and have not been done in a confirmed background. At least, transcriptional results should be repeated in the -cr2 line.
- Fig. 3A: To consolidate results and include a positive control, OsAMT3.2 transcript levels should be shown also in pHapH::OsGATA8 lines.
- Fig. 3g-h: Why does the knock-out of OsAMT3.2 lead to a higher total number of tillers? Does this mean that ammonium uptake suppresses tiller formation in rice?
- Figs. 3h and 4g: i) The N-dependent regulation of OsGATA8 expression is not clear and should be laid down. ii) If OsGATA8 is upregulated by high N (as to be assumed) and promotes excessive tiller formation via OsTCP19 suppression, why is there so little difference between Nip and OsGATA-cr1 in the number of productive tillers between LN and HN? Actually, this difference should be higher at HN than at LN.
- Fig. S5-a shows a heatmap of differentially expressed genes involved in the nitrogen metabolism pathway, serving as an exploratory method to subsequently select candidate genes affected by GATA8. However, there are further important nitrogen transporters incl. NRT1.1B and AMT2.3 that were not considered. Moreover, AMT3.2 was even among the least upregulated, and AMT1.2 does not even appear in the heatmap. This creates a discrepancy for the follow-up experiments. What role do these genes play downstream of GATA8?
- The main results rely on the formation of productive tillers or panicles as proxy for NUE. This raises the question whether OsGATA8-dependent NUE is the same when using total shoot biomass at the end of tillering (maximum tillering stage) under LN versus HN rather than using grain yield at maturity under LN versus HN. The current data imply that GATA8 may just be a regulator of tillering (which itself is N-dependent), rather than a regulator of NUE. Therefore, it should be verified (in a field study) how GATA8 affects NUE on the basis of total shoot biomass at the end of tillering.
- A major question is to what extent is OsGATA8 a NUE determinant or rather a TF determining one of the major yield components, i.e. number of spike-forming tillers? When OsGATA8 promotes excessive tiller formation, N demand by the plant is rising, this should increase ammonium uptake capacity. However, OsGATA8 suppresses OsAMT3.2. This apparent contradiction should be resolved.

Minor points:

- Line 109: the term 'productive tiller' is not defined. i) Do the authors mean ear- or spike-forming

tillers? ii) At what stage does the formation of an ear or spike make a tiller productive? Please clarify in the text or methods.

- Fig. S2b says tiller node, text (line 102) says tiller bud – clarify!
- Line 53: Clarify gene names to the reader by using synonyms, as done in line 52. For example: NRT1.1B/NPF6.5, and NRT1.1A/NPF6.1.
- Line 96: OsGATA8 should not be in italics as it refers to the encoded protein.
- Line 188: There is no Fig. 1f. Maybe labelling in the fig. Is lacking.
- Line 219-222: These two sentences may be misleading, as it is not the rainfall but the soil microbiome that may lead to ammonium production from nitrate. Re-formulate.
- Line 285: CHL1 is outdated. Use OsNRT1.1B/NPF6.5 instead.
- Line 287: Use OsNRT1.1A/NPF6.1 instead.
- General shortcoming in the data display: Many plots do not show units on y-axes (e.g. is in Fig. S2g the tiller no. per plant or per pot? Is in Fig. S2h proportion in %? All y-axes should be properly assigned with units.
- Consider using a different graphical representation instead of bar graphs. Generally, boxplots accompanied by P-values are less susceptible to visual distortion, especially when the sample size is large. For example, it is difficult to discern clear differences between productive tillers in Fig. 1c, 2d, 3h, or 4g, or in yield in Fig. S7e. Box plots would be more appropriate here.
- Legend in Fig. S3A: In the scheme it remains unclear whether the 245 bp sequence is present in P2-L and P2-H, or not.

Reviewer #2:

Remarks to the Author:

The manuscript by Wu et al. reported the identification of OsGATA8 and its elite allele in coordinating the nitrogen uptake and productive tiller formation in rice. In general, this work gives a very novel angle to dissect the NUE of rice, and also provides a new strategy for NUE improvement by increasing the proportion of productive tiller formation during nitrogen fertilizer use. Possibly, this work would inspire the following research in the field and bring some breakthroughs toward the NUE manipulation in crops. However, for the current version of this paper, it still needs substantial revision for its final publication.

1. The major conclusion of this work is about the coordination of nitrogen uptake and productive tiller formation by OsGATA8. However, it is very difficult to understand how these two processes are coordinated from the presented data in this manuscript, even though the authors showed that OsGATA8 can regulate both OsAMT3.2 and OsTCP19, the components responsible for ammonium uptake and tillering modulation, respectively. The specified role of OsGATA8 in determining the transformation of young tiller to productive tiller or non-productive tiller should be investigated to support its proposed function in this work. The authors may consider whether OsGATA8 is specifically repressed in the young tiller that can be transformed into the productive tiller, in which the nitrogen uptake is greatly enhanced and the following development is thereby promoted.

2. The authors deemed that OsGATA8 regulates the productive tiller proportion via targeting the OsTCP19. This conclusion requires the precondition that OsTCP19 is involved in modulating the productive tiller proportion. Nonetheless, Figure 4 e-g clearly showed that loss-of-function of OsTCP19 (OsTCP19-cr1) results in the increase of both the total tiller and the productive tiller, and the proportion of the productive tiller is unaffected. Thus, the repression of OsTCP19 cannot specifically promote the formation of non-productive tiller. It still lacks the evidence to support the conclusion that

“This OsGATA8-OsTCP19 module promotes the formation of non-productive tillers (line 215)”.

3. The authors proved that 245-indel in the promoter of OsGATA8 is the causal variation responsible for EPNR variation in different rice varieties. To support this conclusion, the authors provide a series of evidence including association assay, promoter activity analysis, different haplotype transgenic assay, and genome-editing. Even though, to make this finding solid, it still needs more evidence:

1) There are 21 indels and 108 SNPs in the promoter region, the authors only select two large indels as the candidate. It needs more evidence to exclude other variations.

2) The genome-editing in the promoter region would provide very solid evidence to support the vital role of the 245 bp-indel in controlling the expression of OsGATA8. However, the D245 genome-editing line carries over 400 bp deletion in the promoter. It is hard to say that the reduced expression of OsGATA8 is resulted from the 245 bp-indel or other deletion fragment.

3) The authors created the pHapL- and pHapH-OsGATA8 transgenic plants, and pHapL-OsGATA8 line displayed more non-productive tiller formation. This result strongly supports the functional variation occurring in the promoter. However, it would be necessary to examine more independent transgenic lines of pHapL- and pHapH-OsGATA8. Only using one transgenic line cannot give a solid conclusion. Unfortunately, this situation occurs through the whole manuscript (the over expression transgenic assays of OsGATA8 and AMT3.2 only have one line).

4) It is necessary to present the data of the total tiller, productive tiller and the proportion of the productive tiller of NILOsGATA8-H. These results will further support the function of OsGATA8-H.

4. The authors find that OsGATA8-H is subjected to the artificial selection as the increased use of nitrogen fertilizer. But, I cannot find any explanation or discussion about the reason of this phenomenon in this manuscript. As the authors proposed, the increased application of nitrogen would result in the dramatic increase of the non-productive tiller formation, thus the artificial selection toward OsGATA8-H will decrease the non-productive tiller formation under high nitrogen. Examination of the proportion of productive tiller using the OsGATA8-H/L varieties under low- and high nitrogen will be helpful to answer this question.

5. Some minor points:

1) The authors should give more clear statement about how to select the three-candidate gene based on the GWAS signal.

2) The method for total tiller number investigation needs to be stated clearly. All the plant materials have the same growth periods? If not so, how to keep the consistency for each plant material during collecting the total tiller number?

3) The difference of OsAMT3.2 expression in Aso and NILOsGATA8-H is very slight in the shoot (Supplementary Figure 14 c). The authors should examine OsAMT3.2 expression in the root.

4) The deletion fragment in D245 is 403-bp (line 125) or 430-bp (line 491)?

5) Both the figure legend and Metatrails and Methods should be supplemented with more detail, and some mistakes should be avoided. For example, the information about the NILs (the precise size of the introduction fragment) should be provided; the method for “15N accumulation assay” (lines 733-735) showed that “Then, the seedlings were pretreated with 2 mM (NH₄)₂SO₄ and 2 mM KNO₃ for 1 week and transferred to the solutions with 0.2 mM and 2 mM (15NH₄)₂SO₄ and 15KNO₃ for 10 min”, but I cannot find any data related to the 15N-nitrate uptake assay.

Reviewer #3:

Remarks to the Author:

The manuscript by Wu et al. describes a series of extensive works related to NUE (nitrogen use efficiency) related QTL in rice. First, they identified OsGATA8 is the NUE QTG (Fig.1). Then, they

identified the deletion in OsGATA8 is the key QTN which may reduce the OsGATA8 expression in cultivars (Fig2, SFig.3). Next, they identified OsAMT3.2 and 1.2 as target genes of OsGATA8, which can promote nitrogen uptake and increase the yields (Fig.3). Furthermore, they identified that OsTCP19 as another target of OsGATA8 to control tiller number in rice (Fig.4). Finally, based on haplotype analysis, they proposed OsGATA8-H is an elite haplotype in increase NUE (Fig.S14) and find a better combination of the related QTLs (Fig, S15) and propose a model to explain NUE (Fig.5).

Major comments;

When I read this manuscript, I like this work and appreciated it very much. However I read it repeatedly, I realized a flaw on this work due to ambiguous definition of NUE and EPNR. EPNR is a kind of new word in this field. The biological causes to affect EPNR can be various. But we do not know anything of the target of OsTCP19 yet.

In addition, we know that NUE can be affected by subjective cultivation environments. For example, an elite cultivar in temperate climates cannot be the one in tropical climates since it often flower too early due to its photoperiodic flowering. It indicate that the same cultivars give us very different NUE due to the timing of floral transition under subjected cultivation environment. Apparently the regulation of tiller number controls such as by SL biosynthesis and signaling are largely independent of the photoperiodic flowering, thus, the timing of floral transition can affect NUE or EPNR largely. This also tell us that to consider NUE, we really care about the genetic background. However, in this paper, they used several cultivars such as NIP and 9311 without caring about this point. These indicate that the data described here can be drastically changed greatly when they change the transplanting date in the same field or cultivation areas. It may happen when they changes the photoperiod or ambient temperature too.

The authors were not intended but forced to be trapped to consider the balance between tiller number control and floral transition when they designed the experiment. Then, in this work, it was not enough.

Thus, they have to check the related traits under clearly distinct cultivation conditions such as short-days and long-days at least.

Minor comments,

Identification of target genes of OsGATA8 by DAP-SEQ data is not easy to understand (Fig. S10). The authors can present how OsGATA-8 can bind the promoter regions of downstream target genes by RNA-SEQ data.

OsTCP19 target genes should be analyzed by stage specific- and tissue specific- RNA-SEQ analysis to reveal the genuine biological meaning of EPNR or NUE.

Can their EPNR be related to fertility rate and/or panicle size and grain size? The readers would like to know it. If no such data involved in specific biological events, this paper is not so biological although it is really agronomical.

Author Rebuttal to Initial comments

2.	Reviewers'	Comments:
		#1:
Reviewer		
Comment: Lines 82 to 89: EPNR should not be regarded as a measure for NUE but rather as a proxy, because panicle number is not correlating with plant biomass or grain yield, esp. when grain number and size (TKW) are low. EPNR may sometimes serve as proxy, but this statement as well as the conclusion in line 110/111 are not justified.		

Response: We agree that EPNR (which has been renamed as PTNR in the revised manuscript for “productive tiller number ratio” to be consistent with the rest of the manuscript) may not serve as a direct measurement of NUE but rather as a proxy/parameter. The extent to which PTNR serve as an accurate indicator of NUE can be affected by other agronomy parameters, such as grain number, grain size, and flowering time. To this end, we have collected the following data to support PTNR as a valid a proxy/parameter of NUE in our experimental setup.

The number of productive tillers has a major influence on yield, and tiller number has been applied as a key parameter in NUE-related studies in rice and led to the identification of several key regulators of NUE and yield in rice. For instance, Wu et al. identified NGR5 (*nitrogen-mediated tiller growth response 5*) as a key regulator of nitrogen use efficiency and rice yield through a genetic screening for mutants defective on tillering response to nitrogen supply². Liu et al. performed GWAS on tiller (panicles per plant) response to nitrogen (TRN) and identified *OsTCP19* as a key regulator of NUE and yield¹; in the same study, the authors also provided evidence that NUE has the highest correlation with productive tiller number.

To verify that PTNR can be used as a key criterium of NUE for our GWAS study, we performed a correlation analysis of NUE (calculated by dividing yield by the total N input) with various agronomy traits, including productive tiller number, plant height, heading date, 1,000-grain weight, grain length and grain width between high N and low N supply (**Item 1 in this response letter, or Figure S2 in the revised version**). Consistent with previous studies, our results also suggest that NUE is most closely correlated with the productive tiller number ratio (PTNR) and the plant height ratio (PHR) (**Item 2 in this response letter, or Figure S3 in the revised version**). These results suggest that PTNR is an effective indicator of NUE, thus we performed PTNR-based GWAS study to identify key regulators of NUE.

To substantiate the role of *OsGATA8* as a negative regulator of NUE (line 110/111 in the original version of the manuscript), we collected the following lines of evidence:

1. We examined yield of *OsGATA8* near-isogenic lines (NILs) under LN and HN conditions. We found that the Aso^{*OsGATA8*-H} plants, which have a relatively lower level of *OsGATA8* expression, displayed increased yield under both LN and HN conditions.

2. At the mature stage, $Aso^{OsGATA8-H}$ plants displayed higher biomass and total nitrogen content under both LN and HN conditions, indicating a higher NUE.

These results were based on experiments using yield, biomass, or total nitrogen content, thus are more direct measurements of NUE. *OsGATA8* negatively correlates with the yield, N content, and nitrogen use efficiency (NUE) (Item 10b, 11 and 22 in this response letter, or Fig. 5, S25 and S30 in the revised version), which supports the conclusion that *OsGATA8* is a negative regulator of NUE.

Item 1: The number of productive tillers and plant height positively correlate with soil nitrogen content.

(a-b) Number of productive tillers, plant height, heading date, thousand-grain weight, grain length, and grain width of 117 varieties under low and high nitrogen field conditions. These results show

that plant height and the number of productive tillers in rice positively correlate with soil nitrogen content.

Item 2:

PTNR and PHR positively correlate with NUE.

(a-d) Pearson's correlation coefficients were analyzed for NUE of 117 varieties with PTNR, PHR, HDR, TGWR, GLR, and GWR under LN and HN field conditions. Rice plants with high PTNR and PHR show higher NUE compared with those with low PTNR and PHR under both high

nitrogen (HN) and low nitrogen (LN) conditions.

NUE = yield per plant/nitrogen applied per unit area. LN, low nitrogen (75 kg/ha net nitrogen); HN, high nitrogen (300 kg/ha net nitrogen); PTNR, productive tiller number ratio (productive tiller number under LN condition / productive tiller number under HN condition); PHR, plant height ratio (Plant height under LN condition / Plant height under HN condition); HDR, heading date ratio (Heading date under LN condition / Heading date under HN condition); TGWR, thousand-grain weight ratio (Thousand-grain weight under LN condition / Thousand-grain weight under HN condition); GLR, grain length ratio (Grain length under LN condition / Grain length under HN condition); GWR, grain width ratio (Grain width under LN condition / Grain width under HN condition).

Comment: Fig. S1a: The Manhattan plot does not really identify a significant QTL, as the probability score of the marker-trait association in the QTL does not appear to be significant but similar as outside of the confidence interval.

Response: We thank the reviewer for this comment. For gene discovery by GWAS, minimizing the false positive rate often has a higher priority than controlling the false negative rate. In most of these cases with large sample size, genome-wide significance thresholds will be required. In practice, in a small sample size (100~500), an over-stringent threshold would abolish any GWAS signals.

In this study, we adopt a relatively loose threshold (P value $< 10^{-4}$) to declare the nominal signals, which would unavoidably result in false positives. However, there is no doubt that true signals can be uncovered as well. To rule out the potential false positives, we proposed the following four filtering steps to narrow down the candidate genes list, followed up by functional validations described in our previously papers. In the revised version of manuscript, we have cited the strategy of candidate gene prioritization in our previously published paper³.

We re-described the strategy of our filtering procedure.

Firstly, SNVs meeting nominal threshold, p value $< 1 \times 10^{-4}$, at least two times in the successive three years, will be regarded as candidate TAS (Trait-Associated SNVs). Secondly, we will filter out the loci in which there is no gene in the LD block through gene annotation and LD analysis. We estimated 95% confidence interval of top TAS by 1000 permutation tests. As you can see, the 95% confidence intervals are wider than LD blocks (95% CI of PTNR is 13,114,467~13,993,067, 95% CI of PHR is 13,113,314 ~ 13,993,097). Thirdly, we further filter out some loci based on reference query, related public rice database, pathway and network analyses. Lastly, we performed real time PCR (RT-PCR) for multiple potential candidate genes

within the LD block to obtain the most likely candidate genes for follow-up functional validation through complementation tests.

Four further steps were carried out to validate the two candidate genes as described in the main text as follows:

1. Annotation: On chromosome 1, the candidate gene of PTNR and PHR were predicted to reside in the region spanning bp chr1:13,548,357 to 13,572,267 (Item3a in this response letter, or Fig. S4a in the revised version), containing 457 gene-localized SNPs (Table S1 and S2). Three genes are present in this LD block (Item3b in this response letter, or Fig. S4b in the revised version).
2. Expression: By gene expression analyses, we found that *Os01g0343300*, which encodes a GATA family transcription factor, was transcriptionally induced by high nitrogen application (Item3b, 8a in this response letter, or Fig. S4b in the revised version).
3. Haplotype analysis: The two *OsGATA8* haplotypes contain two large indel variations in the core population (Item 4 in this response letter, or Fig. S17 in the revised version), and showed differences in PTNR and PHR (Item 3c in this response letter, or Fig. S4c in the revised version).
4. Gene function validation: To further study the role of *OsGATA8* in NUE, we constructed knock-out lines of Nipponbare by using the CRISPR/Cas system (Fig. 1a in the revised version). The agronomic trait of PTNR was significantly improved in *Nip-cas* as compared to Nip (Item 9 in this response letter, or Fig. 1c in the revised version). These data suggest an important role of *OsGATA8* in regulating NUE.

Item 3: Identification of *OsGATA8* on chromosome 1

(a) Local Manhattan plot (top) and linkage disequilibrium (LD) heatmap (bottom) surrounding the peak on chromosome 1 using P values of PHR and PTNR. PHR, plant height ratio under LN/HN conditions; PTNR, productive-tiller-number ratio under LN/HN conditions. HN, high nitrogen (300 kg/ha net nitrogen); LN, low nitrogen (75 kg/ha net nitrogen).

(b) Heat map of the relative expression of the three candidate genes near the peak in (a) under three nitrogen supply conditions (0-, 1-, and 5-mM NH_4NO_3) based on qRT-PCR of rice seedlings. The color key (blue to red) represents fold changes of gene expression.

(c) The PTNR, PHR, and the relative expression of *OsGATA8* in 117 rice varieties. Values represent mean \pm SD derived from three biological repeats. The dotted lines represent the average values of two haplotypes varieties. In c, P values were calculated with Student's t test

Item 4: Natural allelic variation of the *OsGATA8* promoter.

Structure of *OsGATA8* (top) and association mapping with its promoter variants (bottom), showing that the 245-bp (chr1:13,569,676) deletion is significantly associated with PTNR ($P = 1.78 \times 10^{-5}$). Dots above dashed line indicate variants that are significantly associated with PTNR. Red dots connected with the dashed lines indicate two large indels that are associated with PTNR. x axis, physical position of the *OsGATA8* promoter.

Comment: Fig. 3: the genetic and phenotypic analysis of the effect of *OsGATA8* deletion in crisper-lines has been done with –cr2, whereas the mRNA-seq. and transcriptional regulation of AMTs has been done with another independent line –cr1. Thus, transcriptional results in the –cr1 line may be due to different off-target or side effects and have not been done in a confirmed background. At least, transcriptional results should be repeated in the –cr2 line.

Response: We thank the reviewer for pointing out this inconsistency in sample choice. We have carefully addressed this issue in the revised manuscript by repeating the RNA-seq analysis using WT (Nipponbare) and two independent CRISPR knockout lines of *OsGATA8* (–cr1 and –cr3), which was cultured in IRRI nutrient solution. The newly performed RNA-seq resulted in 619 differentially expressed genes (DEGs), among which 19 genes annotated in the transporter activity process (GO:0005215) in Gene Ontology analysis (Item 5a in this response letter, and Fig. S6c in the revised version). In the transporter activity process (GO:0005215), *OsAMT3.2* again showed up (Item 5b in this response letter, and Fig. S6d in the revised version). Two times RNA-seq results

both confirmed that the expression of *OsAMT3.2* is negatively regulated by *OsGATA8*. The genetic and phenotypic analysis of the effect of *OsGATA8* deletion showed that *OsAMT3.2* was upregulated in both *OsGATA8* CRISPR lines (Fig. 2a, b in the revised version). These results are consistent with what we had described in the original version of the manuscript.

a

b

Item 5:
OsGATA8 negatively regulates nitrogen uptake by transcriptionally repressing *OsAMT3.2*.

(a) Gene Ontology (GO) enrichment analysis of 619 DEGs in *OsGATA8* knockout lines under Nip background.

(b) The relative expression of DEGs in transporter activity process (GO:0005215). The color key (blue to red) represents gene expression (FPKM) as fold changes. The gene-encoding proteins are shown on the left.

Comment: • Fig. 3A: To consolidate results and include a positive control, *OsAMT3.2* transcript levels should be shown also in *pHapH::OsGATA8* lines.

Response: Good suggestion! We quantified the expression level of *OsAMT3.2* in the *pHapH::OsGATA8* line (See below Item 6) and included the results in Fig. S31c in the revised manuscript. Briefly, the expression of *OsAMT3.2* was notably reduced in two independent *pHapH::OsGATA8* lines but enhanced in the *OsGATA8-cr1* line. These results demonstrate a negative correlation between the expression level of *OsGATA8* and that of *OsAMT3.2*, indicating that *OsGATA8* negatively regulates the expression of *OsAMT3.2*.

Item 6: The relative expression of *OsAMT3.2* in *OsGATA8* transgenic plants Different letters indicate significant differences ($P < 0.05$, one-way ANOVA, Duncan's new multiple range test); for P values, see Source Data.

Comment: Fig. 3g-h: Why does the knock-out of *OsAMT3.2* lead to a higher total number of tillers? Does this mean that ammonium uptake suppresses tiller formation in rice?

Response: We appreciate the reviewer's insightful comment. To validate the effect of *OsAMT3.2* on tillering, we generated two independent knockout lines of *OsAMT3.2* (*cr1* and *cr2*). The *cr1* line contained a 22bp deletion in the coding region resulting in premature termination of

translation. The cr2 line contained a 1bp insertion in the coding region resulting in frameshift mutation (Item 7b in this response letter, or Fig. S10d in the revised version). qRT-PCR assay showed a significant decrease in *OsAMT3.2* expression in both the cr1 and cr2 lines compared to wild type (Item 7c in this response letter, or Fig. S10c in the revised version). These results validated the cr1 and cr2 are knockout lines for *OsAMT3.2*.

Although the total number of tillers was not reduced, the number of productive tillers and the proportion of productive tillers (PT%) decreased in the two *OsAMT3.2* knockout lines (Item 7d, e in this response letter, or Fig. S11b, c in the revised version).

Based on these results, we conclude that knocking out *OsAMT3.2* reduces the number of productive tillers and the proportion of productive tillers, while it does not have a consistent effect on the total number of tillers.

Item 7: *OsAMT3.2* positively regulates rice yield traits and NUE.

(a) Phenotypes of the *OsAMT3.2* knockout and overexpression lines under LN and HN conditions at the maximum tillering stage and mature stage.

(b) The mutation in the *OsAMT3.2* CRISPR knockout line in the Nipponbare background. Black boxes: the coding regions of *OsAMT3.2*; White box: the 5' UTR of *OsAMT3.2*; White arrow: the 3' UTR of *OsAMT3.2*; Red line: the mutation site. The guide RNA targeting site and protospacer adjacent motif (PAM) are indicated. A red dash indicates the deletion of one base and inserted bases are marked in red. The arrows indicate mutation types.

(c) Relative expression of *OsAMT3.2* in Nipponbare (Nip), *OsAMT3.2* knockout and overexpression lines. Values represent mean \pm SD derived from three biological repeats.

(d-e) The numbers of total number of tillers, productive tillers, and the proportion of productive tillers (PT%) of the genotypes in (a) under LN and HN conditions. “ Δ ” represents the percentage difference compared with the total number of tillers. $n \geq 20$ plants.

In b, c different letters indicate significant differences ($P < 0.05$, one-way ANOVA, Duncan's new multiple range test); for P values, see Source Data.

Comment: Figs. 3h and 4g: i) The N-dependent regulation of OsGATA8 expression is not clear and should be laid down. ii) If OsGATA8 is upregulated by high N (as to be assumed) and promotes excessive tiller formation via OsTCP19 suppression, why is there so little difference between Nip and OsGATA-cr1 in the number of productive tillers between LN and HN? Actually, this difference should be higher at HN than at LN.

Response: We appreciate these insightful comments. To address the first point, we tested the expression of *OsGATA8* under different nitrogen concentrations (Item 3b, 8a in this response letter, or Fig. S4b in the revised version) and found that the expression of *OsGATA8* is indeed induced by high nitrogen.

To address the second concern, we conducted additional experiments in the revised version as follows. Although there is little difference between Nip and OsGATA8-cr1 in the number of productive tillers between LN and HN, **the proportion of productive tillers (PT%)** is significantly increased in the *OsGATA8* knockout or *OsGATA8-H* plants (Item 9 in this response letter, or Fig. 1 in the revised version). However, this increase of PT% in the *OsGATA8* knockout or *OsGATA8-H* plants under both LN and HN is likely caused by different reasons. The expression of *OsAMT3.2* was repressed by *OsGATA8* in roots under both LN and HN conditions (Item 18d in this response letter, or Fig. S24a in the revised version), and *OsTCP19* was only repressed by *OsGATA8* in tiller buds under HN conditions (Item 8b and 18e in this response letter, or Fig. S24b, c in the revised version). Under low N conditions, the nitrogen uptake plays a more dominant role in determining NUE since sufficient nitrogen supply helps to promote the conversion of tillers into productive ones. Under such conditions, the reduced expression of *OsGATA8-H* or *OsGATA8* knockout relieves its repression on the transcription of the ammonium transporter gene *OsAMT3.2*,

which leads to increased ammonium uptake, allowing increased supply of nitrogen to rice tillers to promote their development into effective panicles, and consequently, increased proportion of productive tillers and increased NUE (Item 12 in this response letter, or Fig. 2 and 5h, i in the revised version). Under high nitrogen conditions, nitrogen supply is not a rate-limiting factor, while the prevention of excessive tillering plays a dominant role in maintaining high NUE, because when too many tillers are formed, their productivity drops, leading to reduced NUE. The reduced expression of *OsGATA8-H* or *OsGATA8* knockout leads to an increased expression of *OsTCP19*, which would prevent excessive tiller formation under high nitrogen conditions, contributing to increasing the proportion of productive tillers (Item 8b, 12b in this response letter, or Fig S24 and Fig. 5i in the revised version). We have included this discussion in the revised manuscript.

Item 8: Expression of *OsGATA8* and *OsTCP19* under different nitrogen concentrations

(a) Expression of *OsGATA8* under different nitrogen concentrations.

(b) Expression of *OsTCP19* in WT and *OsGATA8* knockout line under different nitrogen concentrations.

In **a-b**, different letters indicate significant differences ($P < 0.05$, one-way ANOVA, Duncan's new multiple range test); for P values, see Source Data

Item 9: OsGATA8 negatively regulates PTNR and the proportion of productive tillers in rice.
(a) The phenotypes of Nipponbare (Nip), the *OsGATA8*-cr knockout lines, and the *OsGATA8*

overexpression lines at the maximum tillering stage (about 40 days after transplanting) and the mature stage under LN and HN conditions. These phenotypes indicate that *OsGATA8* negatively regulates PTNR and the proportion of productive tillers in rice.

Scale bars, 20 cm. LN, low nitrogen (75 kg/ha net nitrogen); HN, high nitrogen (300 kg/ha net nitrogen).

(b) Relative expression of *OsGATA8* increased in the *OsGATA8* overexpression lines compared with Nip. Total RNA was extracted from root tissue of 2-week-old seedlings. Data are mean \pm SD ($n = 3$ biologically independent samples).

(c) The PTNR of Nip, the *OsGATA8*-cr knockout lines, and the *OsGATA8* overexpression line at the mature stage under LN and HN conditions. $n \geq 20$ plants. PTNR, productive-tiller-number ratio (Effective panicle number under LN condition / Effective panicle number under HN condition). These results show that *OsGATA8* negatively regulates PTNR.

(d) The total number of tillers and productive tillers of Nip, the *OsGATA8*-cr knockout lines, and the *OsGATA8* overexpression lines under LN and HN conditions. The result shows that *OsGATA8*-cr knockout lines show higher proportion of productive tillers (PT%) compared with Nip and OE lines. “ Δ ” represents the percentage difference compared with the total number of tillers. $n \geq 20$ plants; values represent mean \pm SD.

(e) The proportion of productive tillers (PT%) and the number of non-productive tillers of Nip, the *OsGATA8*-cr knockout lines, and the *OsGATA8* overexpression lines under LN and HN conditions. $n \geq 20$ plants; values represent mean \pm SD. These results show that *OsGATA8* negatively regulates PTNR and positively regulates non-productive tillers.

In **c-e**, the horizontal bars in the bar charts represent the minimum, 25th percentile, median, 75th percentile and maximum values; in **b-d**, different letters indicate significant differences ($P < 0.05$, one-way ANOVA, Duncan’s new multiple range test); for P values, see Source Data.

Comment: Fig. S5-a shows a heatmap of differentially expressed genes involved in the nitrogen metabolism pathway, serving as an exploratory method to subsequently select candidate genes affected by *GATA8*. However, there are further important nitrogen transporters incl. *NRT1.1B* and *AMT2.3* that were not considered. Moreover, *AMT3.2* was even among the least upregulated, and *AMT1.2* does not even appear in the heatmap. This creates a discrepancy for the follow-up experiments. What role do these genes play downstream of *GATA8*?

Response: We thank the reviewer for raising the concern on *OsNRT1.1B* and *OsAMT2.3*. We have revised the manuscript to clarify our line of thoughts involved in the discovery of *OsAMT3.2* as a target of *OsGATA8*.

Although *OsNRT1.1B* showed up-regulation in the DEG analysis, our nitrogen influx assays showed that *OsGATA8* specifically affected the ammonium uptake, but not nitrate uptake. Therefore, we did not further pursue on the nitrate transporter *OsNRT1.1B*.

We subsequently examined ammonium transporter genes to test whether they are the targets of *OsGATA8*. Through a DAP-Seq analysis, we identified the promoters of *OsAMT3.2* and *OsAMT1.2* as being directly recognized by *OsGATA8*, but not that of *OsAMT2.3*. Therefore, we did not further pursue *OsAMT2.3*.

Comment: The main results rely on the formation of productive tillers or panicles as proxy for NUE. This raises the question whether *OsGATA8*-dependent NUE is the same when using total shoot biomass at the end of tillering (maximum tillering stage) under LN versus HN rather than using grain yield at maturity under LN versus HN. The current data imply that *GATA8* may just be a regulator of tillering (which itself is N-dependent), rather than a regulator of NUE. Therefore, it should be verified (in a field study) how *GATA8* affects NUE on the basis of total shoot biomass at the end of tillering.

Response: Thanks for the comments. In the revised manuscript, we measured the total shoot biomass at the end of tillering (maximum tillering stage) under LN and HN, respectively. We have obtained the following new results, which have been included in the revised manuscript.

To elucidate how *OsGATA8* regulates NUE in rice at the maximum tillering stage, we quantified the biomass and NUE in *OsGATA8* NIL line under LN and HN conditions. At maximum tillering and mature stages, biomass, plot yield, and NUE were significantly increased in *Aso^{OsGATA8-H}* line under LN; differently, while at the maximum tillering stage, no differences were observed in biomass and NUE compared with *Aso* under HN conditions (Item 10a, b in this response letter, or Fig. S25b, c in the revised version). We also found that the total number of tillers in *Aso* was significantly higher than that of the *Aso^{OsGATA8-H}* line at maximum tillering stage, however, its biomass and NUE were not superior to that of *Aso^{OsGATA8-H}* (Item 10a, 22b in this response letter, or Fig. S25b and 5b in the revised version). These results suggest that *OsGATA8* negatively regulates rice NUE that is determined by the final yield (Item 10b and 22e in this response letter, or Fig. S25c and 5e in the revised version), which cannot be evaluated using the total number of tillers and biomass at the maximum tillering stage.

Item 10: *OsGATA8* negatively regulates rice NUE.

(a-b) Biomass per plot, yield per plot (10 plants \times 4 rows) and NUE of *OsGATA8* NIL line at maximum tillering stage and mature stage under LN and HN conditions. These results verify that *OsGATA8* negatively regulates rice yield and NUE at the mature stage.

In **a** and **b**, different letters indicate significant differences ($P < 0.05$, one-way ANOVA, Duncan's new multiple range test); for P values, see Source Data.

Comment: A major question is to what extent is *OsGATA8* a NUE determinant or rather a TF determining one of the major yield components, i.e. number of spike-forming tillers?

Response: We thank the comment by reviewer. To clarify whether *OsGATA8* a NUE determinant or regulator of tiller development, we quantified the tiller number of *OsGATA8* near isogenic lines under both LN and HN condition. To clarify to what extent *OsGATA8* is a NUE determinant, we carried out additional experiments during the revision, which is described in detail below.

Firstly, we carried out a field trial showing that the Aso^{*OsGATA8-H*} line not only exhibited higher PTNR, increased proportion of productive tillers, and increased grain yield per plant (Item 22 in this response letter, or Fig. 5b-g in the revised version), but also showed increased yield per plot and NUE at the mature stages (Item 10b, 11 in this response letter, or Fig. 5e in the revised version). These results indicate that *OsGATA8* is a NUE regulator through modulating NUE, rather than only determining the number of spike-forming tillers.

Secondly, we also revealed that *OsGATA8*, as a NUE determinant, negatively regulates nitrogen uptake by repressing transcription of the ammonium transporter gene *OsAMT3.2* (Fig. 2 in the revised version). On the other hand, *OsGATA8* also promotes tiller formation by repressing transcription of *OsTCP19*, a negative modulator of tillering (Fig. 3 in the revised version).

Thirdly, we provided genetic and molecular biological evidence to clarify the molecular mechanisms underlying coordination of these two pathways (Item 12 in this response letter, or Fig. 5h-i in the revised version). The involvement of *OsGATA8* in both nitrogen uptake and tiller formation suggests that its biological function is beyond determining one yield component, but rather serves as a major NUE determinant.

Item 11: *OsGATA8-H* significantly improves rice yield and NUE.

(a) The phenotypes of *OsGATA8* NIL line at the mature stage under LN and HN conditions. LN, low nitrogen (75 kg/ha net nitrogen); HN, high nitrogen (300 kg/ha net nitrogen).

(b) The panicle length and grain number per plant of Aso^{*OsGATA8-H*} and line at the mature stage under LN and HN conditions.

In (a-b), the Aso^{*OsGATA8-H*} line exhibited increased panicle length, grain number per panicle and yield per plot at the mature stage.

In **b**, different letters indicate significant differences ($P < 0.05$, one-way ANOVA, Duncan's new multiple range test); for P values, see Source Data.

Item 12: A proposed model of the role of *OsGATA8* in nitrogen uptake and tiller development.

(a) A proposed model of OsGATA8 regulating rice NUE by balancing nitrogen uptake and tillering growth in rice.

(b) A proposed model of the role of OsGATA8 in nitrogen uptake and tiller development in the haplotypes *OsGATA8-H* under low nitrogen and high nitrogen conditions. Under low nitrogen conditions, *OsGATA8-H* confers high PT% predominantly via enhanced ammonium uptake through upregulated *OsAMT3.2* (highlighted in red). Under high nitrogen conditions, *OsGATA8-H* confers high PT% predominantly via maintaining a moderate number of tillers through upregulated *OsTCP19* (highlighted in red).

Comment: When OsGATA8 promotes excessive tiller formation, N demand by the plant is rising, this should increase ammonium uptake capacity. However, OsGATA8 suppresses *OsAMT3.2*. This apparent contradiction should be resolved.

Response: We thank the reviewer for the comment. As we explained in the above model, OsGATA8 negatively regulate NUE and proportion of productive tiller under both LN and HN conditions. Under low nitrogen conditions, *OsGATA8-H* confers high PT% predominantly via enhanced ammonium uptake through upregulated *OsAMT3.2* (highlighted in red). Under high nitrogen conditions, *OsGATA8-H* confers high PT% predominantly via maintaining a moderate number of tillers through upregulated *OsTCP19* (highlighted in red). (Item 12b in this response letter, or Fig. 5i in the revised version).

Minor

points:

Line 109: the term ‘productive tiller’ is not defined. i) Do the authors mean ear- or spike-forming tillers? ii) At what stage does the formation of an ear or spike make a tiller productive? Please clarify in the text or methods.

Response: As shown in the diagram below, we counted the productive tillers of rice during the mature stage, which referred to panicles with more than 5 full grains. Normally, the nitrogen uptake and utilization during the grain filling stage of rice determine the formation of rice grains and productive tillers (Item 13 in this response letter, or Fig. S1 in the revised version).

1 Productive tiller can generate effective spike with at least five grains, otherwise is termed as non-productive tiller.

2 The maximum tillering stage is normally presented at ~40 day of vegetative growth post transplanting, entering reproductive growth. The total tiller number in this study represents the number of tillers longer than 1cm at the maximum tillering stage⁴⁻⁶.

3 Non-productive tillering is equal to the total number of tillers at the maximum tillering stage minus the number of productive tillers at the mature stage.

Item 13: Overapplication of nitrogen fertilizers reduces the proportion of productive tillers in rice.

(a) An illustration of tiller development in rice. Tillers that successfully produce effective panicles become productive tillers, while the rest end up being non-productive tillers. The proportion of productive tillers (PT%) is defined as the ratio between the number of productive tillers and the total number of tillers. The number of tillers in rice shows dynamic changes with growth and development.

(b) The development of tillers and panicles in rice grown under low nitrogen and high nitrogen conditions. The counts of various types of tillers and the overall PT% are displayed, showing that over application of nitrogen promotes excessive tiller formation, especially non-productive tillers.

(c-d) The tillering dynamic changes of wild type in (b) with rice transplanting days under LN and HN conditions. The proportion of productive tillers (PT%) were calculated as shown in (a). Eight out of eleven tillers, accounting for 72.7%, turned out to be productive tillers under low nitrogen condition, while 14 out of 24 tillers, only 58.3%, under high nitrogen condition due to overapplication of nitrogen fertilizers. Therefore, over application of nitrogen reduces the proportion of productive tillers (PT%) of rice.

• Fig. S2b says tiller node, text (line 102) says tiller bud – clarify!

Response: Experimental data for tiller bud was omitted in the original version and has been added as shown in the Item 14, or Fig. S5b in the revised version.

Item 14: *OsGATA8* is highly expressed in the roots and tiller buds. Relative expression of *OsGATA8* in various rice tissues. Values represent mean \pm SD derived from three biological replications.

Line 96: *OsGATA8* should not be in italics as it refers to the encoded protein.

Response: We have revised and double-checked the entire manuscript for this formatting issue.

Line 188: There is no Fig. 1f. Maybe labelling in the fig. Is lacking.

Response: We have fixed this issue. This figure is now displayed as **Fig. 1d and Fig. S12a in the revised version.**

Line 219-222: These two sentences may be misleading, as it is not the rainfall but the soil microbiome that may lead to ammonium production from nitrate. Re-formulate.

Response: We thank the reviewer for this valuable suggestion. We have removed the misleading description “rainfall increases the availability of soil nitrogen to rice plants”. We have also removed the results of the correlation between the geographical distribution of *OsGATA8-H* with the regional rainfall and retained the results of the correlation with the regional soil nitrogen content.

Line 285: CHL1 is outdated. Use *OsNRT1.1B/NPF6.5* instead.

Response: The name has been replaced.

Line 287: Use *OsNRT1.1A/NPF6.3* instead.

Response: We would like to clarify that the previously reported gene is *OsNPF6.1*, not *OsNRT1.1A/NPF6.3*.

General shortcoming in the data display: Many plots do not show units on y-axes (e.g. is in Fig. S2g the tiller no. per plant or per pot? Is in Fig. S2h proportion in %? All y-axes should be properly assigned with units.

Response: Revised.

Consider using a different graphical representation instead of bar graphs. Generally, boxplots accompanied by P-values are less susceptible to visual distortion, especially when the sample size

is large. For example, it is difficult to discern clear differences between productive tillers in Fig. 1c, 2d, 3h, or 4g, or in yield in Fig. S7e. Box plots would be more appropriate here.

Response: We have replaced the bar chart with data points greater than 10 to a box chart or fish chart.

Legend in Fig. S3A: In the scheme it remains unclear whether the 245 bp sequence is present in P2-L and P2-H, or not.

Response: We appreciate this comment. For improved visualization, we remade the scheme (Item 15 in this response letter, or Fig. S18a in the revised version), and the 245bp sequence did not exist in P2-L (P2-A) and P2-H (P2-B).

Item 15: Schematic representation of the reporter constructs for the luciferase assay. The black fold lines in P-A, P1-A and P2-A each represents a 150-bp deletion relative to the red box. The black fold lines in P-B and P1-B each represents a 245-bp deletion relative to the green box. Scales represent position relative to ATG (0 bp) of *OsGATA8*.

This revised schematic diagram clearly illustrates that the 245bp sequence does not exist in P2-L (P2-A) or P2-H (P2-B).

Reviewer

#2:

Comment: The major conclusion of this work is about the coordination of nitrogen uptake and productive tiller formation by *OsGATA8*. However, it is very difficult to understand how these two processes are coordinated from the presented data in this manuscript, even though the authors showed that *OsGATA8* can regulate both *OsAMT3.2* and *OsTCP19*, the components responsible for ammonium uptake and tillering modulation, respectively. The specified role of *OsGATA8* in determining the transformation of young tiller to productive tiller or non-productive tiller should be investigated to support its proposed function in this work. The authors may consider whether *OsGATA8* is specifically repressed in the young tiller that can be transformed into the productive tiller, in which the nitrogen uptake is greatly enhanced and the following development is thereby promoted.

Response: We would like to thank the reviewer for this stimulating comment. Following his advice, we have conducted additional experiments to support the role of *OsGATA8-H* promoting the transformation of young tillers to productive tillers.

1 We tracked the total number of tillers and the number of productive tillers at different developmental stages in the *OsGATA8* transgenic lines and the NIL with the *OsGATA8-H* haplotype, and found that the development of tillers into productive ones is negatively impacted by the expression of *OsGATA8* (Item 16, 18a in this response letter, or Fig. S12 and S23a in the revised version).

2 We completed our field trails and quantified the expressions of *OsGATA8* from seedling stage to mature stage in NILs as recommended. We found that the expression of *OsGATA8* in rice tiller buds peaks at the maximum tillering stage and the expression of *OsGATA8* was specifically repressed in the young tillers that can be transformed into the productive tillers (Item 16-18 in this response letter, or Fig. 12, 23, 24 in the revised version). These results suggest that *OsGATA8* determines the transformation of young tiller to productive tiller or non-productive tiller.

3 Reducing *OsGATA8* expressions relieve its repression on the transcription of the ammonium transporter gene *OsAMT3.2*, which leads to increased ammonium uptake (Item 12a in this response letter, or Fig. 5h in the revised version). Consistently, we quantified the nitrogen content in the *OsGATA8* NIL line (*Aso^{OsGATA8-H}*) at the seedling, maximum tillering and mature stages, and found that *Aso^{OsGATA8-H}* plants displayed increased nitrogen content (Fig. S25a-c in the revised version). Our phenotypic analysis of *OsAMT3.2* over-expression and knockout lines revealed a positive correlation between ammonium uptake and the proportion of productive tillers (Item 7 in this response letter, or Fig. S11 in the revised version).

4 We investigated the expression dynamics of *OsAMT3.2* and *OsTCP19* in roots and tiller shoots at seven distinct developmental stages after transplanting, respectively. We found that the expression of *OsAMT3.2* was repressed by *OsGATA8* in roots under both LN and HN conditions

(Item 18d, e in this response letter, or Fig. S24a, b in the revised version), and *OsTCP19* was only repressed by *OsGATA8* in tiller shoots under HN conditions (Item 8b, 18d, or Fig. S24b, c in the revised version). In addition, we found that the expression of *OsAMT3.2* and *OsTCP19* was dynamically regulated by *OsGATA8* during rice growth and development (Item 18d, e in this response letter, or Fig. S24a, b in the revised version). Thus, *OsGATA8-H* increased PT% predominantly via upregulating *OsAMT3.2* to uptake ammonium under LN, whereas downregulating *OsTCP19* to reduce non-productive tillers under HN (Item 8b, 12a and 18d, e in this response letter, or Fig. S24 and 5h in the revised version).

These observations **supported the point speculated by the reviewer, i.e., *OsGATA8* is repressed in the young tiller that can be transformed into the productive tiller, in which the nitrogen uptake is greatly enhanced and the following development is thereby promoted.** We have integrated these contents in Results and Discussion in the revised version.

We wish to thank the reviewer's insightful comment again, which is very constructive and helpful!

a

b

c

Item 16: OsGATA8 suppresses productive tiller formation.

(a) The phenotypes of Nip, the *OsGATA8*-cr knockout lines, and *OsGATA8* transgenic lines exhibited tillering phenotypes at maximum tillering stage and effective panicle (productive tillers) phenotypes at mature stage under LN and HN conditions. Overexpression of *OsGATA8* promotes excessive tiller production and suppresses productive tiller formation. Scale bar, 5 cm. LN, low nitrogen (75 kg/ha net nitrogen); HN, high nitrogen (300 kg/ha net nitrogen). Red arrows indicate non-productive tillers; the pictures in the white boxes are partial enlargements of the non-productive tillers.

(b-c) The tillering dynamic changes of *OsGATA8* knockout and transgenic lines in a with rice transplanting days under LN and HN conditions. The result shows that *OsGATA8* promotes excessive tiller production and suppresses productive tiller formation. *P* values were calculated with Student's *t* test (*, $P < 0.05$; **, $P < 0.01$), and compared with Nip.

Item 17: The expression of *OsGATA8* is specifically repressed in the young tillers that can be transformed into the productive tillers. Total RNA was extracted from tiller base of different leaf-age tillers at different days after transplanting.

Item 18: OsGATA8 plays a dual role in regulating rice tiller formation at vegetative growth and reproductive growth.

(a) The tillering dynamic changes of Aso and Aso^{OsGATA8-H} with rice transplanting days under LN and HN conditions. Aso^{OsGATA8-H} shows reduced total number of tillers and improved formation of productive tiller.

(b-e) Dynamic changes of *OsGATA8*, *OsAMT3.2* and *OsTCP19* transcription levels in Aso and Aso^{OsGATA8-H} tiller buds and roots with transplanting days. These results show that OsGATA8 plays a dual role in regulating rice tiller formation during vegetative growth and reproductive growth.

In **b, d**, total RNA was extracted from tiller buds of rice at different days after transplanting; in **c, e**, total RNA was extracted from roots at different days after transplanting. In **a-e**, *P* values were calculated with Student's *t* test (*, *P* < 0.05; **, *P* < 0.01).

Comment: The authors deemed that OsGATA8 regulates the productive tiller proportion via targeting the OsTCP19. This conclusion requires the precondition that OsTCP19 is involved in modulating the productive tiller proportion. Nonetheless, Figure 4 e-g clearly showed that loss-of-function of OsTCP19 (OsTCP19-cr1) results in the increase of both the total tiller and the productive tiller, and the proportion of the productive tiller is unaffected. Thus, the repression of OsTCP19 cannot specifically promote the formation of non-productive tiller. It still lacks the evidence to support the conclusion that “This OsGATA8-OsTCP19 module promotes the formation of non-productive tillers (line 215)”.

Response: We appreciate the reviewer's careful check. We realize that our original expression “OsGATA8 regulates the productive tiller proportion via targeting the *OsTCP19*” was inaccurate and misleading, because the repression of OsTCP19 cannot specifically promote the formation of non-productive tiller.”

What we would like to propose is that OsGATA8 regulates the productive tiller proportion through **the coordinated regulation of *OsAMT3.2* and the *OsTCP19***. The increase in the proportion of productive tillers in the *OsGATA8* knockout or *OsGATA8-H* plants is due to the combinatory effect of elevated expression of *OsAMT3.2* (increase N supply during tiller development to promote effective panicle formation) and *OsTCP19* (prevents excessive tiller formation).

This has been tested and validated through phenotyping the proportion of productive tillers in the *OsGATA8* knockout or *OsGATA8-H* plants, which is consistent with our model that under LN, the effect of *OsGATA8-H* on PT% is achieved predominantly via upregulated *OsAMT3.2* expression, whereas under HN, this effect is achieved predominantly through elevated *OsTCP19* expression (**Item 8b, 12b in this response letter, or Fig S24 and Fig. 5i in the revised version**).

We added this detailed discussion to the last section of the revised manuscript.

a

b

c

d

e

Item 19: OsGATA8 promotes tillers by transcriptionally repressing *OsTCP19*.

(a) Phenotypes of Nipponbare (Nip), the *OsGATA8* knockout mutant, the *OsTCP19* knockout mutant, and the *OsGATA8/OsTCP19* double knockout mutant at the maximum tillering stage (about one month after transplanting) and the mature stage under LN and HN conditions.

(b) PTNR of the genotypes in (a) under LN and HN conditions. $n \geq 20$ plants; the bars within the violin plots represent the 25th percentile, the median, and the 75th percentile, respectively.

(c-d) The total number of tillers, productive tillers and proportion of productive tillers (PT%) of the genotypes in (a) under LN and HN conditions. “ Δ ” represents the percentage difference compared with the total number of tillers. $n \geq 20$ plants; values represent mean \pm SD.

(e) The number of non-productive tillers for genotypes in the revised version of Fig. 3e under LN and HN conditions.

In b-e, different letters indicate significant differences ($P < 0.05$, one-way ANOVA, Duncan’s new multiple range test).

These results show *OsGATA8* promotes the formation of non-productive tillers through (at least partly) directly repressing the expression of *OsTCP19*.

Comment: The authors proved that 245-indel in the promoter of *OsGATA8* is the causal variation responsible for EPNR variation in different rice varieties. To support this conclusion, the authors provide a series of evidence including association assay, promoter activity analysis, different haplotype transgenic assay, and genome-editing. Even though, to make this finding solid, it still needs more evidence:

1) There are 21 indels and 108 SNPs in the promoter region, the authors only select two large indels as the candidate. It needs more evidence to exclude other variations.

Response: We thank the reviewer for expressing this concern. We carried out the following experiments during the revision.

1. We re-sequenced the promoter of *OsGATA8* and conducted an association analysis with the variants we identified. Association analysis revealed that the 245-bp (chr1:13,569,676) deletion is significantly associated with PTNR (the same as EPNR in previous manuscript version) ($P = 1.78 \times 10^{-5}$) (Item 4 in this response letter, or Fig. S17 in the revised version).

2. We performed luciferase assays using different segmented versions of the *OsGATA8* promoter as follows, 1) P-A and P-B: full length promoter, P1-A and P1B: 1,150-bp, P2-A and P2B: 700-bp, P3-A and P3-B: 664-bp. Our results confirmed that only the 245bp deletion (888-bp upstream of start codon) could affect the activity of the *OsGATA8* promoter (Item 20 in this response letter, or Fig. S18 in the revised version).

3. Through luciferase assays and transgenic validation of the 403-bp region in the D245 line, we found that the 245-bp flanking sequence did not significantly affect the promoter activity and transcriptional level of *OsGATA8* (Item 20, 21 in this response letter, or Fig. S18, 19 in the revised version).

Therefore, we clarified the 245bp deletion was the causal variants for *OsGATA8* function.

Comment: The genome-editing in the promoter region would provide very solid evidence to support the vital role of the 245 bp-indel in controlling the expression of *OsGATA8*. However, the D245 genome-editing line carries over 400 bp deletion in the promoter. It is hard to say that the reduced expression of *OsGATA8* is resulted from the 245 bp-indel or other deletion fragment.

Response: This is closely related to the previous comment. We performed luciferase assays using segmented versions of the *OsGATA8* promoter and used CRISPR to delineate the functional region within the 403-bp fragment in stable genome-edited lines. We found that the flanking sequences (115bp upstream and 43bp downstream) of this 245-bp deletion did not significantly affect the promoter activity and transcriptional level of *OsGATA8* (Item 20, 21, or Fig. S18, 19 in the revised version). Therefore, the reduced expression of *OsGATA8* is resulted from the 245 bp-indel.

Item 20: A 245-bp deletion in the promoter of *OsGATA8* reduces the expression of the gene.

(a) Schematic representation of the reporter constructs for the luciferase assay. The black fold lines in P-A, P1-A and P2-A represent a 150-bp deletion relative to the red box. The black fold lines in P-B and P1-B represent a 245-bp deletion relative to the green box. The black fold lines in P-A-D43, P-A-D115 and P-A-D403 represent deletions relative to the red dotted box.

(b) Luciferase assay of the promoter activities of P-A, P-B, P1-A, P1-B, P2-A, P2-B, P3-A, P3-B, P-A-D115, P-A-D43, and P-A-D403 using rice protoplasts, showing only the 245bp deletion could affect the activity of the *OsGATA8* promoter. The blue columns represent promoter activity equal to that of P-A. The yellow columns represent promoter activity equal to that of P-B. Values represent mean \pm SD derived from three biological repeats.

(c) Relative expression of four homozygous lines, showing the *OsGATA8* allele with the 245-bp deletion in its promoter is a hypomorphic allele with reduced expression.

Values represent mean \pm SD derived from three biological repeats.

In **b, c**, different letters indicate significant differences ($P < 0.05$, one-way ANOVA, Duncan's new multiple range test); for P values, see Source Data.

a

b

Item 21: Alignment of the sequences near the 245-bp variant region within the *OsGATA8* promoter in the four homozygous deletion lines.

(a) Schematic representation of four homozygous lines near the 245-bp variant region of the *OsGATA8* promoter. The green block represents the physical location of the 245-bp sequence.

(b) Alignment of the sequences near the 245-bp variant region within the *OsGATA8* promoter in the four homozygous deletion lines. The 245-bp sequence is indicated with the green box.

Comment: The authors created the pHapL- and pHapH-*OsGATA8* transgenic plants, and pHapL-*OsGATA8* line displayed more non-productive tiller formation. This result strongly supports the functional variation occurring in the promoter. However, it would be necessary to examine more independent transgenic lines of pHapL- and pHapH-*OsGATA8*. Only using one transgenic line cannot give a solid conclusion. Unfortunately, this situation occurs through the whole manuscript (the over expression transgenic assays of *OsGATA8* and *AMT3.2* only have one line).

Response: We appreciate this suggestion. We added all the transgenic lines as requested, at least two KO lines and OE lines. Consistent results have been obtained, and they have been integrated in the revised version of the manuscript.

Comment: It is necessary to present the data of the total tiller, productive tiller and the proportion of the productive tiller of NIL^{*OsGATA8-H*}. These results will further support the function of *OsGATA8-H*.

Response: We thank the reviewer for this good suggestion and conducted the field trials in 2023, and the newly obtained data of the total tiller, productive tiller and the proportion of the productive tiller of Aso^{*OsGATA8-H*} (NIL^{*OsGATA8-H*}) was shown (Item 22 in this response letter, or Fig. 5a-g in the revised version).

The results showed that Aso^{*OsGATA8-H*} line not only exhibited higher PTNR, more productive tillers, increased proportion of productive tillers, and increased grain yield per plant (Item 11 and 22 in this response letter, or Fig. 5b-f in the revised version), but also showed increased yield per plot (10 plants × 4 rows) and NUE mature stages (Item 22e in this response letter, or Fig. 5e in the revised version). The results provided strong evidences to support that *OsGATA8* negatively regulated yield and NUE.

Item 22: Elite haplotype of *OsGATA8* confers higher nitrogen use efficiency.

(a) Phenotypes of Asominori (Aso) and Aso^{OsGATA8-H} under LN and HN conditions at the maximum tillering and mature stages. LN, low nitrogen (75 kg/ha net nitrogen); HN, high nitrogen (300 kg/ha net nitrogen). Scale bars, 20 cm.

(b) The number of total number of tillers and productive tillers of the genotypes in **(a)** under LN and HN conditions. “Δ” represents the percentage difference compared with the total number of tillers. $n \geq 20$ plants; values represent mean \pm SD. The Aso^{OsGATA8-H} line exhibits increased proportion of productive tillers (PT%).

(c-d) Yield per plant of Asominori (Aso) and Aso^{OsGATA8-H} under LN and HN conditions. Scale bars, 10 cm. The Aso^{OsGATA8-H} line exhibits increased yield per plot and NUE.

(e) The yield per plot (10 plants \times 4 rows) and NUE of Aso and Aso^{OsGATA8-H} under LN and HN conditions. Nitrogen use efficiency (NUE = Dry shoot biomass or grain yield / Amount of N fertilizer). The Aso^{OsGATA8-H} line exhibits increased grain yield per plant.

(f-g) PTNR and proportion of productive tillers (PT%) of the genotypes in **(a)**. $n \geq 20$ plants. The Aso^{OsGATA8-H} line exhibits higher PTNR and increased proportion of productive tillers (PT%).

In **b, d, e, g**, the horizontal bars in the bar charts represent the minimum, 25th percentile, median, 75th percentile and maximum values; different letters indicate significant differences ($P < 0.05$, one-way ANOVA, Duncan’s new multiple range test); in **f**, the bars in the violin plots represent 25th percentiles, medians and 75th percentiles; P value was calculated with Student’s t test; for P values, see Source Data.

4. The authors find that *OsGATA8-H* is subjected to the artificial selection as the increased use of nitrogen fertilizer. But, I cannot find any explanation or discussion about the reason of this phenomenon in this manuscript. As the authors proposed, the increased application of nitrogen would result in the dramatic increase of the non-productive tiller formation, thus the artificial selection toward *OsGATA8-H* will decrease the non-productive tiller formation under high nitrogen. Examination of the proportion of productive tiller using the *OsGATA8-H/L* varieties under low- and high nitrogen will be helpful to answer this question.

Response: Thanks for this good suggestion. As suggested, we have conducted field trails in year 2023 to examine the proportion of productive tiller and yield using the *OsGATA8-H/L* varieties under LN and HN conditions (Item 23 and 24 in this response letter, or Fig. S28 and S29 in the revised version). These results revealed that the increased application of nitrogen resulted in the dramatic increase of the non-productive tiller formation in *OsGATA8-L* varieties, and decrease in *OsGATA8-H*. Thus, *OsGATA8-H* might have been under selection for adaptation to fertile soil with high N contents. We added the explanation and discussion on the reason of this phenomenon in the revised version.

Item 23: Phenotypes of *OsGATA8*-H/L varieties under LN and HN conditions at the maximum tillering stage and mature stage.

OsGATA8-H/L varieties exhibit improved formation of productive tillers under both LN and HN conditions.

Scale bars, 20 cm. LN, low nitrogen (75 kg/ha net nitrogen); HN, high nitrogen (300 kg/ha net nitrogen).

Item 24: Elite haplotype of *OsGATA8* confers higher proportion of productive tillers and yield.

(a-b) The proportion of productive tillers and PTNR of *OsGATA8*-H/L varieties under LN and HN conditions, showing that *OsGATA8*-H varieties exhibited increased proportion of productive tillers and PTNR under LN and HN conditions. The dotted lines represent the average values of two haplotypes varieties. LN, low nitrogen (75 kg/ha net nitrogen); HN, high nitrogen (300 kg/ha net nitrogen); PT%, proportion of productive tillers; PTNR, productive-tiller-number ratio (Effective panicle number under LN condition / Effective panicle number under HN condition).

(c) Yield per plant of 175 *OsGATA8*-H/L varieties from Rice 3k, showing that *OsGATA8*-H varieties exhibited increased NUE under LN and HN conditions.

In a-b, *P* values were calculated with Student's *t* test; in c, different letters indicate significant differences ($P < 0.05$, one-way ANOVA, Duncan's new multiple range test); for *P* values, see Source Data.

5. Some minor points:
 1) The authors should give more clear statement about how to select the three-candidate gene based on the GWAS signal.

Response: We described how to select the three-candidate gene based on the GWAS signal in the revised version as follows.

We proposed the following four filter steps of candidate genes prioritization to narrow down the candidate genes list, and then followed up by functional validations as described in our previously published paper³ and cited it in the Materials and Methods part in the revised version. Firstly, SNVs meeting nominal threshold, p value $< 1 \times 10^{-4}$, at least two times in the successive three years, will be regarded as TAS (Trait-Associated SNVs). Secondly, we will filter loci in which there has no gene in the LD block through gene annotation and LD analysis. Thirdly, we will filter some loci based on reference query, related public rice database, pathway and network. Lastly, we performed real time PCR (RT-PCR) for multiple potential candidate genes within a LD block then obtain candidate statistically associated gene for follow-up functional validation through complementation tests.

2) The method for total tiller number investigation needs to be stated clearly. All the plant materials have the same growth periods? If not so, how to keep the consistency for each plant material during collecting the total tiller number?

Response: We described these methods in the revised version as follows.

All the transgenic materials were constructed and compared in the same genetic background with the identical growth periods. We planted all the materials under the identical conditions and observed the total tiller number at seven-day intervals and the maximum tiller number at 40 days post transplanting (the total number of tillers reached the maximum value). We recorded the productive tiller at the mature stage and calculate non-productive tiller number using the formula: non-productive tiller number = total tiller number - productive number (Item 13 in this response letter, or Fig. S1 in the revised version).

3) The difference of *OsAMT3.2* expression in Aso and NIL^{OsGATA8-H} is very slight in the shoot (Supplementary Figure 14 c). The authors should examine *OsAMT3.2* expression in the root.

Response: We examined the expression of *OsAMT3.2* in the roots of NIL line as requested. The difference of *OsAMT3.2* expression in Aso and Aso^{OsGATA8-H} is significant in the root (Item 25 in this response letter, or Fig. S31a, b in the revised version).

Item 25: Elite haplotype *OsGATA8-H* promotes *OsAMT3.2* expression in roots.

(a-b) Relative expression of *OsGATA8* and *OsAMT3.2* in Aso and Aso^{*OsGATA8-H*} under LN and HN conditions, showing that elite haplotype *OsGATA8-H* promotes *OsAMT3.2* expression in roots. Values represent mean \pm SD derived from three biological repeats. HN, high nitrogen (2mM NH₄NO₃); LN, low nitrogen (0.2mM NH₄NO₃). In **a-b**, different letters indicate significant differences ($P < 0.05$, one-way ANOVA, Duncan's new multiple range test).

4) The deletion fragment in D245 is 403-bp (line 125) or 430-bp (line 491)?

Response: The deletion fragment in D245 line is 403-bp. We apologize for this mistake in the original version. We have gone through the manuscript to make sure that no additional typos alike are present.

5) Both the figure legend and Metatrails and Methods should be supplemented with more detail, and some mistakes should be avoided. For example, the information about the NILs (the precise size of the introduction fragment) should be provided; the method for “¹⁵N accumulation assay” (lines 733-735) showed that “Then, the seedlings were pretreated with 2 mM (NH₄)₂SO₄ and 2 mM KNO₃ for 1 week and transferred to the solutions with 0.2 mM and 2 mM (15NH₄)₂SO₄ and 15KNO₃ for 10 min”, but I cannot find any data related to the ¹⁵N-nitrate uptake assay.

Response: We double checked the figure legend and Materials and Methods to supplement with more detail to avoid mistakes.

The information about the NILs (the precise size of the introduction fragment) has been provided in **Item 26a-c**, or **Fig. S22a-c** in the revised version.

We previously presented the result of $(^{15}\text{NH}_4)_2\text{SO}_4$ in ^{15}N -nitrate uptake in the original version. The result of $^{15}\text{KNO}_3$ has also been added in Item 26d, or Fig. S6b in the revised version.

Item 26: Schematic diagram of NIL lines in this article.

- (a) Schematic diagram of Aso^{*OsGATA8-H*} under Aso background.
- (b-c) Schematic diagram of 9311^{*OsGATA8-H*} and 9311^{*OsGATA8-H/OsTCP19-H*} under 9311 backgrounds.
- (d) ¹⁵N-nitrate uptake assay of Nip and *OsGATA8-cr1* line.

In d, *P* values were calculated with Student's *t* test.

Reviewer**#3:**

Comments: When I read this manuscript, I like this work and appreciated it very much. However I read it repeatedly, I realized a flaw on this work due to ambiguous definition of NUE and EPNR. EPNR is a kind of new word in this field. The biological causes to affect EPNR can be various. But we do not know anything of the target of *OsTCP19* yet.

Response: We wish to thank the reviewer for this question on the definition of NUE and PTNR (the same as EPNR in previous manuscript version).

The number of productive tillers has a major influence on yield, and tiller number has been applied as a key parameter in NUE-related studies in rice and led to the identification of several key regulators of NUE and yield in rice. For instance, Wu et al. identified *NGR5* (*nitrogen-mediated tiller growth response 5*) as a key regulator of nitrogen use efficiency and rice yield through a genetic screening for mutants defective on tillering response to nitrogen supply². Liu et al. performed GWAS on tiller (panicles per plant) response to nitrogen (TRN) and identified *OsTCP19* as a key regulator of NUE and yield¹; in the same study, the authors also provided evidence that NUE has the highest correlation with productive tiller number. These results suggest that the number of productive tillers has a major influence on yield, and tiller number has been applied as a key parameter in NUE-related studies in rice.

To substantiate PTNR could be used as a key criterion of NUE for GWAS study, we performed correlation analysis of NUE with different agronomy traits, including productive tiller number, plant height, heading date, 1,000-grain weight, grain length and grain width between high N and low N supply (Item 1 in this response letter, or Figure S2 in the revised version). Consistent with previous studies, our results also suggest that NUE is most closely correlated with the ratio of productive tiller number (PTNR) and plant height (PHR) (Item 2 in this response letter, or Figure S3 in the revised version). These results suggest that PTNR is an effective indicator of NUE, thus we performed PTNR-based GWAS study to identify key regulators of NUE.

We collected solid genetic and molecular evidence to show that *OsGATA8* negatively regulates nitrogen uptake by repressing transcription of the ammonium transporter gene *OsAMT3.2*. Meanwhile, it promotes tiller formation by repressing transcription of *OsTCP19*, a

negative modulator of tillering¹. Moreover, we identify *OsGATA8-H* as an elite haplotype with reduced expression, which confers enhanced nitrogen uptake, an increased proportion of productive tillers, and higher NUE under both high and low N conditions (Item 12a in this response letter, or Fig. 5h in the revised version). Under low N conditions, the reduced expression of *OsGATA8-H* relieves its repression on the transcription of the ammonium transporter gene *OsAMT3.2*, which leads to increased ammonium uptake, allowing increased supply of nitrogen to rice tillers to promote their development into effective panicles, and consequently, increased proportion of productive tillers and increased NUE (Fig. 2 in the revised version). In addition, the reduced expression of *OsGATA8-H* also leads to an increased expression of *OsTCP19*, which would prevent excessive tiller formation under high nitrogen conditions, contributing to increasing the proportion of productive tillers (Fig. 3 in the revised version). Under high nitrogen conditions, the excessive tillering growth failed to compete with the main tillers for nutrients due to insufficient nitrogen supply for photo-assimilated products⁷. When the expression of *OsGATA8-H* was induced, it promotes the expression of *OsAMT3.2* and *OsTCP19*, resulting in the productive tillers with sufficient photo-assimilated products, thus increasing the proportion of productive tillers and increased NUE. Therefore, we speculated a wide utility of *OsGATA8-H* in promoting NUE and yield in rice under a broad range of nitrogen conditions based on its dual role in the transcriptional regulation of *OsAMT3.2* and *OsTCP19* (Item 12b in this response letter, or Fig. 5i in the revised version). Further, we demonstrate that NUE and yield can be improved under both HN and LN conditions by combining *OsGATA8-H* and *OsTCP19-H*, the elite haplotype of *OsGATA8* and *OsTCP19*, respectively (Fig. S32e in the revised version). Our results not only greatly deepen our understanding of the molecular mechanisms regulating NUE, but also demonstrate an effective strategy to boost NUE and yield in rice (likely applicable to other crops as well). Thus, we expect that this work should generate immediate excitement in the fields of plant biology, agronomy, genetics and crop breeding.

Comments: In addition, we know that NUE can be affected by subjective cultivation environments. For example, an elite cultivar in temperate climates cannot be the one in tropical climates since it often flower too early due to its photoperiodic flowering. It indicate that the same cultivars give us very different NUE due to the timing of floral transition under subjected cultivation environment. Apparently the regulation of tiller number controls such as by SL biosynthesis and signaling are largely independent of the photoperiodic flowering, thus, the timing of floral transition can affect NUE or EPNR largely. This also tell us that to consider NUE, we really care about the genetic background. However, in this paper, they used several cultivars such as NIP and 9311 without caring about this point. These indicate that the data described here can be drastically changed greatly when they change the transplanting date in the same field or cultivation areas. It may happen when they changes the photoperiod or ambient temperature too.

Response: We appreciate this concern on the environmental effects on NUE. We agree with the reviewer that NUE can be affected by various environmental conditions, for instance, the flower time and temperature. A recent study has also revealed that high nitrogen supply can delay the flowering time, which involves in some key clock genes (LHY1/Nhd1) in rice⁸. Numerous studies in *Arabidopsis* have demonstrated that light regime has a major influence on yield and NUE in plants⁹. It is therefore crucial to exclude the influence of the timing of floral transition on NUE. Indeed, we had taken this into consideration when selecting the population for GWAS. As a consequence, all the 117 rice cultivars could grow productive tillers with normal grains for harvest. The flowering time of this population was not significantly affected by nitrogen (Item1 in this response letter, or Fig. S2 in the revised version). This allowed us to germinate, transplant, and cultivate the 117 cultivars simultaneously in the same fields (HN or LN), which had minimized the effect from fluctuation in environmental conditions on NUE. We clarified these considerations in experimental design in the revised manuscript as follows.

We have attempted to avoid the side-effect caused by varied flowering time among rice landraces by using a core rice population, which consists of in total 117 *indica* and *jaпонica* rice landraces showing extreme NUE-related phenotypes. The NUE-associated traits of these landraces in the High N and Low N input field were quantified for three consecutive years¹⁰. The obtained results are a steady and repeatable, allowing us to identify regulators of NUE.

This approach has been commonly applied in GWAS in rice. *OsTCP19* was also identified through the GWAS approach that confers NUE using a core population consisting of 110 *indica*, *jaпонica* and *aus* rice landraces for GWAS¹, highlighting the influence of nitrogen-responsive tiller formation on NUE.

We identified the *OsGATA8*-activated modules from a forward genetic approach using GWAS with three-year field test with the evaluations of a series of parameters associated with NUE and yield. GWAS based on large-scale experiments under field conditions are more realistic and more competent for identifying genes for the NUE improvement in NUE.

Minor comments,

Identification of target genes of *OsGATA8* by DAP-SEQ data is not easy to understand (Fig. S10). The authors can present how *OsGATA8* can bind the promoter regions of downstream target genes by RNA-SEQ data.

Response: We appreciate this suggestion. DNA affinity purification sequencing (DAP-seq) identifies transcription factor binding sites (TFBS) by expressing transcription factors in vitro, independent of antibodies and species, and has been widely used in transcriptional regulation and epigenomics studies since its introduction^{11,12}. We added the binding elements of *OsGATA8* from DAP-seq and presented in Figure S14. We have described the steps of this experiment in detail in

Materials and Methods. We combined the results of Chip-qPCR and RT-qPCR to further validate the regulation of target genes by OsGATA8.

OsTCP19 target genes should be analyzed by stage specific- and tissue specific- RNA-SEQ analysis to reveal the genuine biological meaning of EPNR or NUE.

Response: In this study, we first identified transcription factor OsGATA8 as a crucial regulator of productive tiller response to nitrogen. Our study then emphasized on the exploration of target genes of OsGATA8, and *OsTCP19* was one target gene of OsGATA8 to regulate tiller response to N. We agree with the review to perform stage specific- and tissue specific- RNA-SEQ analysis for downstream components.

We appreciate this insightful suggestion. *OsTCP19* and its target genes have been analyzed by stage specific- and tissue specific- RNA-SEQ analysis to reveal the genuine biological meaning of PTNR (the same as EPNR in previous manuscript version) or NUE¹. In this study, we measured the expression of *DLT*, which is a known direct target of the transcriptional regulation by *OsTCP19* at the tiller buds at the maximum tillering stage using RT-qPCR assays¹ (Item 27 in this response letter). Our results indicate that the expression level of *DLT* is positively regulated in the *OsGATA8* overexpression lines, which is consistent with our model that *OsTCP19* participates in the regulation of PTNR and NUE by OsGATA8.

Item 27: *OsGATA8* positively regulates the expression level of *DLT*

(a) The relative expression of *DLT* in WT, *OsGATA8-cr1* and *pOsGATA8::OsGATA8-1* overexpression plants. Values represent mean \pm SD derived from three biological repeats. Different letters indicate significant differences ($P < 0.05$, one-way ANOVA, Duncan's new multiple range test). Total RNA was extracted from tiller buds at the maximum tillering stage.

(b) The relative expression of *DLT* in WT and *OsGATA8-cr1* line under different nitrogen concentrations. Total RNA was extracted from shoot base of rice seedling.

Can their EPNR be related to fertility rate and/or panicle size and grain size? The readers would like to know it. If no such data involved in specific biological events, this paper is not so biological although it is really agronomical.

Response: The reviewer's question on the biological relevance of PTNR (the same as EPNR in previous manuscript version) is insightful and important. We agree that such information would be informative to a broader readership. All the 117 rice cultivars could grow productive tillers with normal grains for harvest in Nanjing. The flowering time, fertility rate, panicle size and grain size of this population were not significantly affected by nitrogen, which had minimized the effect from fluctuation in these traits on NUE. We clarified these considerations in experimental design in the revised manuscript.

During the revision, we assess the panicle size and grain number per panicle of plant with various genotypes of *OsGATA8* (Item 11b in this response letter, or Fig. S30b). From the plants we investigated, PTNR exhibits a positive correlation with panicle size and grain number per panicle, which indicates that the ability of the rice plant to adjust the number of effective panicles according to nitrogen availability may be linked to its ability to effectively produce grains.

We propose that enhanced ammonium uptake increases the nitrogen supply to rice tillers, which promotes their ability to sustain the formation of an effective panicle. The expression of *OsGATA8* is induced by HN, which negatively regulates nitrogen uptake and positively regulates rice tillering. Expression of *OsGATA8* gradually increases during the period of vegetative growth and reaches a maximum at the maximum tillering stage (Item 18b, c in this response letter, or Fig. S23b, c in the revised version). The expression of *OsAMT3.2* and *OsTCP19* is dynamically regulated by *OsGATA8* in roots and tiller buds, respectively (Item 18d, e in this response letter, or Fig. S24a, b in the revised version). Under high nitrogen conditions, *OsGATA8* is highly expressed, which suppresses the expression of *OsAMT3.2* and *OsTCP19*, resulting in the lack of nitrogen in the non-productive tillers with insufficient photo-assimilated products⁷. Our discovery provides biological insights into how NUE is regulated by establishing an intrinsic link between nitrogen uptake and the development of productive tillers.

Reference:

1. Liu, Y. *et al.* Genomic basis of geographical adaptation to soil nitrogen in rice. *Nature* **590**, 600-605 (2021).
2. Wu, K. *et al.* Enhanced sustainable green revolution yield via nitrogen-responsive chromatin modulation in rice. *Science* **367**(2020).
3. Yu, J. *et al.* Genome-wide association studies identify OsWRKY53 as a key regulator of salt tolerance in rice. *Nat Commun* **14**, 3550 (2023).
4. Naokuni Endo-Higashi & Izawa, T. Flowering time genes Heading date 1 and Early heading date 1 together control panicle development in rice. *Plant Cell Physiol* **52**, 1083-94 (2011).
5. Hotton, S.K. *et al.* Phenotypic Examination of *Camelina sativa* (L.) Crantz Accessions from the USDA-

- ARS National Genetics Resource Program. *Plants (Basel)* **9**(2020).
6. Tang, J. *et al.* Effect of the Applied Fertilization Method under Full Straw Return on the Growth of Mechanically Transplanted Rice. *Plants* **9**, 399 (2020).
 7. Kebrom, T.H. *et al.* Inhibition of Tiller Bud Outgrowth in the tin Mutant of Wheat Is Associated with Precocious Internode Development. *Plant Physiology* **160**, 308-318 (2012).
 8. Zhang, S. *et al.* Nitrogen Mediates Flowering Time and Nitrogen Use Efficiency via Floral Regulators in Rice. *Curr Biol* **31**, 671–683 (2020).
 9. Lin, Y.L. & Tsay, Y.F. Influence of differing nitrate and nitrogen availability on flowering control in Arabidopsis. *J Exp Bot* **68**, 2603-2609 (2017).
 10. Tang, W. *et al.* Genome-wide associated study identifies NAC42-activated nitrate transporter conferring high nitrogen use efficiency in rice. *Nat Commun* **10**, 5279 (2019).
 11. Taiji Kawakatsu *et al.* Epigenomic Diversity in a Global Collection of Arabidopsis thaliana Accessions. *Cell*, 492-505 (2016).
 12. Bartlett, A. *et al.* Mapping genome-wide transcription-factor binding sites using DAP-seq. *Nature Protocols* **12**, 1659-1672 (2017).

Decision Letter, first revision:

9th Apr 2024

Dear Dr. Wan,

Thank you for submitting your revised manuscript "The elite haplotype OsGATA8-H confers yield increase by coordinating nitrogen uptake and productive tiller formation in rice" (NG-A62867R). It has now been seen by the original referees and their comments are below. The reviewers find that the paper has improved in revision, and therefore we'll be happy in principle to publish it in Nature Genetics, pending minor revisions to satisfy the referees' final requests and to comply with our editorial and formatting guidelines.

Sincerely,
Wei

Wei Li, PhD
Senior Editor
Nature Genetics

New York, NY 10004, USA
www.nature.com/ng

Reviewer #1 (Remarks to the Author):

The authors have conducted several additional experiments and could address all my points of concern in a satisfactory manner. I do not have further open questions, as I think the data and their interpretation are now sound.

A final problem remains that several plots do not show units on the y-axis and that the measured or calculated trait is not clearly described in the corresponding text of the legend.

- Fig. 1c, 2h: The y-axis legend shows PTNR in %, whereas the legend says "PTNR, productive tiller number ratio". This is in contradiction, because a ratio is not expressed in % (% = percentage, not ratio). This contradiction also refers to other plots like Fig. 3f, 4 f etc. This requires correction and uniformity in trait characterization throughout the manuscript.
- Supp. Fig. 16e etc.: It is unclear whether the right plot in (e) shows number or %? The legend is only refers to the left plot. If the unit is simply number per plant, please say "(plant⁻¹)", thereby write "-1" in superscript.
- Fig. 5h: In the model please replace "promoting productive tillers" just by the trait "Productive tiller number". This provides more consistency.

Reviewer #2 (Remarks to the Author):

The revised manuscript has been substantially improved, and most of my concerns were also addressed properly. I have no further questions about this manuscript, and totally support its publication.

Reviewer #3 (Remarks to the Author):

According to the authors' responses to the editor's and reviewers' comments, I agree that this revised manuscript have been enough improved.

Author Rebuttal, first revision:

In the second round of revision, we have addressed the remaining concerns by reviewer #1. In addition, we have also carefully checked the manuscript and made adjustments to the formats to meet the editorial requirements.

Detailed, point-by-point responses to comments

Reviewer #1 (Remarks to the Author):

The authors have conducted several additional experiments and could address all my points of concern in a satisfactory manner. I do not have further open questions, as I think the data and their interpretation are now sound.

A final problem remains that several plots do not show units on the y-axis and that the measured or calculated trait is not clearly described in the corresponding text of the legend.

- Fig. 1c, 2h: The y-axis legend shows PTNR in %, whereas the legend says “PTNR, productive tiller number ratio”. This is in contradiction, because a ratio is not expressed in % (% = percentage, not ratio). This contradiction also refers to other plots like Fig. 3f, 4 f etc. This requires correction and uniformity in trait characterization throughout the manuscript.

Response: We appreciate the suggestions. We have provided the missing labels for all axes where applicable in the plots and the description of the pertinent traits in the corresponding legends.

We have also corrected the y-axis of all PTNR and have standardized the representation of PTNRs as the ratios instead of the percentage (%) throughout the manuscript.

- Supp. Fig. 16e etc.: It is unclear whether the right plot in (e) shows number or %? The legend only refers to the left plot. If the unit is simply number per plant, please say “(plant⁻¹)”, thereby write “-1” in superscript.

Response: We appreciate Reviewer #1’s concern. We clarified the information by revising the label of each axis. We use the label “Productive tillers per plant” to avoid confusion.

- Fig. 5h: In the model please replace “promoting productive tillers” just by the trait “Productive tiller number”. This provides more consistency.

Response: We appreciate this important comment, which helps improving the precision of the claim. It is the number of the productive tillers that is concerned in the current study. We have thus replaced “promoting productive tillers” by “Productive tiller number” as kindly suggested.

Reviewer #2 (Remarks to the Author):

The revised manuscript has been substantially improved, and most of my concerns were also addressed properly. I have no further questions about this manuscript, and totally support its publication.

Reviewer #3 (Remarks to the Author):

According to the authors' responses to the editor's and reviewers' comments, I agree that this revised manuscript have been enough improved.

Final Decision Letter:

9th May 2024

Dear Dr. Wan,

I am delighted to say that your manuscript "The elite haplotype OsGATA8-H coordinates nitrogen uptake and productive tiller formation in rice" has been accepted for publication in an upcoming issue of Nature Genetics.

Your paper will be published online after we receive your corrections and will appear in print in the next available issue. You can find out your date of online publication by contacting the Nature Press Office (press@nature.com) after sending your e-proof corrections.

Please note that *Nature Genetics* is a Transformative Journal (TJ). Authors may publish their research with us through the traditional subscription access route or make their paper immediately open access through payment of an article-processing charge (APC). Authors will not be required to make a final decision about access to their article until it has been accepted. Find out more about Transformative Journals

Authors may need to take specific actions to achieve compliance with funder and institutional open access mandates. If your research is supported by a funder that requires immediate open access (e.g. according to Plan S principles) then you should select the gold OA route, and we will direct you to the compliant route where possible. For authors selecting the subscription publication route, the journal's standard licensing terms will need to be accepted, including [a href="https://www.nature.com/nature-portfolio/editorial-policies/self-archiving-and-license-to-publish"](https://www.nature.com/nature-portfolio/editorial-policies/self-archiving-and-license-to-publish). Those licensing terms will supersede any other terms that the author or any third party may assert apply to any version of the manuscript.

If you have not already done so, we invite you to upload the step-by-step protocols used in this manuscript to the Protocols Exchange, part of our on-line web resource, natureprotocols.com. If you complete the upload by the time you receive your manuscript proofs, we can insert links in your article that lead directly to the protocol details. Your protocol will be made freely available upon publication of your paper. By participating in natureprotocols.com, you are enabling researchers to more readily reproduce or adapt the methodology you use. [Natureprotocols.com](http://natureprotocols.com) is fully searchable, providing your protocols and paper with increased utility and visibility. Please submit your protocol to <https://protocolexchange.researchsquare.com/>. After entering your nature.com username and password you will need to enter your manuscript number (NG-A62867R1). Further information can be found at <https://www.nature.com/nature-portfolio/editorial-policies/reporting-standards#protocols>

Sincerely,
Wei

Wei Li, PhD
Senior Editor
Nature Genetics
New York, NY 10004, USA
www.nature.com/ng